



# Total Column Optical Depths Retrieved from CALIPSO Lidar Ocean Surface Backscatter

Robert A. Ryan[1], Mark A. Vaughan[2], Sharon D. Rodier[3], Jason L. Tackett[2], John A. Reagan[4], Richard A. Ferrare[2], Johnathan W. Hair[2], Brian J. Getzewich[2]

[1]Coherent Application, Inc. - Psionic LLC, Hampton, VA, USA
[2]NASA Langley Research Center, Hampton, VA, USA
[3]ADNET System Inc., Bethesda, MD, USA
[4]Department of Electrical and Computer Engineering, University of Arizona, Tucson, AZ, USA

*Correspondence to*: Robert A. Ryan (robert.a.ryan@nasa.gov)



**Abstract.** This paper introduces the new Ocean Derived Column Optical Depth (ODCOD) algorithm. ODCOD is now being used to retrieve column optical depths from the 532 nm measurements acquired by the Cloud-Aerosol Lidar with Orthogonal Polarization (CALIOP) onboard the Cloud-Aerosol Lidar Infrared Pathfinder Satellite Observations (CALIPSO) spacecraft. ODCOD retrieves total column optical depths using the lidar backscatter signal return from the ocean surface, together with collocated wind speed estimates from Modern-Era Retrospective analysis for Research and Applications, Version 2

(MERRA-2). An advantage of ODCOD retrievals is that the column optical depths include contributions from particulates throughout the entire column including regions with attenuated backscatter below the CALIOP layer detection thresholds. In contrast, the standard CALIOP processing only estimates optical depths for clouds and aerosols detected by the CALIOP layer detection scheme. In this paper we describe the ODCOD algorithm, develop uncertainty estimates, and characterize the ODCOD retrievals relative to existing datasets. The paper presents detailed comparisons of ODCOD retrievals to collocated

measurements from Langley Research Center's airborne high spectral resolution lidars (HSRL), daytime estimates derived from Moderate Resolution Imaging Spectroradiometer (MODIS), and daytime and nighttime optical depths estimates from the Synergized Optical Depth of Aerosols (SODA) algorithm. ODCOD aerosol-only optical depth estimates are higher compared to airborne HSRL measurements by $0.009 \pm 0.043$ (median $\pm$ median absolute deviation) or $6\,\% \pm 27\,\%$ relative difference, lower than MODIS by $-0.009 \pm 0.041$ ($8.0\,\% \pm 34\,\%$ relative difference), higher in the daytime than SODA by

$0.004 \pm 0.035$ ($12\,\% \pm 34\,\%$ relative difference), and higher in the nighttime by $0.027 \pm 0.034$ ($20\,\% \pm 33\,\%$ relative difference). In addition to being a new method of retrieving column optical depth, ODCOD's estimates are independent from the standard CALIOP optical depth retrieval algorithms and have potential for further advances in the CALIPSO data record both to validate CALIOP estimates and as a potential column constraint for future improvements to extinction retrievals.

## 1 Introduction

The Cloud-Aerosol Lidar with Orthogonal Polarization (CALIOP) (Hunt et al., 2009) aboard the Cloud-Aerosol Lidar Infrared Pathfinder Satellite Observations (CALIPSO) spacecraft (Winker et al., 2010) acquired over 17 years of near continuous observations beginning in mid-June of 2006 and concluded August 1st, 2023. CALIPSO's 98.2° orbit inclination yielded near global coverage, allowing for measurements of the location, extent, and optical properties of clouds and aerosols from 82° S to 82° N. CALIOP transmits linearly polarized laser light at 1064 nm and 532 nm, with detectors for the

total backscattered signal at 1064 nm and both the parallel and perpendicular polarizations of the backscatter at 532 nm (Hunt et al., 2009). The calibrated perpendicular and parallel signals are summed to retrieve the total attenuated backscatter at 532 nm and volume depolarization ratios are obtained by dividing the perpendicular signals by the parallel signals (Powell et al., 2009). The science data products retrieved from the CALIOP measurements are reported at three standard processing levels (Vaughan et al., 2023). Level 1 products report calibrated profiles of attenuated backscatter coefficients for all three

measurement channels, along with instrument state parameters (e.g., laser energies and calibration coefficients) and relevant ancillary data such as profiles of atmospheric temperature and pressure. Level 2 products report geophysical parameters





derived from the level 1 calibrated measurements (Winker et al., 2009). These parameters include layer top and base altitudes for all atmospheric and surface features detected in the backscatter profiles (Vaughan et al., 2009) and the identification of atmospheric layers according to type and subtype (Liu et al., 2019; Kim et al., 2018; Avery et al., 2020); and

layer optical properties such as optical depths and vertically resolved profiles of particulate extinction and backscatter coefficients (Young et al., 2018). Level 3 products report monthly averages of level 2 retrievals composited on uniform spatial grids (Tackett et al., 2018; Kar et al., 2019, Winker et al., 2023).

This paper describes a newly implemented algorithm for estimating the total column optical depth from the lidar attenuated

backscatter return from the ocean surface within the CALIOP suite of algorithms that provide the level 2 data products. This new algorithm is called the Ocean Derived Column Optical Depths or ODCOD algorithm. ODCOD is developed from the work of Venkata and Reagan (2016; hereafter, VR2016) and stands on the legacy of other works that have highlighted the potential of lidar measurements of the ocean surface. In Reagan and Zielinskie's 1991 paper, they recognized that they could use "the strong return signals from ground/sea reflections to improve upon information that can be retrieved from spaceborne

lidar observations" (Reagan and Zielinskie, 1991). Several studies have shown the link between surface wind speed, wave slope variance, and sea surface directional reflectance (Menzies et al., 1998; Lancaster et al., 2005; Hu et al., 2008). Josset et al. (2008) developed the innovative, multi-sensor Synergized Optical Depth of Aerosols (SODA) algorithm that combines ocean surface measurements from CloudSat and CALIOP with Advanced Microwave Scanning Radiometer (AMSR) wind speed retrievals to estimate column optical depths of clouds and aerosols. In contrast, ODCOD derives estimates of column

optical depth based solely on CALIOP measurements augmented by wind speeds obtained from Modern-Era Retrospective analysis for Research and Applications, Version 2 (MERRA-2) reanalysis data (Gelaro et al., 2017). Beginning with the Version 4.51 (V4.51) release of the Lidar Level 2 (LL2) CALIPSO data products, ODCOD retrievals of 532 nm total column optical depths are reported wherever a qualified ocean return signal is available.

In the remainder of the paper, we provide a comprehensive overview of the ODCOD retrieval. Section 2 reviews the theory underpinning the algorithm, describes the necessary inputs, the associated profile selection process, and develops uncertainty estimates. Section 3 characterizes ODCOD's performance by making comparisons to collocated airborne high spectral resolution lidar (HSRL) measurements and to the aerosol optical depths (AODs) reported by the Moderate Resolution Imaging Spectroradiometer (MODIS) and SODA data sets. In Sect. 4 we discuss strategies for utilizing ODCOD to improve

the standard CALIOP extinction and optical depth retrievals. In closing, Sect. 5 summarizes the advances made by the ODCOD algorithm and provides some concluding remarks.



## 2 Ocean Derived Column Optical Depth Technique

In this section we describe the details of the ODCOD technique, the required inputs, associated uncertainties, and the contents of the new ODCOD data sets reported in the CALIOP V4.51 LL2 data products.

### 75 2.1 Description of Ocean Derived Column Optical Depth Technique

The ODCOD algorithm estimates the attenuation of laser light through the Earth's atmosphere by comparing the magnitude of the attenuated backscatter return from the ocean surface to an unattenuated modelled surface return. The difference between the measured and modelled return is then used to estimate the atmosphere's two-way transmittance from which an estimate of the atmosphere's optical depth is derived. ODCOD starts by fitting the profile of attenuated backscatter measured

from the ocean surface to a model of the receiver's impulse response function (IRF) output of a lidar laser pulse. Mathematically, this idealized impulse response model (IRM) is shifted in time (i.e., vertically within the atmospheric column) and then scaled in magnitude to achieve the best fit to the CALIOP surface backscatter measurements. Next, an ideal unattenuated model of the ocean surface backscatter is calculated. The particulate two-way transmittance of the atmospheric column above the surface is then derived from the ratio of the integrals (i.e., areas) of the shifted and scaled

IRM and the unattenuated ocean surface backscatter model.

When CALIOP's receiver records a backscatter signal from a hard target such as Earth's surface, the returned signal is a large sudden pulse of photons. The lidar's onboard post-detector electronics employ a third-order low-pass Bessel filter. This filter is downstream of the detectors at both 532 and 1064 nm (VR2016). An important feature of the Bessel low-pass filter is

that it preserves the total energy of an incoming pulse. When a large, narrow pulse enters this filter, its peak amplitude is reduced but the pulse is transformed in the time domain such that the area under the original narrow pulse is preserved. The output of the filter can be characterized by its IRF which describes the output signal when the input signal is an impulse, also known as a delta function. When passed through the onboard low-pass filter, backscattered light from CALIOP's laser pulse produces the low-pass filter IRF. Due to the Bessel filter and IRF effect, hard target returns are spread over multiple adjacent

CALIOP range bins.

CALIOP employs onboard averaging to reduce data storage and downlink size (Hunt et al., 2009). Onboard the satellite, profile data is sampled into bins at 10 MHz or every 0.1 μs, corresponding to a vertical sampling resolution of 15 m. At 532 nm, the 10 MHz samples acquired between 8.2 km above mean sea level and 0.5 km below are averaged over two

vertical bins, yielding a downlink resolution of 30 m. Consequently, even with the filter broadening effect of the IRF, the entire ocean return signal is contained in only three to four downlinked 30 m range bins at 532 nm. These few range bins contain the information about the attenuation overhead in the magnitude of the signal reaching the surface.

To characterize CALIOP's IRF, CALIOP instrument engineers conducted laboratory experiments prior to launch in which
short, high intensity pulses of laser light were measured using the CALIPSO engineering model receiver electronics and
flight-qualified photomultipliers. Since the original model parameters were not published in VR2016, we developed an
analytic model of the IRF using the concepts they outlined. We fit digitizer readings from the engineering model receiver
electronics to a hyperbolic tangent for the rising part and a Gaussian curve for the falling part. Figure 1 shows a typical
example, where the digitizer readings for the impulse response (purple diamonds) are shown plotted as a function of elapsed
time from the pulse onset. The analytic model is set to zero for all times prior to the pulse onset.

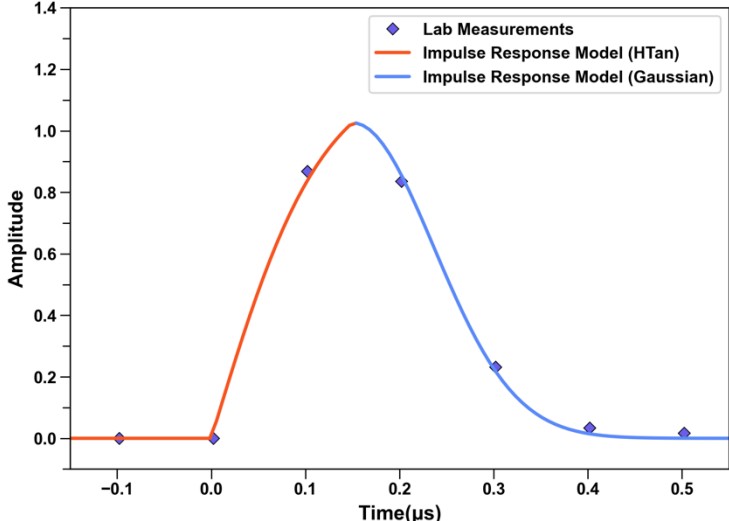

**Figure 1: The CALIOP IRM derived from laboratory measurements. The purple diamonds show lab measurements taken at intervals of 0.1 μs, while the red and blue solid lines show the analytic curve fit to the rising signal prior to the peak and the decaying signal following the peak, respectively.**

VR2016 found that a similar approximation falls within 1 % of expected 3rd order Bessel function IRFs and has similar
characteristics to the CALIOP filter. Because the lidar laser pulse was used to create the IRM, rather than a true delta
function, the IRM is theoretically a closer approximation to a CALIOP hard target return than using a pure delta function
impulse response output. The IRM is used to estimate the magnitude of CALIOP's surface return.

The magnitude of the surface return has a direct relationship to the total two-way transmittance. VR2016 relate the area
under the IRM to the total two-way transmittance with Eq. (1):

$$A_{IRM} = \frac{2T^2(z_s)R_\lambda(\omega,\theta)}{c} \tag{1}$$





In this expression, $A_{IRM}$ is the area under the IRM where $R_\lambda$ is the ocean surface retro or backscatter reflectance which is a function of the wind speed at 10 m above the ocean surface ($\omega$) and the off-nadir angle of the spacecraft ($\theta$), and $c$ is the speed of light. $T^2(z_s)$ is the total two-way transmittance of the atmosphere at the ocean surface, which is the product of the molecular and ozone two-way transmittances ($T_M^2(z)$), and the particulate two-way transmittance ($T_P^2(z)$). Rearranging this equation yields an expression for the two-way transmission due to particulates alone, as shown in Eq. (2). This is the governing equation for ODCOD, from which optical depths are retrieved.


$$T_P^2(z_s) = \frac{cA_{IRM}}{2R_\lambda(\omega,\theta)T_M^2(z_s)} \tag{2}$$

As in standard CALIPSO processing, $T_M^2(z_s)$ is estimated from molecular and ozone number densities obtained from the MERRA-2 model (Kar et al., 2018). VR2016 relate $R_\lambda$ to wind speed using a revised and updated version of the ocean surface backscatter reflectance model developed by Lancaster et al. (2005) using Eq. (3):


$$R_\lambda(\omega,\theta) = \big(1 - W(\omega)\big)F_\lambda(\omega,\theta) + 0.2W(\omega) \tag{3}$$

where $W$ is the fraction of the surface covered with whitecaps, $F_\lambda$ is the Fresnel retro reflectance of ocean water, and the factor of 0.2 is the estimated Fresnel retro reflectance of ocean whitecaps. The whitecap fraction $W$ is estimated by Eq. (4):

$$W(\omega) = 2.95 \times 10^{-6}(\omega)^{3.37} \tag{4}$$


The whitecap fraction model used by ODCOD is the empirical fit found in Lancaster et al. (2005) but with updated coefficients provided by VR2016. The ocean surface backscatter reflectance model is comprised of an ocean water component and a white cap component. As wind speeds increase, the fraction of the total ocean surface backscatter reflectance attributed to white caps increases significantly. Furthermore, the magnitude of $F_\lambda$ varies as a function of off-nadir

angle, which is relevant for CALIOP calculations. At launch, the CALIPSO off-nadir angle was fixed at 0.3°. However, the off-nadir angle was increased to 3.0° in November 2007 (Winker et al., 2009) and 3.0° continued as the nominal off-nadir angle until the end of the mission. In Eq. (5), we adopt the estimate of $F_\lambda$ as a function of wind speed and off-nadir angle from Josset et al. (2010b):

$$F_\lambda(\omega,\theta) = \frac{\xi_\lambda e^{-\left(\frac{\tan^2\theta}{\langle s(\omega)\rangle^2}\right)}}{4\pi\langle s(\omega)\rangle^2 \cos^5\theta} \tag{5}$$




where $\xi_\lambda$ is the Fresnel coefficient at wavelength $\lambda$, which we estimate as 0.0213 at 532 nm and 0.0202 at 1064 nm (Vaughan et al., 2019), and $\langle s(\omega)\rangle^2$ is the wave slope variance which is also a function of wind speed. While VR2016 used the wave slope variance approximation proposed in Lancaster et al. (2005), for ODCOD we have chosen to use the piecewise approximation developed by Hu et al. (2008). This approximation was developed using CALIOP measurements
and AMSR wind speeds and directly relates the two primary quantities used in the ODCOD retrieval and is shown in Eq. (6):

$$
\begin{aligned}
\omega < 7\,\tfrac{m}{s} \qquad\qquad & \langle s(\omega)\rangle^2 = 1.46 \times 10^{-2}\sqrt{\omega} \\[6pt]
7\,\tfrac{m}{s} \le \omega < 13.3\,\tfrac{m}{s} \qquad\qquad & \langle s(\omega)\rangle^2 = 0.003 + 5.12 \times 10^{-3}\omega \\[6pt]
13.3\,\tfrac{m}{s} \le \omega \qquad\qquad & \langle s(\omega)\rangle^2 = 0.138\, log_{10}(\omega) - 0.084
\end{aligned}
\tag{6}
$$

As illustrated in Fig. 1, ODCOD approximates the IRM using the piecewise analytic function given in Eq. (7).

$$
f_{IRM}(t_s) = \alpha \times
\begin{cases}
0 & for : t_s \le 0.0\mu s \\
a_h\, tanh(\omega_h t_s) & for : 0.0\mu s < t_s \le b_p \\
a_g e^{-(\omega_g[t_s - g])^2} & for : t_s > b_p
\end{cases}
\tag{7}
$$


Integrating this function yields the requisite $A_{IRM}$ estimates. Table 1 shows the parameters which define the ODCOD IRM and were selected to fit the lab measurements.

| symbol | interpretation | value |
|---|---|---|
| $\alpha$ | Scale Factor | Unknown; calculated during the curve fitting process |
| $t_s$ | Sample Time Delay | Unknown; calculated during the curve fitting process |
| $a_h$ | TANH Vertical Scale Factor | 1.14 |
| $a_g$ | Gaussian Vertical Scale Factor | 0.9695 |
| $\omega_h$ | TANH Horizontal Scale Factor | 8.39 |
| $\omega_g$ | Gaussian Horizontal Scale Factor | 8.186 |
| $g$ | Gaussian Horizontal Shift | 0.15 $\mu s$ |
| $b_p$ | Piecewise Function Breakpoint | 0.15 $\mu s$ |

Table 1: Piecewise variables and values used to generate the ODCOD impulse response model.





The first step in fitting the IRM to the measured ocean surface attenuated backscatter signal is to find the sample time delay, $t_s$. The sample time delay is the temporal offset between the actual ocean surface signal onset and the midpoint of the CALIOP onboard or downlinked sample range bin. Because the time delay is not a function of the IRM, there is not a unique IRM value for every time delay within the surface pulse. Instead, the ratio of two consecutive samples is used which does have a unique value for any time delay within the surface return.


By using the ratio of two points, the scale factor term $\alpha$ cancels out and we can use an unscaled IRM to find two samples that match the ratio of two known points. However, we don't know the magnitude of the original onboard samples and instead only know the downlinked samples. The ratio of two consecutive downlinked samples can also be used in the same way but we must create a mapping of the IRM to an onboard averaged IRM. Equation (8) shows the relationship of the IRM to the

downlinked samples and can be thought of as a downlinked IRM (DIRM).

$$f_{DIRM}(t_s) = \frac{\left(f_{IRM}(t_s - 0.05\ \mu s) + f_{IRM}(t_s + 0.05\ \mu s)\right)}{2} \tag{8}$$

By using the downlinked samples and because the downlinked sample ratio is a function of the IRM, the scale factor still cancels, and we can then find the time delay of the original onboard samples. To do this, we use a simple bisecting search

algorithm. Figure 2 shows how the IRM, DIRM, and the ratio of two consecutive DIRM samples are all connected by the sample time delay. The annotated aqua circles are the IRM at $t_s - 0.05\ \mu s$ and $t_s + 0.05\ \mu s$ and are averaged to make the purple diamond of the DIRM at $t_s$. The time delay of $t_s$ has a unique orange diamond DIRM ratio value. All time delays after the onset of the DIRM have unique values and can be used to solve for the time delay.





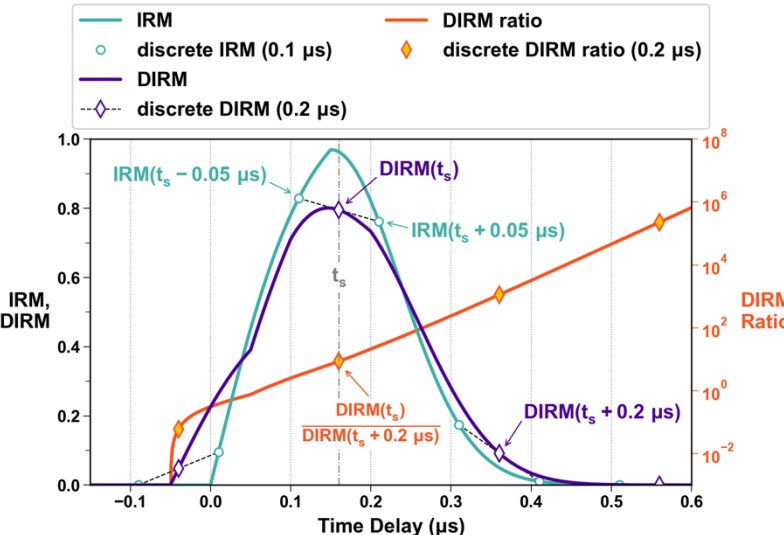


**Figure 2: The CALIOP IRM (aqua line) with one possible set of discrete samples (circles) and the resulting DIRM (purple) with the downlinked samples (purple diamonds) that result from the discrete samples of the IRM linked (black dashed line). Overplotted on the right axis is the ratio of the DIRM (orange) at time delay $t_s$ over the DIRM at time delay $t_s + 1$ with the discrete ratios (orange diamonds). Highlighted by the annotations are the IRM, DIRM and DIRM ratio for the specific time delay $t_s$ annotated and marked**
**by the vertical line to show how all three tie together.**

Two adjacent test samples must be selected to take the ratio of and find the time delay. If the CALIOP surface detection algorithm has correctly identified the surface top and base altitudes, at least two vertically adjacent downlinked samples should be part of the surface return signal. In an ideal signal, any measurement that is not part of the surface return would be completely zero and any measurement on or after the surface return onset would be non-zero. However, the measured

downlinked samples before the surface return onset are rarely if ever actually zero so it is critical that whatever consecutive sample pair is selected is part of the surface return. To ensure the test samples are part of the surface return, the largest measurement of the suspected surface return and the largest adjacent measurement are chosen as the test samples. These points are assumed to have at least one onboard sample that is part of the surface return due to their magnitude and have the best signal-to-noise ratio which will minimize error in finding the time delay due to noise. If the number of downlinked

samples between the CALIOP detected surface top and base exceed four range bins, a surface detection failure is suspected, and no retrieval is attempted.

If a downlinked sample's time delay places it before the onset of the IRM (i.e. a time delay less than zero), that sample may still be part of the surface return as long as one of the averaged onboard samples is part of the surface. A minimum time

delay can be calculated for the DIRM as a function of the number of onboard samples averaged and is calculated by $t_{min} = T_s(1 - n)/2$ and $t_{max} = T_s(1 + n)/2$ where $T_s$ is 0.1 µs and $n$ is the number of samples averaged onboard (i.e. two for


CALIOP at 532 nm). Based on the derived time delay, we calculate which sample of the surface return the test sample is by adding or subtracting the time between range bins until the time delay falls within $t_{min}$ and $t_{max}$.

Since the time delay tells us which points of the surface return have been found, if the CALIOP surface detection algorithm is missing points before or after the identified surface measurements, ODCOD will include them to ensure the correct surface points are used. These actions only affect ODCOD and do not attempt to alter CALIOP's reported surface detections.

The scale factor is found by starting with a scale factor of $1 \times 10^{-5}$ and incrementally increasing this value until the sum of
the squared error (SSE) between the measured samples of the surface return and the matching DIRM samples is no longer reduced. Then a bisecting search algorithm is used to find a SSE less than $1 \times 10^{-5}$ or successive scale factors have a relative difference of less than $1 \times 10^{-5}$. The remaining differences in the samples are used to estimate the uncertainty in the area under the IRM.

With the time delay and scale factor determined, the integral of the IRM is simply the integral of the two parts of the unscaled IRM model multiplied by the scale factor, as shown in Eq. (9):

$$A_{IRM} = \alpha \left( a_h \int_{0.0\mu s}^{b_p} \tanh(\omega_h t) \, dt + a_g \int_{b_p}^{\infty} e^{-(\omega_g [t-g])^2} \, dt \right) \tag{9}$$

The final pieces of the equation are the off-nadir angle, which is known from the spacecraft attitude data, and the horizontal
wind speed magnitude at 10 m above the ocean surface, which is obtained from the MERRA-2 model. We have found however, that the MERRA-2 10 m winds over the ocean are biased low and require a correction to obtain unbiased optical depth estimates. An in-depth discussion of the wind speed bias correction is given in Sect. 2.2.1. These corrected MERRA-2 winds are used in the ODCOD algorithm.

Using the IRM has some advantages over techniques that use the integrated attenuated backscatter of the surface when estimating the area of the surface return. Namely, the IRM does not significantly include contributions from light scattered from below the ocean surface. The 532 nm light is largely extinguished within only a few tens of meters in the ocean (VR2016) but will make a small contribution to the overall return. Significant scattering from below the air-ocean interface would effectively widen the surface pulse but not introduce a uniform enhancement of the measurements. Some subsurface
enhancement can occur in the individual measurements as a function of their respective time delays, but that enhancement is assumed to be less significant than the overall effect. Furthermore, since only the two largest points of the return are used in the fitting process, any enhancement in the tail of the return is immaterial. Any subsurface enhancement of the area under the



fit IRM is assumed to be insignificant compared to other uncertainties of the algorithm and at most will cause a small bias in the retrieval.


In addition, using the IRM overcomes possible effects from the non-ideal transient recovery by the 532 nm photomultipliers following very strong backscatter signals (Hunt et al., 2009). As explained by McGill et al. (2007), "following a strong impulse signal, such as from the Earth's surface or a dense cloud, the signal initially falls off as expected but at some point, begins decaying at a slower rate that is approximately exponential with respect to time." Because ODCOD only fits the

largest points of the surface return, the tail effect phenomenon will not affect the surface return fit. We assume any enhancement from the tail effect in the design of the IRM is insignificant compared to the surface return.

Lastly, by using only information already available in CALIPSO standard processing, ODCOD avoids many of the uncertainties associated with using external measurements from other instruments to attempt a similar retrieval. Note that the

ODCOD retrieval can in principle be applied on the CALIOP 1064 nm channel. However, additional accommodations would need to be made to account for the coarser downlink resolution (60 m at 1064 nm versus 30 m at 532 nm). Implementing ODCOD at 1064 nm is currently under investigation by the CALIOP team.

**2.2 Algorithm Inputs and Uncertainties**

This section examines the necessary inputs to the ODCOD algorithm, their uncertainties, and how those uncertainties affect

the uncertainty of the retrieval. Ancillary inputs are also discussed along with their uses in data filtering and quality assurance.

Applying standard propagation of error techniques (Bevington and Robinson, 1992), the estimated variance in the ODCOD two-way transmittance is shown in Eq. (10).


$$\sigma_{T_P^2}^2 = \left(\frac{\delta T_P^2}{\delta \omega}\right)^2 \sigma_\omega^2 + \left(\frac{\delta T_P^2}{\delta \theta}\right)^2 \sigma_\theta^2 + \left(\frac{\delta T_P^2}{\delta A_{IRM}}\right)^2 \sigma_{A_{IRM}}^2 + \left(\frac{\delta T_P^2}{\delta T_M^2}\right)^2 \sigma_{T_M^2}^2 \qquad (10)$$

The variance of the off-nadir angle, molecular and ozone two-way transmittances, and the estimate of the Fresnel coefficients are assumed to be insignificant compared to that of wind speed and area fit. We choose to ignore these uncertainties in the ODCOD uncertainty estimate so Eq. (10) simplifies to Eq. (11).


$$\sigma_{T_P^2}^2 = \left(\frac{\delta T_P^2}{\delta \omega}\right)^2 \sigma_\omega^2 + \left(\frac{\delta T_P^2}{\delta A_{IRM}}\right)^2 \sigma_{A_{IRM}}^2 \qquad (11)$$



To isolate the change in the two-way transmittance variance due to wind speed arising from multiple terms, we expand the wind speed term in Eq. (11) as shown in Eq. (12) through (14):

$$\frac{\delta T_P^2}{\delta\omega} = \frac{-c \cdot A_{IRM}}{2 \cdot R_\lambda{}^2 \cdot T_{mol}^2 \cdot T_{O_3}^2} \cdot \frac{\delta R_\lambda}{\delta\omega} \tag{12}$$

$$\frac{\delta R_\lambda}{\delta\omega} = (0.2 - F_\lambda)(2.95 \times 10^{-6}(3.37)\omega^{2.37}) + (1 - W)\frac{\delta F_\lambda}{\delta\omega} \tag{13}$$

$$\frac{\delta F_{532}}{\delta\omega} = \xi_{532} \cdot exp\left(-\frac{tan^2\,\theta}{\langle s\rangle^2}\right) \cdot \frac{tan^2\,\theta - \langle s\rangle^2}{4\pi(\langle s\rangle^2)^3\,cos^5\,\theta} \cdot \frac{\delta\langle s\rangle^2}{\delta\omega} \tag{14}$$


The three parts of Eq. (6) relate wind speed to wave slope variance. The change in wave slope variance as a function of wind speed is derived using the three parts of Eq. (15).

$$\omega < 7\frac{m}{s} \qquad \frac{\delta\langle s\rangle^2}{\delta\omega} = \frac{1.46\times10^{-2}}{2\sqrt{\omega}}$$

$$7\frac{m}{s} \leq \omega < 13.3\frac{m}{s} \qquad \frac{\delta\langle s\rangle^2}{\delta\omega} = 5.12 \times 10^{-3} \tag{15}$$

$$13.3\frac{m}{s} \leq \omega \qquad \frac{\delta\langle s\rangle^2}{\delta\omega} = \frac{0.138}{\omega\,ln(10)}$$

The partial derivative of two-way transmittance with respect to area under the IRM is shown in Eq. (16).

$$\frac{\delta T_P^2}{\delta A_{IRM}} = \frac{c}{2R_\lambda(\omega,\theta)T_M^2(z_s)} \tag{16}$$

To estimate the variance in the ODCOD two-way transmittance, estimates of the variance in wind speed and area of the shifted and scaled IRM are required and discussed in the next two sections. Finally, the uncertainty in optical depth can be calculated as:


$$\sigma_\tau = \sqrt{\frac{\sigma_{T_P^2}^2}{(2T_P^2)^2}} \tag{17}$$

If we recall Eq. (6) for the wave slope variance, we know there are two distinct discontinuities in the piecewise function at 7 m s[-1] and 13.3 m s[-1]. Due to the discontinuities, there are also discontinuities in the analytical estimation of the uncertainty in the optical depths retrieved by ODCOD due to Eq. (15) being part of the overall uncertainty estimate. These





discontinuities can easily be seen in the 5km ODCOD estimation of standard deviation as calculated by Eq. (17) when
binned by wind speed (Fig. 3(a)). This is unfortunate, as the uncertainties for retrievals with wind speed values near the
discontinuities will make the uncertainty estimate less certain. Nonetheless, extensive comparisons of ODCOD aerosol-only
optical depths to other data sets demonstrate that in general, ODCOD uncertainty estimates provide a reliable estimate of the
overall uncertainty in the ODCOD retrieval (Sect. 4).

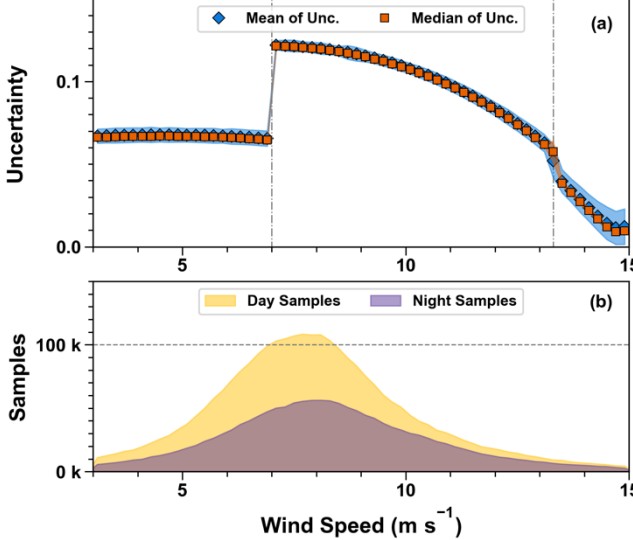


**Figure 3: Panel (a) shows ODCOD 5 km AOD standard deviation estimate means (blue diamonds) calculated by Eq. (17) with
standard deviation envelope and medians (orange squares) with median absolute deviation (MAD) envelope binned by wind speed.
Grey reference lines at 7 and 13.3 m s-1 coincide with the breakpoints in the wave slope variance model in Eq. (6). Panel (b) shows
the number of samples for daytime (yellow) and nighttime (purple). Both panels represent the data for March 2008–February 2011.**






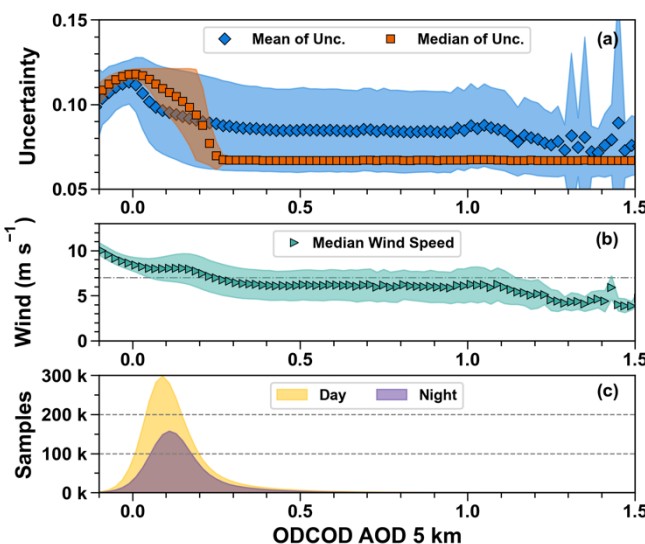

**Figure 4: Panel (a) shows ODCOD 5 km AOD standard deviation estimate means (blue diamonds) calculated by Eq. (17) with standard deviation envelope and medians (orange squares) with median absolute deviation (MAD) envelope and binned by ODCOD 5 km AOD. Panel (b) shows the median wind speed with MAD envelope for the same data in Panel (a) with a grey reference line at 7 m s-1. Panel (c) shows the number of samples in daytime (yellow) and nighttime (purple). All panels represent the data for March 2008–February 2011.**

To understand the major impacts on AOD uncertainty in Fig. 4(a), we mainly need to consider the wind speeds in these regimes and how they affect the uncertainty overall. The uncertainty due to $A_{IRM}$ only plays a minor role and is mostly responsible for the smaller fluctuations of the values in Fig. 3(a). Figure 4(b) shows that generally, the median wind speeds fall between 5–8 m s$^{-1}$, with only AOD values below zero and above about 1.1 being much outside the range. The larger wind speeds for AOD below 0 are attributed to the fact that at lower wind speeds, the ocean is more reflective and has a higher chance of saturating the detectors at very low optical depths. Because ODCOD retrievals are not performed for saturated surface returns the returns that qualify for attempting ODCOD will statistically have higher wind speeds. This surface-saturation sampling bias is discussed in more detail in Sect. 3.1.1. Figure 3(a) shows that uncertainty increases substantially for wind speeds just above the 7 m s$^{-1}$ threshold as compared to lower wind speeds. This means that for the distribution of wind speeds found in nature, there will be a bimodal distribution of uncertainty values for ODCOD optical depths, with a tight distribution around 0.065 and a wider distribution around the 0.12. The fraction of each distribution in any collection of data binned by AOD will depend on the standard deviation of the wind speeds and how close to the threshold value the wind speed distribution is. It is the bimodal nature of these uncertainties that explain the large features of the ODCOD uncertainty landscape.

We will now analyze a few of the specific largescale features of the uncertainty distribution more closely. Starting on the left of Fig. 4(a), while it only represents a small fraction of the data, the uncertainties of negative AOD retrievals are smaller than





AODs near 0. Negative optical depths typically occur due to noise in the surface return measurements of optically thin
atmospheric conditions. Noise in the measurements can cause the IRM fit to over or underestimate the surface return but
when the surface return is estimated too large and the optical depth is nearly zero, the retrieved estimate can be less than
zero. In optically thin atmospheric conditions, the surface return can become bright enough to saturate the detectors and
when the wind speed is low, the ocean surface reflectivity is increased which further increases the surface return and
exacerbates the saturation. This means successful retrievals at very low optical depths occur preferentially at higher wind
speeds. Higher wind speeds are where the contribution of uncertainty due to wind speed is smaller compared to that of lower
wind speed values. In Fig. 4(b), we see the wind speed ranges drop from 10 m s$^{-1}$ down to about 8 m s$^{-1}$ which aligns on
Fig. 3(a) with uncertainty values of less than 0.11 up to 0.12 which is the same as what is shown on Fig. 4(a).

The mean and median uncertainties separate for AOD values from 0 to 0.25 because there is a mixture of retrievals with
wind speeds above and below the 7 m/s threshold, thereby yielding higher or lower uncertainties. Figure 4(b) shows the wind
speed median is about 8 m s$^{-1}$ but then drops below the 7 m s$^{-1}$ threshold by 0.25 AOD approaching a constant 6 m s$^{-1}$. Again,
this is consistent with Fig. 3(a) where near the 7 m s$^{-1}$ threshold the uncertainty is basically chosen from either about 0.12
when the wind speed is above the threshold and about 0.065 when below.

Finally, the strong skew and relatively constant mean and standard deviation for AOD greater than 0.25 is because the wind
speeds for these retrievals are more often below 7 m s$^{-1}$. The median being a nearly constant value of 0.065 with a median
absolute deviation (MAD) of around 0.001 yet the mean being 0.085 with a standard deviation of about 0.025 is indicative of
the strongly bimodal distribution of uncertainties.

In the CALIOP data products, ODCOD uncertainties are reported as the optical depth estimated standard deviation derived
according to Eq. (17). In general, ODCOD filtered for wind speeds between 3–15 m s$^{-1}$ has an uncertainty of 0.12 ± 0.05 (76
% ± 40% relative uncertainty). Daytime uncertainty is 0.11 ± 0.01 (79 % ± 45 % relative) and nighttime is 0.11 ± 0.01 (71 %
± 34 % relative). Despite the discontinuities and difficulties in making the estimation, ODCOD's reported standard deviation
estimates are a good approximation of the standard deviation in the retrieved optical depth.

**2.2.1 Wind Speed**

Wind speed is the largest source of uncertainty in the ODCOD algorithm; however, MERRA-2 does not provide standard
deviation estimates for the wind speeds reported in their product. We have therefore adopted a global wind speed uncertainty
based on the available literature. According to Archer and Jacobson (2005), "the global average 10 m wind speed over the
ocean from measurements is 6.64 m s$^{-1}$." Similarly, Wentz et al. (2005) report a maximum standard deviation of AMSRE



wind speeds relative to ocean buoy measurements of less than 1.00 m s⁻¹. From these measurements we derive an estimated

relative wind speed standard deviation of $\epsilon_{\omega_M} = 1.00 \, m \, s^{-1} / 6.64 \, m \, s^{-1} \approx 0.151$.

In addition to random uncertainties, another issue that must be addressed is potential bias in the MERRA-2 winds at 10 m

above the ocean surface. Compared to buoy and other in situ measurements, Carvalho (2019) found a low bias in MERRA-2

winds from -0.16 to -0.83 m s⁻¹ over ocean. We also found a similar bias of -0.52 ± 0.53 m s⁻¹ (global average) during our

development of a correction to mitigate MERRA-2 wind speed biases, detailed below. To demonstrate the significance of a

wind speed bias, a low wind speed bias of -0.5 m s⁻¹ will cause a high bias of approximately 0.02 in an ODCOD optical

depth.

To develop a wind speed correction, we analyzed the 10 m medium frequency surface wind speed reported by the Advanced

Microwave Scanning Radiometer for Earth Observing System (AMSRE) aboard the NASA Aqua Satellite from June 2006 to

October 4, 2011, and the Advanced Microwave Scanning Radiometer 2 (AMSR2) aboard the Global Change Observation

Mission - Water (GCOM-W1) satellite from May 2012 through December 2020. The AMSR data sets were chosen due to

the respective spacecraft's approximately 90 s separation from the CALIPSO spacecraft. Their proximity allows for near

instantaneous coincident measurements between CALIOP and the AMSR instruments. When comparing the two AMSR

instruments, no significant bias was found so we use aggregated AMSRE and AMSR2 data to cover nearly the entire

CALIPSO mission. These wind speeds were collocated to the 333 m CALIOP footprint and compared to the GMAO

MERRA-2 10 m surface wind speeds retrieved from the inst2d_met data parameters and reported the CALIOP LL2 V4.51

data products.


From the median differences calculated, we generated 14 lookup tables consisting of wind speed ranges from 1 m s⁻¹ to

41 m s⁻¹ in 3 m s⁻¹ increments. Each table is a three dimensional 164 by 180 by 12 lookup table comprised of one-degree

latitudes from 82° S to 82° N, two-degree longitudes, and month of the year for the respective wind speed range. We

calculated the median wind speed differences between the AMSR instruments, and the MERRA-2 reported wind speeds for

375    each month. To interpolate a wind speed bias correction for any wind speed, latitude, longitude, and time, we required each

grid box in the lookup table to have a minimum of five observations to average. If enough data was not available for any

given grid box, we re-binned the data onto larger-sized grids until the minimum data requirement was met and then

populated all the fine grid boxes with the coarser grid value to fill in the map. This technique allows the wind speed bias

values, which are only available over ocean, to be populated over land. Over land values are rarely, if ever, used and then

380    only on those occasions where ODCOD is attempting a retrieval very near shore. This bias correction produces a more

AMSR-like wind estimate.





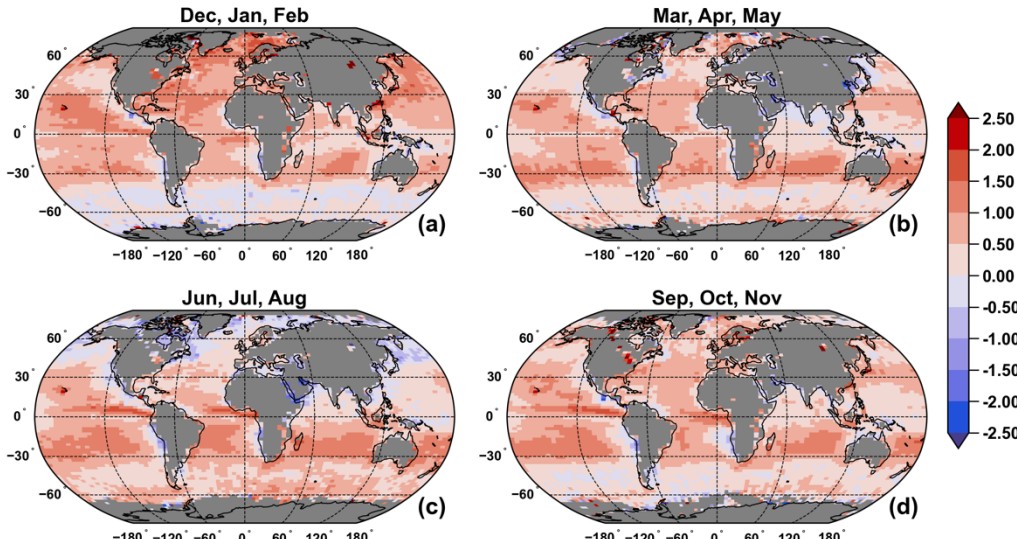

**Figure 5: MERRA-2 wind speed corrections applied to wind speeds between 3–15 m s⁻¹ used in ODCOD 5 km aerosol-only retrievals**
385   **for each of the four seasons, March 2008 through February 2011.**

Figure 5 shows the median additive corrections applied to the MERRA-2 wind speeds between 3–15 m s⁻¹ used in ODCOD

5 km aerosol-only retrievals and illustrates that in nearly all regions an addition is required to correct the low bias in

MERRA-2 winds. The latitude, longitude, season, and MERRA-2 wind speed of the CALIOP profile is used to calculate the

bias correction by four-dimensional linear interpolation. In this way, a correction can be derived for any observation along

the CALIPSO orbit track.

To estimate the uncertainty in the derived correction, analysis of the means and standard deviations in the AMSRE and

AMSR2 data find the mean relative standard deviation for the correction factors as $\epsilon_{\omega_A} = 0.2537$. Since the correction

factor is an additive bias correction, the overall relative uncertainty in the wind speeds used for ODCOD is estimated by

Eq. (18).

$$\epsilon_\omega = \sqrt{\epsilon_{\omega_M}^2 + \epsilon_{\omega_A}^2} = \sqrt{(0.151)^2 + (0.2537)^2} = 0.2950 \tag{18}$$

This means the variance is $\sigma_\omega^2 = \epsilon_\omega^2 \cdot (\omega_M + C_A)^2$ where $C_A$ is the AMSR derived wind speed bias correction. Since wind

speed is used to estimate both wave slope variance and whitecap fraction, wind speed plays a significant role in the

uncertainty estimate.





Wind speed is also used as a filtering criterion. Within the ODCOD algorithm, surface returns where wind speeds fall outside of the inclusive range (0.025–43) m s$^{-1}$ are not attempted. Estimating the reflectivity and fraction of whitecaps with Eqs. (3) and (4) introduces more uncertainty into the model but no analysis has been done at this time to determine the

magnitude of that uncertainty. More discussion on filtering ODCOD data is found in Sect. 3.1.1.

### 2.2.2 Surface Return Area

Another source of uncertainty in the ODCOD algorithm comes from fitting the measured values of the surface return signal to the DIRM. This uncertainty arises from the measurements, which include both random noise and systematic biases such as calibration uncertainty and potential subsurface contributions to the signals, and from possible errors in the IRM

representation of the true IRF output. The variance in the area is estimated from the squared differences between the measured surface return and the time shifted and scaled DIRM. We assume that all mentioned sources of uncertainty are captured by this fit error. Since the area is the integral of the IRM multiplied by the scale factor determined by the retrieval, the variance of the area is given by Eq. (19).

$$\sigma^2_{A_{IRM}} = \left(C_h + C_g\right)^2 \sigma^2_\alpha \tag{19}$$


The constants $C_h$ and $C_g$ are the integrals of the hyperbolic tangent function from time delay zero to $b_p$ and the Gaussian function from $b_p$ to infinity. The variance of the scale factor $\sigma^2_\alpha$ is the mean squared error of the fit of the measured points of the ocean surface to the DIRM.

### 2.2.3 Additional Screening Inputs

Other inputs to the algorithm include the depolarization ratio of the surface return, the negative signal anomaly flag, and the surface saturation flags found in the CALIOP data products. ODCOD requires the surface depolarization ratio to be below 0.15 for a retrieval to be attempted. This threshold is an attempt to ensure that any surface returns with significant sea ice or surface debris are not attempted (Lu et al., 2017). This threshold may also filter out some shallow water cases where significant ocean bottom return could contaminate the retrieval, however a more stringent requirement may be desired if

using coastal or shallow water data. Surface depolarization ratios are reported for all surfaces detected in the CALIOP V4.51 LL2 data products.

Other considerations are the negative signal anomaly (NSA) and surface saturation. The NSA occurs when an unusually large negative attenuated backscatter coefficient is measured in a range bin immediately preceding a very large positive

backscatter return from a strongly scattering target such as Earth's surface (Tackett et al., 2018). Since NSA cannot be determined whether they properly belong to the surface return or to the atmospheric return immediately above, these events





are excluded from ODCOD processing. Also excluded are any profiles where surface saturation is detected. Surface saturation occurs when the magnitude of the signal received by the lidar detectors exceeds the maximum value the detectors can accurately measure. Surface saturation would lead to an improper fit of the IRM, and underestimate of $A_{IRM}$, and a high

bias in the estimate of the column optical depth. Saturated surfaces are identified when the surface saturation flags reported in the LL1 product indicate surface saturation or possibly saturated data.

**2.3 ODCOD Output Products**

CALIPSO V4.51 LL2 data products report 532 nm ODCOD effective optical depth estimates and uncertainties at 333 m, 1 km, and 5 km resolution as well as the MERRA-2 10 m wind speed components, the wind speed correction values, and an

ODCOD QC flag. To calculate the wind speed used by ODCOD from these data products, a user must calculate the wind speed magnitude from the components and add the reported correction value.

The ODCOD QC flag is a 32-bit unsigned integer where each bit used has a specific meaning, as described in Appendix A. The flag is designed such that when interpreted as an integer value, any QC flag value below 64 is an attempted retrieval and

the data could conceivably be used although further quality filtering as described in Sect. 3.1.1 should be considered.

**3 Performance Assessment**

In this section we assess ODCOD retrievals by comparing ODCOD aerosol optical depths (AOD) to collocated AOD retrievals from airborne High Spectral Resolution Lidar (HSRL) measurements and seasonally averaged AOD from two independent satellite-based retrieval techniques. To facilitate comparisons that are meaningful for scientific interpretation,

data quality filtering is applied to each of the data sets to exclude suspected poor quality and anomalous data. The data selection procedure is described first for all data sets in Sect. 3.1, followed by the comparison analysis in Sect. 3.2.

**3.1 Data Selection**

**3.1.1 ODCOD Data Selection**

Unless otherwise stated, all ODCOD data in the comparisons in Section 3.2 is from March 2008 through February 2011 and

are the latest version 4.51 Lidar Level 1 (LL1) and V4.51 LL2 data products. We select data for aerosol-only profiles, where CALIOP has not detected clouds at any resolution within a given retrieval column. This is important because ODCOD is always a total column optical depth estimate with no way to isolate different feature types in the vertical profile. To select the highest quality ODCOD retrievals, the limitations of the ODCOD models and the CALIOP instrument are considered to guide filtering criteria. Unless otherwise stated, the following data quality filters are applied to all data in the comparisons in





this paper. Selecting retrievals with wind speeds in the range 3–15 m s$^{-1}$, surface integrated attenuated backscatter at 532 nm (SIAB) < 0.0413 sr$^{-1}$ in the daytime or < 0.0353 sr$^{-1}$ in the nighttime, and surface integrated depolarization ratio 532 nm (SIDR) < 0.05 improves agreement of ODCOD to the other datasets presented and should be considered when using the ODCOD product.

Filtering profiles where the wind speeds fall outside the range 3–15 m s$^{-1}$ will retain approximately 94 % of retrievals and provide ocean reflectivity estimates for which the model is mostly in the less complex ocean water regime with only a small fraction of the total reflectivity attributed to white caps. To use nearly exclusively ocean water reflectance, a stricter requirement of 3–8 m s$^{-1}$ could be used, but this restriction only retains on the order of 51 % of retrievals and does not significantly improve how the data compares to other optical depth estimates over the 3–15 m s$^{-1}$ filter.


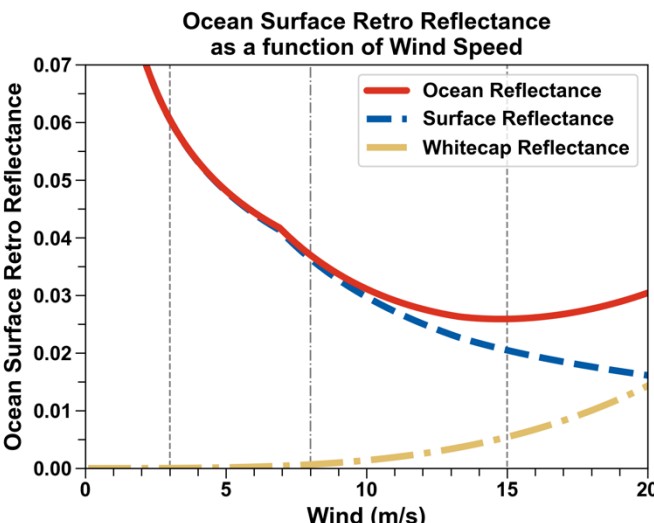

**Figure 6: The ODCOD ocean surface retro reflectance model at 532 nm and off nadir angle of 3° as a function of wind speed with markers at 3 m/s, 8 m/s, and 15 m/s as reference for possible filter thresholds.**

The ODCOD ocean surface reflectance model, described by Eq. (3) and shown in Fig. 6, approaches an asymptotic value as

the wind speed approaches 0 m s$^{-1}$ and as a result increases rapidly when the wind speed drops below 3 m s$^{-1}$. The large change with small variations of wind speed makes the estimate of surface reflectivity in this part of the model significantly less certain. Filtering profiles with wind speeds less than 3 m s$^{-1}$ removes on the order of 7 % to 8 % of profiles and avoids surface reflectivity estimates in the region of the model with higher variability. Furthermore, as wind speeds increase much above 8 m s$^{-1}$, the contribution of whitecaps to ocean reflectance increases. Because instantaneous white cap reflectivity is

known to be a complex amalgam of multiple factors in addition to wind speed (Dierssen, 2019), ODCOD's wind speed-only white cap reflectivity estimate becomes increasingly less certain at higher wind speeds. Filtering profiles with wind speeds



greater than 15 m s$^{-1}$ removes on the order of 1 % to 2 % of retrievals and avoids surface reflectivity estimates with significant contribution from whitecaps.

ODCOD does not attempt a retrieval when the CALIOP surface saturation flags show surface saturation or possibly saturated data. However, because samples are averaged vertically prior to downlink, surface saturation can still go undetected. In this study, we filter profiles with SIAB > 0.0413 sr$^{-1}$ in the daytime and > 0.0353 sr$^{-1}$ in the nighttime which removes approximately 20 % of ODCOD profiles day or night but captures approximately 98 % of profiles where SIAB corresponds to surface saturation (Fig. 7). The upper panels of Fig. 7 show distributions of SIAB versus windspeed for

aerosol-only profiles having valid ODCOD retrievals. The lower panels show SIAB versus windspeed for surface-saturated aerosol-only profiles for which no ODCOD retrieval was attempted. The horizontal red lines seen in all four panels represent the SIAB threshold above which valid ODCOD retrievals are considered suspicious due to possible undetected surface saturation.

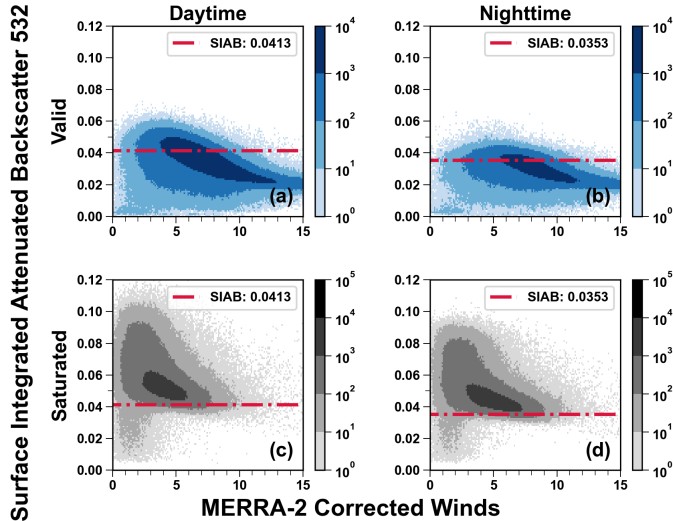

**Figure 7: The top row shows SIAB at 532 nm as a function of wind speed for CALIOP determined aerosol-only 5 km ODCOD valid retrievals for daytime (left) and nighttime (right) measurements; distributions on the bottom row show SIAB as a function of wind speed for surface-saturated aerosol-only 5 km profiles for which ODCOD retrivals are not attempted. The red dashed line shows the proposed thresholds for day and night.**

The magnitude of surface return signal that will saturate the detectors is different in the day and night due to the difference in

CALIOP variable gain amplifier settings (Hostetler et al., 2005). High, and highly variable, solar background signals during daytime operations dictate the use of lower amplifier gains to minimize digitizer overflows in the daytime measurements. Daytime gains are lower by a factor of approximately 6.5, which accounts for the difference in the day and night SIAB thresholds and explains why different distributions can be seen day and night (Fig. 7(a) and Fig. 7(c), and Fig. 7(b) and Fig. 7(d), respectively). On the order of 27 % of daytime aerosol-only profiles over ocean are rejected for ODCOD retrieval due



to flagged surface saturation and that number increases to 43 % at night. Using the median and MAD of the SIAB distribution of saturated profiles shown in Fig. 7(c) and Fig. 7(d), we calculate the thresholds for the SIAB filter from the median of the SIAB (0.0543 daytime and 0.0457 nighttime) minus two times the MAD.

In this study, we also filter profiles with SIDR greater than 0.05. This threshold removes 1 % to 2 % of the ODCOD profiles.

ODCOD does not attempt a retrieval when the SIDR is greater than 0.15 to avoid retrievals over ocean covered by ice. However, small amounts of sea ice, surface debris, and ocean bottom returns in shallow water are all places where the SIDR might be elevated and the ODCOD retrieval would be attempted but may be invalid. Figure 8 shows the distribution of SIDR for CALIOP determined aerosol-only profiles for valid ODCOD retrievals as a function of wind speed.

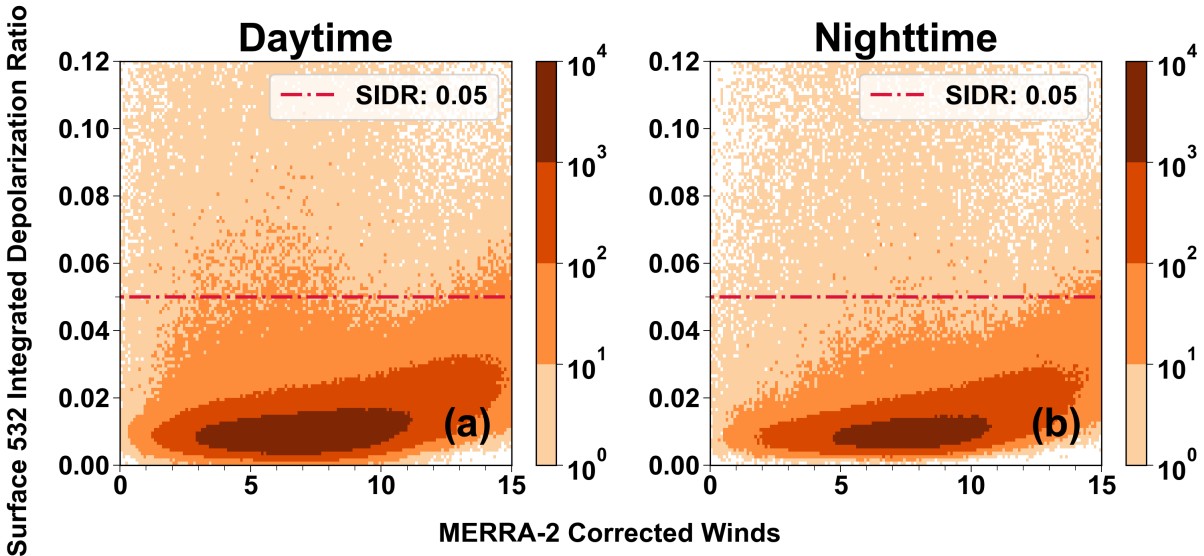

**Figure 8: SIDR as a function of wind speed for CALIOP determined aerosol-only 5 km ODCOD valid retrievals in daytime (left) and nighttime (right) with the SIDR filtering threshold of 0.05 marked with a dashed red line.**

Figure 8 shows a tight distribution of SIDR values around the median of approximately 0.01 for valid aerosol-only ODCOD retrievals. While there appears to be a slight wind speed dependance, we use this simple threshold to filter profiles with SIDR greater than 0.05 to remove the larger outliers from the distribution.

**3.1.2 Airborne HSRL Data Selection**

To assess how well ODCOD performs against airborne HSRL measurements, we compare ODCOD 5 km retrievals to AODs measured by Langley Research Center's High Spectral Resolution Lidar (HSRL-1) (Hair et al., 2008) and High Spectral Resolution Lidar Version 2 (HSRL-2) (Burton et al., 2018; Ferrare et al., 2023). The HSRL instruments provide quality atmospheric measurements from high altitude aircraft. HSRL-1 and its successor HSRL-2 have been operating in various

field campaigns since 2006. We compare ODCOD 5 km retrievals to measurements acquired by the HSRL-1 and HSRL-2





during campaigns in which CALIPSO overpasses occurred. The dates and HSRL field campaigns during which these CALIPSO underflights occurred can be found in Appendix B.

These scenes were selected from 152 CALIPSO underflights conducted by the Langley HSRL team. The selection process
was performed by considering the time and distance of each measurement to the coincident CALIPSO overpass 5 km footprint. The selection criteria require the HSRL measurement and associated ODCOD 5 km retrieval midpoint to have a time difference of less than 60 minutes and a spatial offset of less than 5 km. We choose to use 5 km ODCOD profiles because ODCOD averages the surface returns before retrieving and provides a more consistent retrieval over the scene. We also apply the ODCOD filters described in Sect. 3.1.1 for aerosol-only profiles. Once collocated and filtered, we require a
minimum of four matching ODCOD 5 km retrievals in the flight and require that no atmospheric features be reported in CALIOP's LL2 vertical feature mask above the HSRL data top altitude (approximately 8.5 km) of the matching points. Due to the locations of available underflights that satisfy these conditions, the scenes used occur exclusively in the North American and Venezuelan Basins of the Atlantic Ocean as shown in Fig. 9.

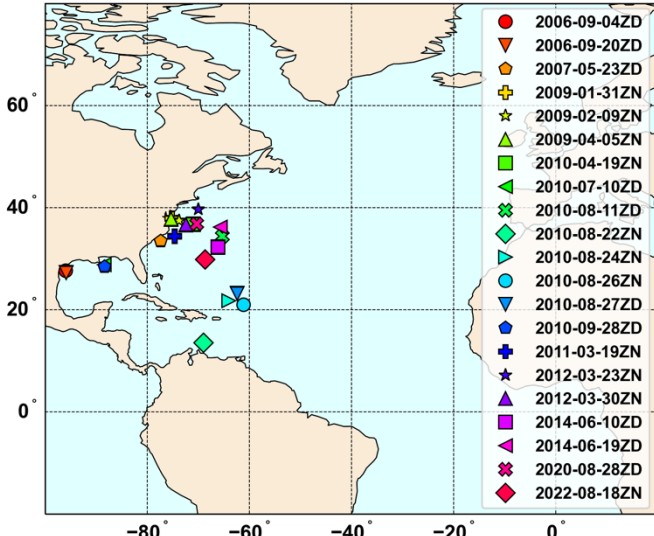

**Figure 9: Locations of the HSRL underflights used in the comparisons to ODCOD 5km.**

The HSRL processing calculates AOD for a given profile directly from the measured molecular channel and estimated molecular backscatter coefficients of the atmosphere computed from reanalysis model temperature and pressure data, (Hair et al., 2008). Rogers et al. (2009) compared HSRL profiles of aerosol extinction and AOD results to established measurements and found agreement within 0.01 km$^{-1}$ for extinction and 6% for AOD (532 nm). The HSRL aircrafts fly
significantly lower than CALIPSO, so when comparing AODs it is critical to consider light extinction above the altitude at which the HSRL measurements begin. The CALIOP signal will attenuate between the CALIOP and HSRL data top altitudes and must be accounted for when comparing HSRL and ODCOD AOD retrievals to estimate similar quantities. Even if no



layers are directly detected by CALIOP above the aircraft, there are still undetected background particulates (e.g., stratospheric aerosols), and we can expect the HSRL AOD measurements to be lower than the estimates from ODCOD.


Since only scenes with no atmospheric features detected above the HSRL aircraft are selected for this analysis, only undetected background particulates need to be account for. To do so, a background particulate optical depth is calculated for the given scene from the CALIOP LL2 data top altitude (30.1 km) down to the HSRL data top altitude which can vary during any given flight. To estimate the background particulate optical depth, the CALIOP LL1 532 nm attenuated

backscatter that spans the 80 km block or blocks of CALIOP LL2 data, aligned with the 80 km average of the LL2 data product, is selected as well as the 80 km on either side, excluding detected atmospheric features. This single vertical profile of background-only attenuated backscatter has the same vertical altitude structure as the CALIOP LL1 data and is an averaged minimum of 240 km horizontally. We applied the same a priori estimate of background aerosol extinction-to-backscatter ratio of 50 sr$^{-1}$ as the CALIOP level 3 stratospheric aerosol product (Kar et al., 2019), and performed a Fernald

retrieval of particulate optical depth (Fernald et al., 1972) from 30.1 km (the CALIOP Level 2 data top altitude) to the HSRL data top altitude. Each HSRL AOD measurement is adjusted by adding the estimate of background AOD above the aircraft before averaging to the closest ODCOD retrieval. Using a lidar ratio of 50 sr$^{-1}$ provides a median correction for the scenes of $0.018 \pm 0.005$ with a median HSRL data top altitude of $8.54 \pm 0.18$ km. In contrast, using a lidar ratio as low as 28.75 sr$^{-1}$ (Kim et al., 2017) would provide a median correction of $0.009 \pm 0.003$ which sets the lower bound on the error in this

correction estimate. Each HSRL measurement is matched to the closest in distance ODCOD 5 km retrieval and averaged to provide one HSRL comparison value and a standard deviation for each ODCOD 5 km estimate. The results of the comparisons are presented in Sect. 3.2.3.

### 3.1.3 MODIS Data Selection

To assess how well ODCOD performs on a global scale, we compare ODCOD 5 km retrievals in CALIOP determined

cloud-free profiles to collocated and interpolated MODIS aerosol optical depths reported in MODIS MYD04 (Levy et al., 2015). The collocations are determined by the collopak software suite provided by the University of Wisconsin (Nagel and Holz, 2009) and utilize the CALIPSO Version 4.51 LL1 product, V4.51 LL2 product, Collection 6.1 MODIS MYD03 1 km product, and Collection 6.1 MODIS MYD04 10 km product. The MODIS data selected is the MODIS Effective_Optical_Depth_Average_Ocean at wavelengths 470 and 550 and then interpolated in latitude and longitude to the

midpoint of each ODCOD 5 km CALIOP determined cloud-free profile and in wavelength to 532 nm. MODIS only uses cloud-free pixels for the optical depth estimates, so no further cloud screening is done for the MODIS data. MODIS data are chosen such that the MODIS confidence flag is greater than zero and quality considered *useful* by the MODIS quality flags. We also apply the ODCOD filters described in Sect. 3.1.1 for aerosol-only profiles. All data is compared one-to-one such





that both datasets require a valid retrieval to be used in this analysis. The results of the comparisons are presented in Sect.
580   3.2.4.

### 3.1.4 SODA Data Selection

To assess how well ODCOD performs compared to another established method of estimating optical depth from the ocean surface return, we compare ODCOD single shot (333 m) retrievals in CALIOP determined cloud-free profiles to the corresponding Synergized Optical Depth of Aerosols (SODA) retrievals. While similar, there are some distinct differences
between the algorithms. SODA CPR (Cloud Profiling Radar) uses the surface return from CALIOP, the surface return from CloudSat's CPR, and the wind speeds from AMSRE to make a multi-instrument estimate of the total column effective optical depth (Josset et al., 2008; Josset et al., 2012). The spatial and temporal collocations and calibrations of each of the SODA instruments adds uncertainties to the SODA retrieval that ODCOD does not have. SODA also implements additional calibration corrections by comparing the CPR and lidar surface returns as well as another calibration applied only at
nighttime (Josset et al., 2010a). SODA also integrates the attenuated backscatter signal over a fixed range about the ocean surface. The upper limit includes 2 bins above the peak and the lower limit is determined by the CALIOP LL2 range bin number 572. Typically, this makes the range from 0.053 km to -0.277 km. SODA applies a subsurface correction to estimate the lidar return from beneath the air-ocean interface instead of the fitting technique employed by ODCOD. These algorithm differences introduce different uncertainties in the SODA algorithm. SODA 333m Version 2.30 and 2.31 was used to
compare with ODCOD 333 m retrievals and we apply the ODCOD filters described in Sect. 3.1.1 for aerosol-only profiles. Since SODA is reported at the same footprint as CALIOP, no collocation is necessary. However, the SODA scene flags and quality assurance flags are used to retain only (a) valid scenes (b) located over ocean that (c) are wholly free of sea ice and for which the surface signals (d) are not close to the total attenuation threshold and (e) not saturated in either 532 nm channel, (f) CloudSat data is not missing, (g) AMSR data is not missing, and (h) AMSR sea surface temperature and liquid
water path are valid. All data is compared one-to-one such that both datasets require a valid retrieval to be used in the analysis. The results of the comparisons are presented in Sect. 3.2.5.

### 3.2 Results

This section reports results of analysis on comparing daytime and nighttime ODCOD 5 km retrievals, ODCOD at different resolutions, comparisons to the NASA LaRC airborne HSRL instrument, comparisons to MODIS and SODA, and finally a
summary of all the comparisons done in this section.

### 3.2.1 ODCOD

The near-global coverage provided by ODCOD both day and night is well suited for studying the regional distribution of aerosol optical depths. ODCOD has the potential to be used for studying cirrus clouds as well, but cirrus clouds are not the





focus of the paper. CALIOP's standard cirrus retrievals have already been well-validated with airborne lidar (Hlavka et al.,
2012) and MODIS (Holz et al., 2016) and can be confidently used for studying cirrus clouds. While ODCOD does provide
an estimate when transparent water clouds are present, a significant effort remains to understand the multiple scattering
effects of water clouds on ODCOD and the significant impact on the reported optical depths.

Figures 10 and 11 show seasonal aerosol-only ODCOD optical depths in 5 km profiles filtered as described in Sect. 3.1.1
where no clouds are detected by CALIOP. Elevated values are seen in regions where high aerosol loading is expected:
Saharan dust outflow is evident in the tropical Atlantic Ocean in June–August (e.g., Ridley et al., 2012), Asian dust is
evident over the northern Pacific Ocean in March-May (e.g., Liu et al., 2013), and smoke from biomass burning is
discernable off the southwest Africa coast in September-November (e.g., Yu et al., 2012) despite persistent marine
stratocumulus in the region. These regional patterns of enhanced optical depth are consistent with AOD observed by MODIS
(Remer et al., 2008).

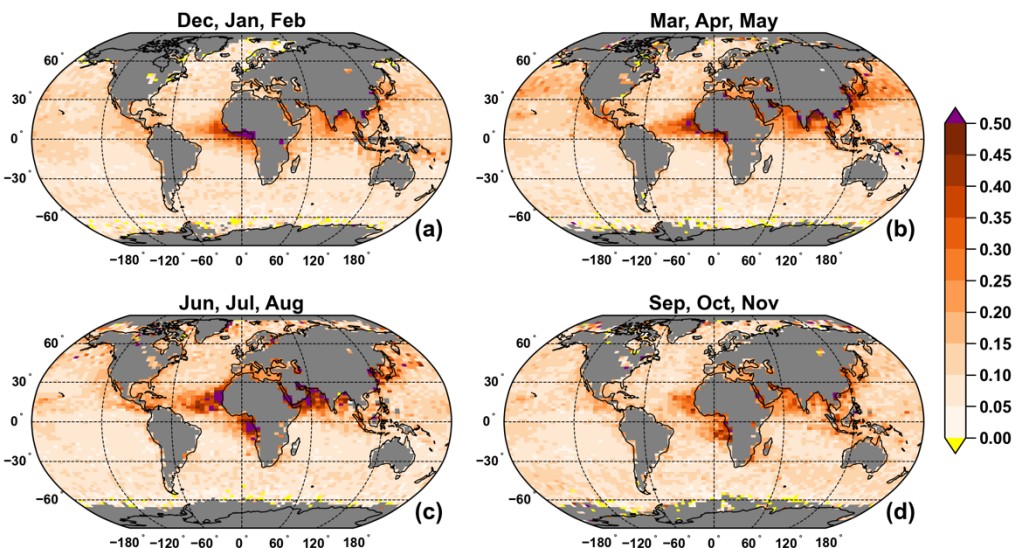

**Figure 10: Daytime seasonal median ODCOD aerosol optical depth, March 2008 through February 2011.**



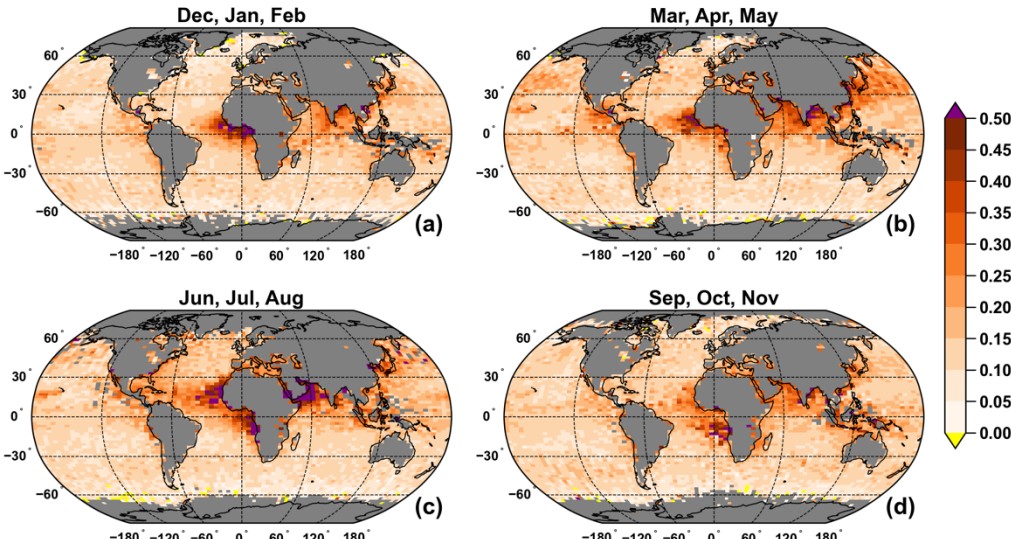

**Figure 11: Nighttime seasonal median ODCOD aerosol optical depth, March 2008 through February 2011.**

Globally, the average optical depth reported by ODCOD is higher at night compared to the day (Fig. 12). This is at least in part due to the sampling bias caused by the greater occurrence of surface saturation at night due to differences in CALIOP's daytime and nighttime detector gains (Hunt et al., 2009). Because ODCOD retrievals are not performed on profiles with saturated surface returns, which preferentially occur in low optical depth scenes, successful nighttime ODCOD retrievals yield higher average optical depths compared to those during the day. The average nighttime optical depth is higher because the lower optical depth columns are not represented: the median optical depth difference is on the order of $0.03 \pm 0.07$ higher at nighttime with a relative difference of around 22 % globally. Statistically, we can evaluate the differences between day and night using an unequal variance t-test which shows that the null hypothesis p-value $<< 0.05$ and the 95 % confidence interval for night minus day is approximately 0.026 to 0.027. Regionally some differences are much greater.

Solar background noise introduces another source of day-night bias. In columns with high particulate optical depths, detection of the surface returns used by ODCOD requires distinguishing a strongly attenuated surface peak from the ambient background noise. In cloud-free skies over oceans, CALIOP's daytime SNR is, conservatively, a factor of around 6 lower than at night, resulting in a much broader daytime background noise envelope in the region of tenuous surface returns. This large noise enhancement impedes the detection of attenuated surface peaks, and thus truncates the high end of the daytime ODCOD distribution at a substantially lower value than at night. However, the fraction of ODCOD aerosol-only profiles that have an optical depth greater than 1.0 at both day and night is less than 0.1 % and thus have little effect on the statistics here and are not considered.





While some of this day-night AOD difference could be due to true natural variation in daytime and nighttime aerosols or an unaccounted-for bias in the wind speed data day to night, a significant portion is due to this saturated surface sampling bias which is especially acute when averaging cloud-free columns. The absence of solar background noise during nighttime observations allows optically thin clouds to be detected much more often at night, which in turn will also cause a sampling bias when attempting to study aerosol-only profiles because fewer profiles in general will be considered aerosol-only at night

and more clouds will go undetected in the daytime (Liu et al., 2019).

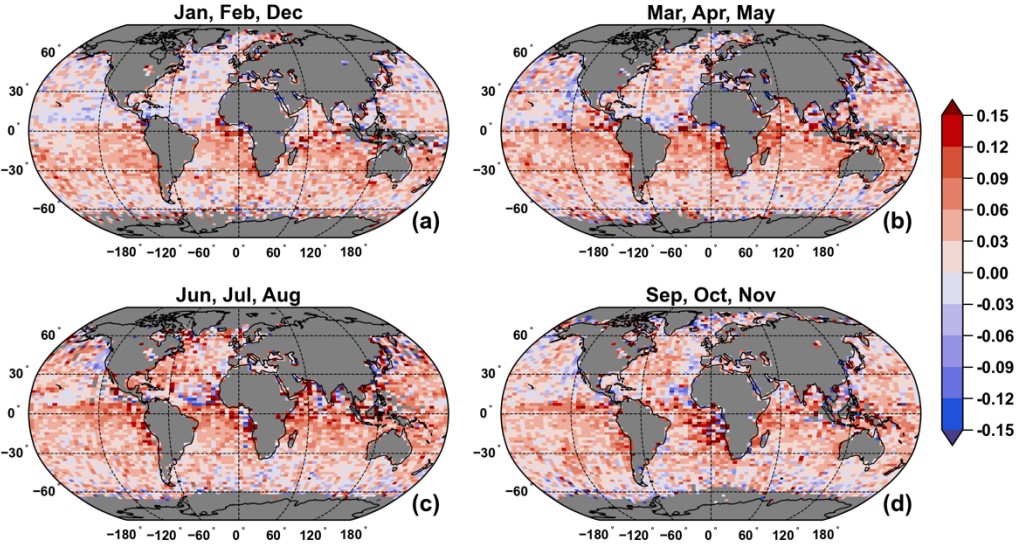

**Figure 12: Nighttime median minus daytime median seasonal ODCOD aerosol optical depth differences, March 2008 through February 2011.**

Figure 12 shows regional 5 km ODCOD nighttime median minus daytime median aerosol-only optical depth differences. The data for Fig. 12 has been filtered as described in Sect. 3.1.1 with separate SIAB thresholds used for day and night.



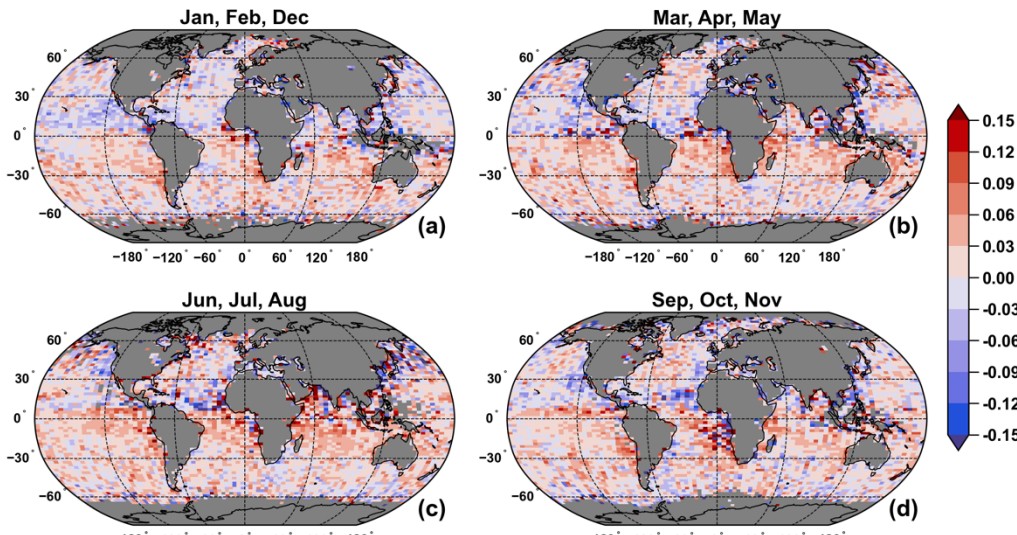

**Figure 13: Nighttime median minus daytime median seasonal ODCOD aerosol optical depth differences with daytime SIAB filter applied to both day and night, March 2008 through February 2011.**

To demonstrate the impact of the surface-saturation sampling bias, we experimentally modified the surface saturation filter described in Sect. 3.1.1 to use the nighttime SIAB threshold for both day and night observations. It is critical to understand that by applying the same SIAB threshold filter, we are removing some good-quality low daytime data that will no longer be represented in the average. The median night minus day differences in Fig. 13 shows that the daytime median optical depth drops to $0.010 \pm 0.006$, or about 7% relative difference. Statistically, we still find a slight difference between night and day

with the unequal variance t-test which showing that the null hypothesis p-value $\ll 0.05$ but the 95 % confidence interval for drops to approximately 0.004 to 0.005. This experiment confirms that the sampling bias from surface-saturation differences day and night is a significant contributor to the day to night differences in ODCOD retrievals.

### 3.2.2 ODCOD at Different Spatial Averaging Resolutions

   ODCOD is reported in the CALIOP LL2 data products at single shot (333 m), 1 km, and 5 km resolutions. For the 1 km and

5 km data products, the ODCOD algorithm uses the averaged surface return detected at each resolution rather than averaging finer resolution ODCOD retrievals. By averaging the surface return before retrieval, the ODCOD Impulse Response Model (IRM) fitting procedure has a better chance of correctly fitting the IRM to the surface points. Even a relatively small noise perturbation to one of the two surface return points used to fit the IRM can cause the time delay to be incorrectly estimated which typically results in a surface return area that is too small and thus an optical depth that is too large.


   This average-then-retrieve approach is important because averaging finer resolution retrievals would bias the coarser resolution low anytime the surface was not detected in all finer resolution profiles. For example, if a retrieve-then-average



approach were used to estimate ODCOD at 5km and only some of the 15 single shots within the 5km average detected the ocean surface, only those that did detect the surface would have valid ODCOD retrievals to average. Only averaging the

surface-detected ODCOD retrievals would mean not accounting for the very high optical depths of the missing retrievals, and the estimate would be bias significantly low. More missing single shot estimates also means a more significant bias. By averaging the surface return backscatter data for all 15 shots regardless of if they detected a surface or not, ODCOD is better representing the ocean return magnitude for the 5km profile and the optical depth CALIOP is able to detect. However, we must recognize that an estimate of the optical depth with totally attenuated single shot columns within averaged profiles will

still not represent the overall optical depth of the 5km column as the lidar does not fully penetrate the atmosphere to the ocean surface in those regions.

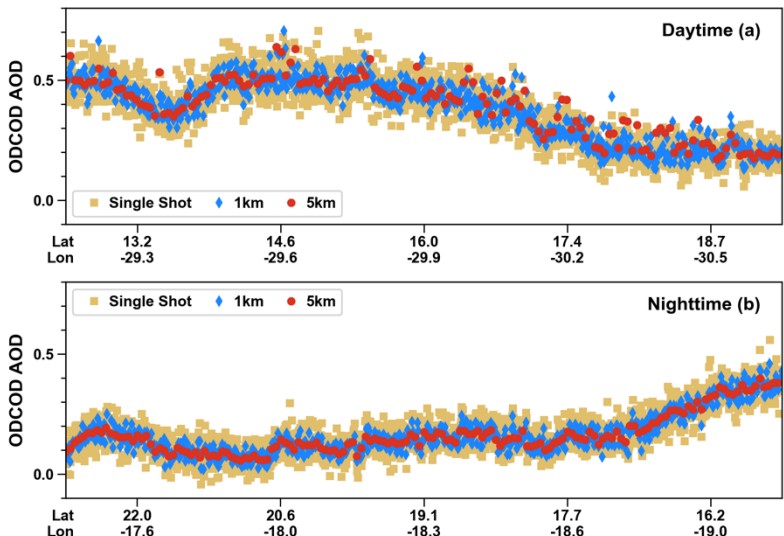

**Figure 14: ODCOD retrievals of aerosol-only profiles as determined by CALIOP at single shot (333 m) (yellow squares), 1 km (blue**
**diamonds), and 5 km (red circles) resolutions for a daytime and nighttime scene in panels (a) and (b) respectively. The daytime scene is over the international atomic times 2008-05-05T15:33:30 to 2008-05-05T15:35:24 and the nighttime scene is 2008-05-28T03:08:19 to 2008-05-28T03:10:18.**

Figure 14 shows a daytime and nighttime scene selected for having a large number of consecutive aerosol-only ODCOD retrievals with filtering according Sect 3.1.1. In general, the retrievals show that the 5 km retrievals are less noisy and fall on

top of the 1 km retrievals which are again less noisy and fall on top of the single shot retrievals. In the daytime, the effects of the solar background radiation can be seen from the larger spread of the data at each resolution as well as occasional deviations of the resolutions from lying directly on top of one another. However, nighttime is very well behaved and shows how well the ODCOD average and retrieve technique does when not having to contend with solar background noise.





Typically using the coarsest resolution ODCOD data is the right choice for analysis however, with proper selection and
taking care to not bias averages, averaging finer resolution retrievals can also provide reasonable estimates.

### 3.2.3 Comparisons to airborne HSRL

In general, ODCOD 5 km compares well with HSRL aerosol optical depth retrievals and shows little to no bias when day
and night are considered together with the median difference being $0.009 \pm 0.043$ (6 % ± 28% relative difference; N=395)

with ODCOD higher. A paired sample t-test shows the difference is not statistically significant with a p-value of 0.36 and a
95 % confidence interval for the mean difference of -0.005–0.014. Separately, ODCOD compares slightly lower in the
daytime and slightly higher at night but with uncertainties larger than the difference in either. The median difference in the
daytime is $-0.037 \pm 0.052$ (-12 % ± 25%; N=149) with ODCOD lower, the p-value shows the difference is statistically
significant, and the 95 % confidence interval for the mean difference is -0.044 to -0.019. The median difference at night is

$0.021 \pm 0.032$ (14 % ± 25%; N=246) with ODCOD higher, the p-value also shows the difference is statistically significant,
and the 95 % confidence interval for the mean difference is 0.014 to 0.039.

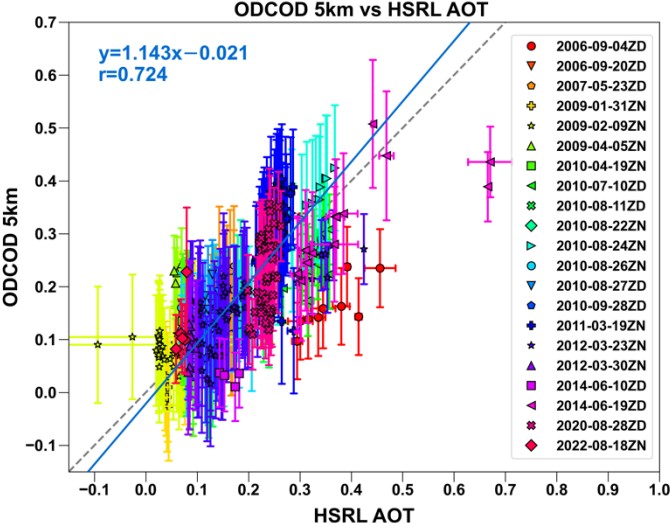

**Figure 15: HSRL AOD compared to 5 km ODCOD retrievals for 21 select HSRL underflights of CALIPSO. The dashed grey line is**
**the one-to-one line and the solid blue line is the orthogonal best fit line with fit parameters shown in blue in the upper left corner.**

Figure 15 shows the collocated and quality filtered 5 km ODCOD retrievals as a function of above-aircraft adjusted HSRL
measurements from the 21 CALIPSO underflights that met the selection criteria outlined in Sect. 3.1.2. The HSRL values are
mean values computed over the ODCOD 5 km averaging interval, with the standard deviation of the HSRL measurements
shown as the error bars. The error bars of the ODCOD retrievals show the reported ODCOD standard deviation estimates

from the LL2 data products. Spatial and temporal colocation differences will introduce some uncertainties into these one-to-





one comparisons. Nevertheless, Fig. 15 shows that when considered over multiple flights and at both day and night, that while there are statistically significant differences, there is agreement with the HSRL instrument measurements even over a variety of optical thickness scenes.

### 3.2.4 Comparisons to MODIS

In general, the global median difference between ODCOD 5 km daytime retrievals and MODIS interpolated 532 nm AOD is -0.009 ± 0.041 (8% ± 35%; N=1,999,068) with ODCOD lower. Applying a paired sample t-test shows the differences are statistically significant but the 95 % confidence interval for the mean difference is -0.0069 to -0.0067. Regionally, ODCOD tends to report higher aerosol optical depths in the southern oceans from March through August and seems to swing to lower optical depths in December through February. ODCOD also tends to report higher aerosol optical depths north of 30 ° N
from September through February but the difference is less during March through August.

Table 3 and Fig. 16 show how ODCOD 5 km daytime estimates compare to MODIS retrievals for data acquired by both instruments from March 2008 through February 2011.

| | Mean ± Std. Dev. | | Median ± M.A.D. | | Mean of Differences (ODCOD - MODIS) | Median of Differences (ODCOD - MODIS) | Number of Samples |
|---|---|---|---|---|---|---|---|
| | ODCOD | MODIS | ODCOD | MODIS | | | |
| **DJF** | 0.121 ± 0.103 | 0.129 ± 0.092 | 0.103 ± 0.048 | 0.108 ± 0.036 | -0.007 ± 0.066 | -0.010 ± 0.040 | 593,824 |
| **MAM** | 0.135 ± 0.124 | 0.143 ± 0.122 | 0.109 ± 0.053 | 0.112 ± 0.044 | -0.008 ± 0.069 | -0.010 ± 0.042 | 503,438 |
| **JJA** | 0.115 ± 0.114 | 0.121 ± 0.113 | 0.094 ± 0.046 | 0.095 ± 0.042 | -0.007 ± 0.069 | -0.009 ± 0.044 | 385,034 |
| **SON** | 0.128 ± 0.107 | 0.133 ± 0.102 | 0.108 ± 0.048 | 0.108 ± 0.037 | -0.005 ± 0.066 | -0.008 ± 0.040 | 516,772 |
| **Yearly** | 0.125 ± 0.112 | 0.132 ± 0.107 | 0.104 ± 0.049 | 0.107 ± 0.039 | -0.007 ± 0.067 | -0.009± 0.041 | 1,999,068 |

**Table 2: Statistics summarizing the comparison results. Each row summarizes the seasonal sets of months designated by the first letter of the months in question such as December, January, and February as DJF with the final row labelled Yearly as the total statistics for all data March 2008 through February 2011.**





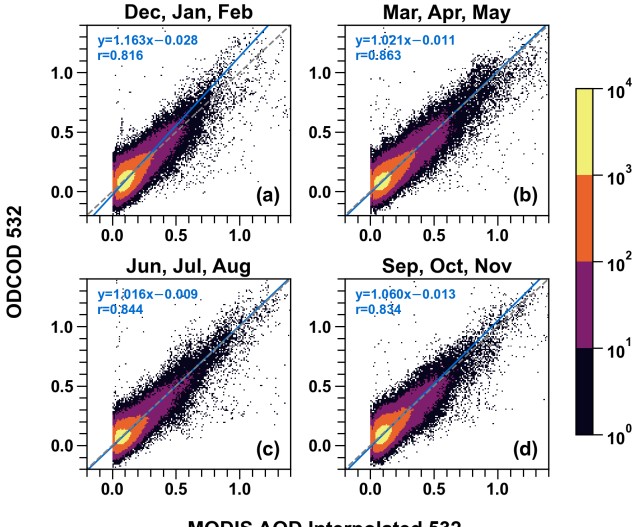

**740** **Figure 16: Collocated ODCOD aerosol optical depth at 5 km resolution on the y-axis and MODIS effective optical depth average ocean collocated and interpolated in latitude, longitude, and wavelength the midpoint of the CALIOP 5 km profile and 532nm wavelength, March 2008 through February 2011.**

Unlike ODCOD, which permits negative optical depths, the MODIS algorithm reports an optical depth of zero when the observed top of the atmosphere is greater than or equivalent to the Rayleigh plus surface signal. While negative optical

**745** depths are non-physical, they arise due to random noise in the original measurements and thus should be retained when computing statistics to avoid introducing high biases when estimating means, medians, correlations, and lines of best fit. Removing data pairs where MODIS reports zero only slightly improves correlation with differences of only a few thousandths and slightly increases the slope with differences of only a few hundredths and does not significantly change the resulting statistics. Globally, the median difference with zeros excluded is still statistically significant at -0.010 ± 0.041

**750** (8 % ± 35% relative difference; N=1,979,372) again with ODCOD slightly lower and the 95 % confidence interval for the mean difference still -0.0069 to -0.0067. Simply removing data pairs where MODIS reports zero may bias the comparison in other ways and since the zeros removed comparison shows little to no significant difference, we consider the comparisons presented reasonable.





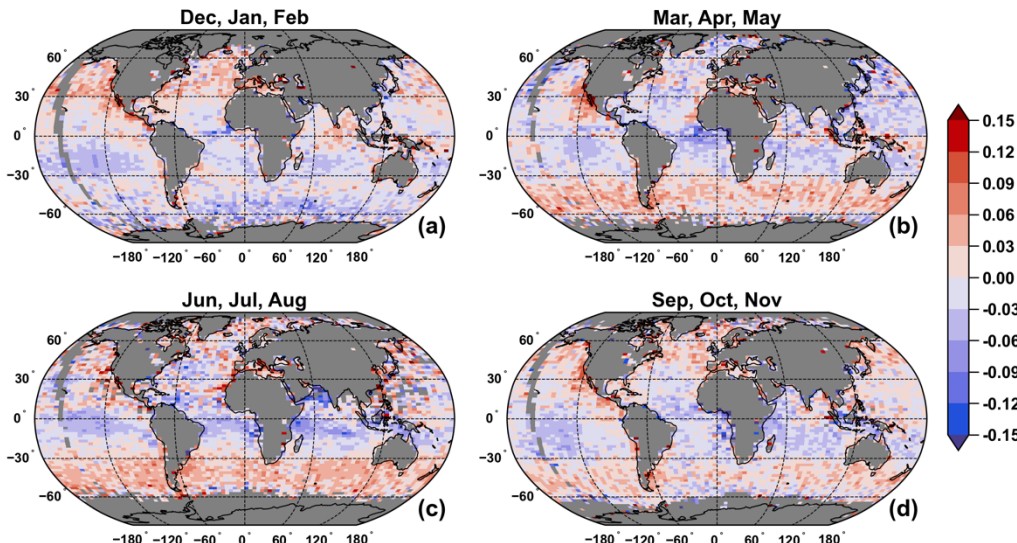


**Figure 17: ODCOD minus MODIS aerosol optical depth difference of the median values, ODCOD higher shown in red and ODCOD lower shown in blue, March 2008 through February 2011.**

The maps in Fig. 17 provide seasonal depictions of the regional differences between daytime AOD retrieved by MODIS and 5 km ODCOD. The data shown in these maps are the same data shown in Fig. 16 with the difference of the medians of each

bin presented regionally. Figure 16(b) and Fig. 16(c) show notable differences, with ODCOD higher on the order of 0.03 to 0.06 in the Southern Ocean from March through August. These differences largely disappear from September through February. It is also important to consider data filtering in these results. While cloudy pixels are screened by the MODIS algorithm and ODCOD has been screened for clouds along the 5 km track, regions where thin cirrus clouds are frequently found have a higher chance of undetected clouds in the ODCOD retrieval and cloud contamination in the MODIS retrieval.

Cloud contamination could be a source of bias in this comparison as it is known that cloud contamination can bias MODIS high (Spencer et al., 2019; Reid et al., 2022). Also, regions where clouds are detected more frequently will be sampled less often. Additionally, the larger 10 km MODIS pixel compared to the ODCOD 5 km swath will introduce artifacts into the comparison due to the difference in sampling.

### 3.2.5 Comparisons to SODA

In general, ODCOD 333 m retrievals compare well with SODA 333 m daytime aerosol optical depth retrievals and show a relatively small difference globally. The daytime median difference is 0.004 ± 0.035 (1 % ± 34% relative difference; N=21,270,202), ODCOD higher. Applying a paired sample t-test shows the differences are statistically significant but the 95 % confidence interval for the mean difference is 0.0044 to 0.0045. In the nighttime, ODCOD 333 m reports higher values than SODA, with a nighttime median difference of 0.027 ± 0.034 (20 % ± 33% relative difference; N=10,536,357), ODCOD

higher. The paired sample t-test shows the differences are statistically significant with the 95 % confidence interval for the



mean difference being about 0.028. SODA has statistically similar values both day and night with median values of $0.102 \pm$ 0.045 daytime and $0.105 \pm 0.045$ nighttime. Statistically higher optical depths at night due to better surface detection is expected in both datasets as SODA and ODCOD are filtered one-to-one, ODCOD requires a valid surface detection, both are the same footprint, and both use CALIOP data as an input. We would expect the nighttime retrievals to be on the order of
0.02 higher at night as demonstrated in Sect. 3.1.1. Even though the CALIOP data used by SODA version 2.3 and 2.31 is CALIOP LL1 version 4.1, the surface saturation does not change between versions although, calibration changes were made to the CALIOP V4.51 data products polarization gain ratio which improved the CALIOP calibration between version 4.1 and V4.51 (Getzewich et al., 2024). Other than the described nighttime differences, we find the comparisons between ODCOD and SODA to be good and generally acceptable.


During analysis, it was found that SODA has occasional anomalous data that is not filtered by the SODA scene or quality assurance flags. This artifact becomes apparent when plotting ODCOD as a function of SODA, as the anomalous points form striated lines in what appear to be somewhat quantized groupings, many of which are relatively large negative values. The anomalous points can be clearly seen in Fig. 18 and Fig. 19. Preliminary investigations indicate that one primary cause of
these SODA outliers is the inadvertent use by the SODA algorithm of CPR data acquired during CPR calibration maneuvers (Tanelli et al., 2008).

To separate the anomalous data from quality data, the SODA data is binned by the matching ODCOD data in 0.01 optical depth bins and Tukey fences are calculated for each bin. Carling (2000) defines "Tukey's rule" for identifying outliers in a
data set as $c_{low} = q_1 - k_1(q_3 - q_1)$ and $c_{high} = q_3 + k_1(q_3 + q_1)$, where, "$q_1$ and $q_3$ are the sample quartiles [...] and $k_1$ is a constant selected to meet a pre-specified outside rate under some model" and $c_{low}$ and $c_{high}$ are, respectively, the cut-off points beyond which points in the lower and upper tails of the distribution are deemed to be outliers. In our case the pre-specified constant is set to $k_1 = 4.5$ based on visual inspection of the joint distribution (e.g., Figs. 18 and 19). This value retains approximately 99.8 % to 99.9 % of data and clearly labels the anomalous distributions.


Table 3 and Figs. 18 and 19 show how ODCOD at single shot resolution compares to SODA CPR during the day and night for all CALIOP data acquired from March 2008 through February 2011 that fall within the described outlier envelope.

| | Mean ± Std. Dev. | | Median ± M.A.D. | | Mean of Differences (ODCOD - SODA CPR) | Median of Differences (ODCOD - SODA CPR) | Number of Samples |
|---|---|---|---|---|---|---|---|
| | ODCOD | SODA CPR | ODCOD | SODA CPR | | | |
| **DJF (day)** | $0.117 \pm 0.101$ | $0.116 \pm 0.088$ | $0.103 \pm 0.054$ | $0.103 \pm 0.043$ | $0.001 \pm 0.056$ | $0.001 \pm 0.036$ | 5,787,035 |



| | | | | | | | |
|---|---|---|---|---|---|---|---|
| **DJF (night)** | $0.142 \pm 0.105$ | $0.121 \pm 0.095$ | $0.128 \pm 0.052$ | $0.109 \pm 0.044$ | $0.021 \pm 0.055$ | $0.020 \pm 0.034$ | 2,417,916 |
| **MAM (day)** | $0.129 \pm 0.116$ | $0.123 \pm 0.099$ | $0.109 \pm 0.057$ | $0.105 \pm 0.045$ | $0.005 \pm 0.056$ | $0.004 \pm 0.036$ | 5,008,904 |
| **MAM (night)** | $0.163 \pm 0.126$ | $0.132 \pm 0.109$ | $0.140 \pm 0.058$ | $0.110 \pm 0.046$ | $0.031 \pm 0.055$ | $0.030 \pm 0.034$ | 2,444,714 |
| **JJA (day)** | $0.127 \pm 0.128$ | $0.127 \pm 0.114$ | $0.103 \pm 0.057$ | $0.104 \pm 0.048$ | $-0.000 \pm 0.055$ | $-0.001 \pm 0.034$ | 5,408,476 |
| **JJA (night)** | $0.151 \pm 0.105$ | $0.125 \pm 0.112$ | $0.131 \pm 0.055$ | $0.104 \pm 0.044$ | $0.026 \pm 0.054$ | $0.026 \pm 0.033$ | 3,068,930 |
| **SON (day)** | $0.123 \pm 0.105$ | $0.111 \pm 0.094$ | $0.108 \pm 0.054$ | $0.096 \pm 0.045$ | $0.012 \pm 0.053$ | $0.012 \pm 0.034$ | 5,065,787 |
| **SON (night)** | $0.147 \pm 0.104$ | $0.114 \pm 0.093$ | $0.134 \pm 0.053$ | $0.100 \pm 0.043$ | $0.034 \pm 0.056$ | $0.033 \pm 0.035$ | 2,619,174 |

**Table 3: Statistics summarizing the data used in the fits for Figs. 18 and 19. Each row summarizes one of the seasonal sets of months designated by the first letter of the months in question such as December, January, and February as DJF and either daytime or nighttime data.**

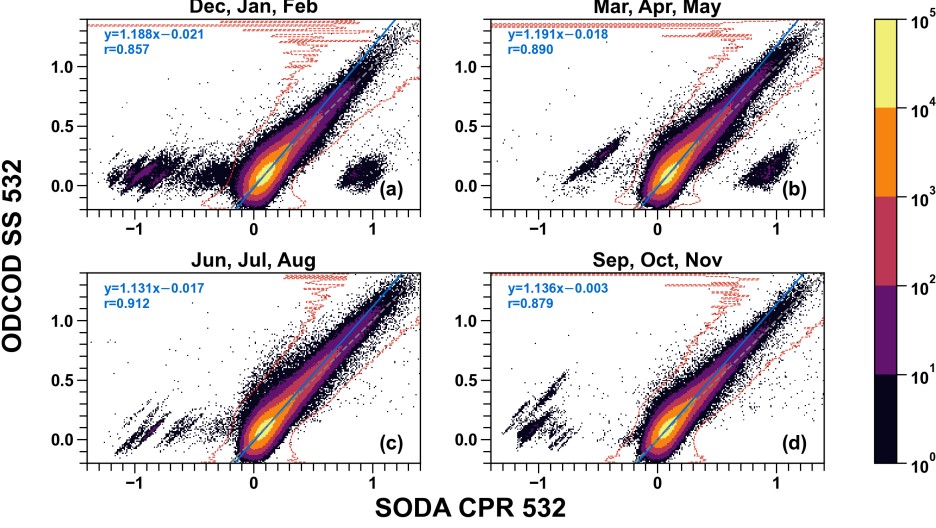

**Figure 18: Daytime ODCOD aerosol optical depth at single shot resolution as a function of SODA CPR effective optical depth with orthogonal fit line in solid blue and one-to-one line in dashed grey for March 2008 through February 2011. The red dashed line shows the extreme outer fence envelope used to filter anomalous data as described in this section. The fit parameters and lines shown on the plots are only for the data found inside the envelopes.**



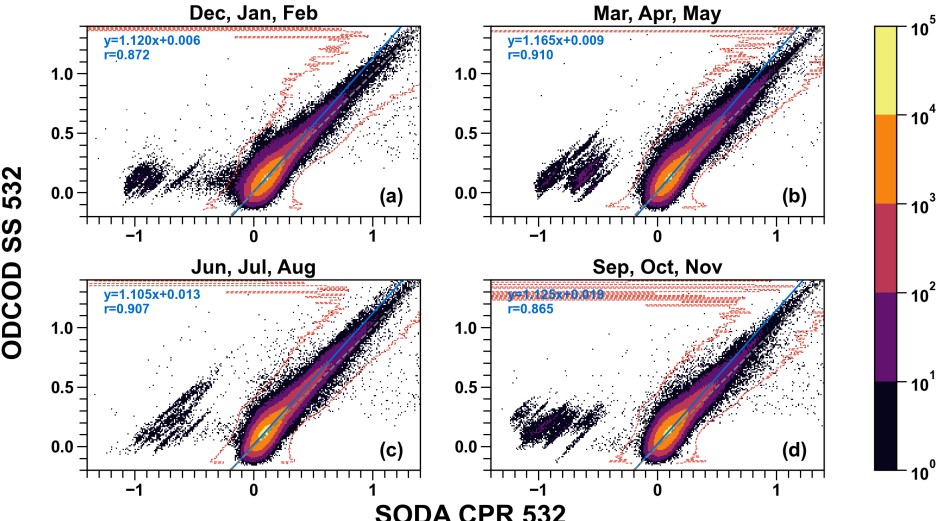

**Figure 19: Depicts the same information as Fig. 18, but for nighttime observations.**

Due to the CloudSat battery anomaly that occurred in April of 2011, CloudSat nighttime data and subsequently SODA CPR
retrievals at night are not available after that time. The ODCOD algorithm itself makes no distinction between day and night
and we are not currently aware of significant differences in the inputs to the algorithm at day versus night. CALIOP's
version 4.1 calibration has been extensively validated during both nighttime and daytime operations (Kar et al., 2018;
Getzewich et al., 2018) and the changes to the V4.51 calibration has shown improvements over version 4.1 (Getzewich et al.,

2024). We note that SODA incorporates its own calibration adjustments to the input data and applies a second nighttime only
calibration (Josset et al., 2010). The reason for the differences at night compared to SODA CPR are not well enough
understood to comment further.

### 3.2.6 Results Summary

The results of all comparisons performed are summarized in Table 4, which shows that even among established datasets the

agreement can vary but ODCOD agrees well on most accounts, especially in the daytime. At nighttime, ODCOD tends to
trend slightly higher than the datasets examined but the lack of well validated and confident nighttime data makes
comparisons to ODCOD in the nighttime more difficult.



| Measurement | Median ± MAD | | Median Difference ± MAD (ODCOD-Measurement) | | Relative Difference (ODCOD-Measurement)/ Measurement | |
|---|---|---|---|---|---|---|
| | **Day** | **Night** | **Day** | **Night** | **Day** | **Night** |
| HSRL | | | | | | |
| ODCOD 5 km | 0.183 ± 0.066 | 0.149 ± 0.0722 | -0.037 ± 0.052 | 0.021 ± 0.032 | -13 % ± 25 % | 14 % ± 25 % |
| HSRL | 0.228 ± 0.084 | 0.133 ± 0.067 | | | | |
| MODIS | | | | | | |
| ODCOD 5 km | 0.104 ± 0.049 | | -0.010 ± 0.041 | | -8 % ±35 % | |
| MODIS | 0.107 ± 0.039 | | | | | |
| SODA | | | | | | |
| ODCOD 333 m | 0.106 ± 0.055 | 0.133 ± 0.055 | 0.004 ± 0.035 | 0.027 ± 0.034 | 1 % ± 34 % | 20 % ± 33 % |
| SODA 333 m | 0.102 ± 0.045 | 0.105 ± 0.045 | | | | |

**Table 4: Summary results of day and night for all comparisons performed in this paper showing median and median absolute deviation (MAD).**

## 830 4 Future Work with CALIOP

Comparing ODCOD optical depth retrievals to the CALIOP standard retrievals present unique opportunities for analysis and improvement to the CALIOP standard algorithms. Since the only CALIOP LL2 algorithm ODCOD relies on is the surface detection algorithm (Vaughan et al., 2017) and the surface detection algorithm only requires the LL1 attenuated backscatter signal about the surface, ODCOD's optical depth estimates are independent from CALIOP's standard V4.51 LL2 reported
optical depths.

One of CALIOP's primary goals is to retrieve vertically resolved profiles of particulate extinction coefficients which can then be integrated to calculate an estimated column optical depth. To retrieve extinction coefficients, CALIOP uses a feature detection algorithm to first isolate regions of the vertical profile with elevated attenuated backscatter (Vaughan et al., 2009),
then prescribes an extinction-to-backscatter ratio (i.e., lidar ratio) for various aerosol types based on the CALIOP aerosol classification (Omar et al., 2009) and cloud/aerosol discrimination algorithms (Liu et al., 2019; Avery et al., 2020; Kim et al., 2018; Young et al., 2018). However, CALIOP only reports extinction coefficients for regions of the vertical profile where the attenuated backscatter signal is above the detection threshold (McGill et al., 2007). Regions of faint scattering from diffuse particulates will fall below these limits and hence go undetected. This inherently means that a small fraction of
the overall particulate extinction will not be included in CALIOP's column optical depth estimates. Kim et al. (2017) estimate CALIOP's undetected optical depth to be on the order of 0.030 ± 0.046. Based on comparisons to MODIS, Toth et al. (2018) report similar estimates of 0.03 to 0.05 for daytime retrievals. Consequently, estimates of global mean aerosol



direct radiative effect derived from CALIOP's standard AOD retrievals are biased low by ~54 % (Thorsen et al., 2017). One of the drivers of developing the ODCOD algorithm was to provide an internally consistent assessment of fraction of the
column optical depth that lies below CALIOP's direct detection thresholds. A comprehensive assessment of the CALIOP standard algorithms based on ODCOD retrievals lies well outside the scope of this paper. However, we provide some general observations to demonstrate the types of questions that an ODCOD versus CALIOP standard analysis could potentially answer.

We compare ODCOD 5 km retrievals to column optical depths calculated from the CALIOP standard stratospheric and tropospheric range resolved extinction coefficient retrievals reported in the CALIOP LL2 profile products. Comparisons with CALIOP standard products are done where CALIOP extinction QC flags report 0 (unconstrained retrieval), 1 (constrained retrieval), or 2 (initial lidar ratio reduced) to ensure a quality extinction profile, as well as the ODCOD filters described in Sect. 3.1.1 for aerosol-only profiles. While all profiles were screened to ensure no clouds were detected in the
5 km profile, no attempt was made to further filter the CALIOP data based on overlying clouds in adjacent profiles which might increase errors in the extinction retrievals of any 20 km or 80 km horizontally averaged layer extinction or cloud aerosol discrimination. All data is compared one-to-one such that both datasets require a valid retrieval to be used in this analysis.

In general, the ODCOD 5 km retrievals are expectedly higher than the CALIOP standard algorithm for retrieving aerosol optical depths. As suggested by Kim et al. (2017) the expected bias should be on the order of $0.030 \pm 0.046$. Figures 20 and 21 compare CALIOP standard aerosol-only optical depths to 5 km ODCOD retrievals in the same profiles for day and night respectively. The daytime median difference is $0.025 \pm 0.047$ (34 % ± 67 % relative difference; N=2,209,551) with ODCOD higher. Applying a paired sample t-test shows the differences are statistically significant and the 95 % confidence interval for
the mean difference is near 0.024. The nighttime median difference is $0.046 \pm 0.047$ (60 % ± 71 % relative difference; N= 1,037,873) again, with ODCOD higher. The t-test shows these differences are also statistically significant and the 95 % confidence interval for the mean difference is near 0.041.



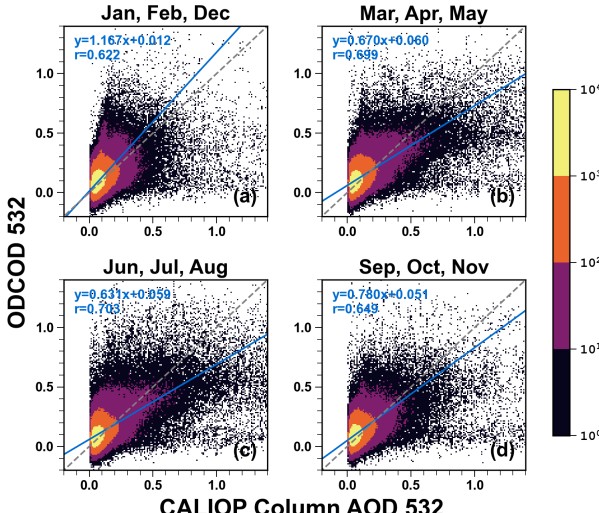

Figure 20: Daytime ODCOD aerosol optical depth at 5 km resolution as a function of CALIOP column optical depth for March 2008 through February 2011.

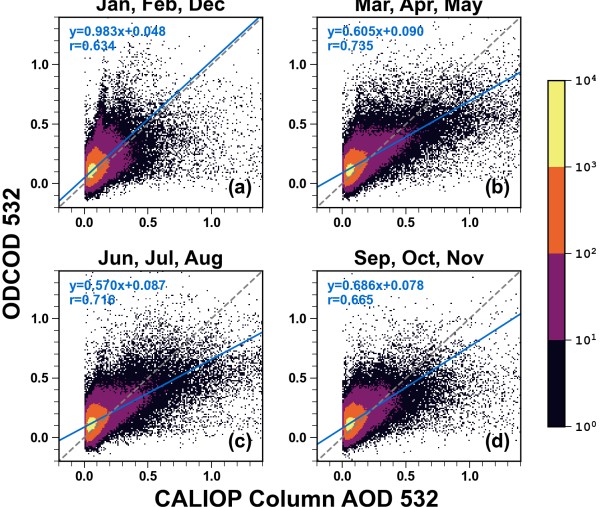

Figure 21: Nighttime ODCOD aerosol optical depth at 5 km resolution as a function of CALIOP column optical depth for March 2008 through February 2011.

This nighttime difference is larger than the day but attempting to estimate the global difference is difficult which is made apparent in Figs. 22 and 23. Some regions show strong positive and negative differences that vary significantly with season. Also, CALIOP's ability to detect clouds and aerosols is different at day than it is at night which will change the distribution of samples considered as aerosol-only profiles.



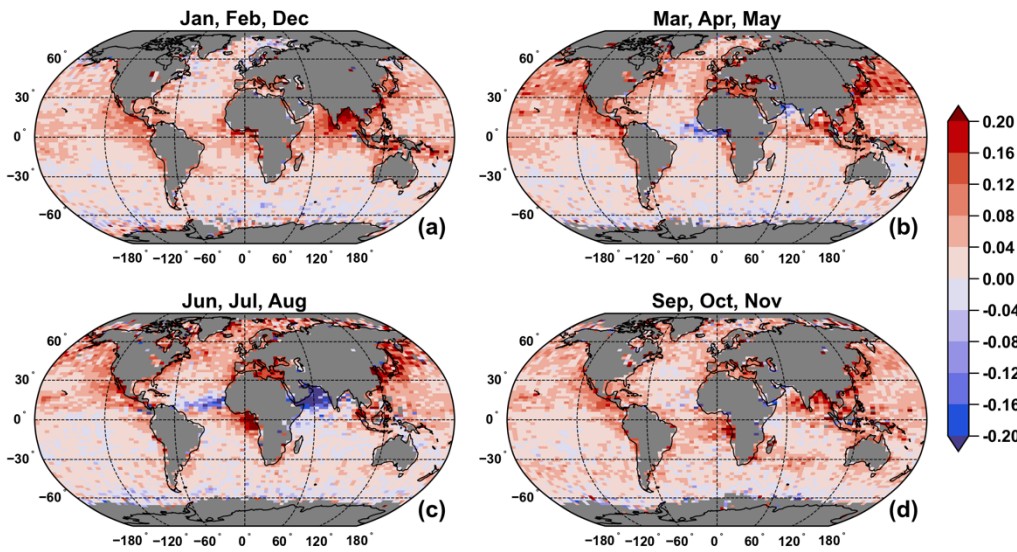


**Figure 22: Daytime aerosol optical depth difference (ODCOD - CALIOP) of the median values, ODCOD higher shown in red and ODCOD lower shown in blue for March 2008 through February 2011.**

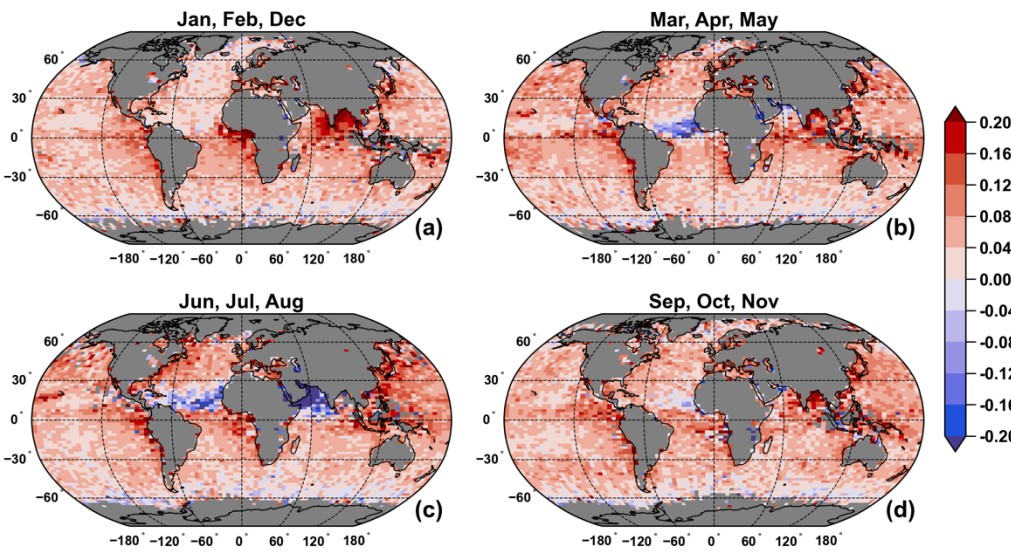

**Figure 23: Nighttime aerosol optical depth difference (ODCOD - CALIOP) of the median values, ODCOD higher shown in red and ODCOD lower shown in blue for March 2008 through February 2011.**

Figures 22 and 23 show how 5 km ODCOD compares to CALIOP 5 km column optical depths in the day and night, respectively. The most obvious difference is that, in general, larger values are retrieved by ODCOD in the northern hemisphere in the daytime and more globally at night. We expect ODCOD to be higher in general because it includes



extinction contributions from aerosols that are not detected by CALIOP's feature detection algorithm. At the same time, we
expect more undetected aerosols to be missed by CALIOP in the daytime due to noise from solar background radiation.
More study is needed to understand what drives the larger differences observed at night.

Some regional differences include the apparent stronger outflow of aerosols from Asia into the northern Pacific in March
through August according to ODCOD. This could occur for a variety of reasons. Aerosol layers that are geometrically thick,
yet optically thin might be missed by CALIOP feature detection or the applied lidar ratio may be too low. The ODCOD
retrieval might be too high due to a regional bias in the modelled wind speeds or due to unaccounted differences in ocean
reflectivity from expected model values.

In the Arabian Sea and Bay of Bengal region, ODCOD optical depths are much smaller than those of the CALIOP standard
retrieval in June – August. Mixtures of dust and marine aerosol are common in this region due to the outflow of dust from
the Arabian Peninsula (Satheesh et al., 2006). The CALIPSO standard algorithm typically classifies these layers as 'dusty
marine' and assigns a lidar ratio of 37 sr, which might be too high on average. Likewise, ODCOD optical depths are lower in
the Saharan dust outflow region in the tropical Atlantic. This could be indicative of high-biased lidar ratios for pure dust or
marine-dust mixtures in this region. However, we are not drawing conclusions here and a rigorous analysis is needed to
evaluate these questions.

**5 Conclusions**

CALIPSO's Version 4.51 Lidar Level 2 data products report a new estimate of total column effective optical depth derived
from the ocean surface lidar backscatter return by the Ocean Derived Column Optical Depths (ODCOD) algorithm.
Multiyear comparisons show that ODCOD daytime estimates of aerosol optical depth (AOD) have statistically significant
differences to several established and validated datasets, but those differences are generally small on the order of several
thousandths. Relative to daytime retrievals, ODCOD nighttime AOD estimates tend to be higher; however, in-depth global
comparisons are hindered by the lack of well understood and validated nighttime data derived from other sensors. Different
CALIOP amplifier gains during the daytime and nighttime portion of the orbit cause the lidar surface return to saturation
more frequently at night and the lack of solar background radiation also allows the surface to be detected more readily when
the surface return is very small. Since ODCOD requires unsaturated surface detections, both effects will cause a statistical
sampling bias of nighttime higher than daytime. However, the sampling bias does not account for differences between
datasets when compared on a profile-by-profile bases. ODCOD's 5 km optical depth has a median uncertainty of $0.12 \pm 0.05$
($76\% \pm 40\%$ relative) that increases as optical depth becomes small. The most significant source of uncertainty in the
ODCOD retrieval is wind speed. Through an AMSR-derived wind speed correction applied to the MERRA-2 winds, the
ODCOD algorithm attempts to reduce wind speed biases in the retrieval. Further sources of uncertainty are the surface return



model, the surface return model parameters, the surface retro reflectance model, noise in the lidar surface return, the lidar calibration coefficients, and the lidar gain coefficients. A better reflectivity model that accurately characterizes phytoplankton, salinity, temperature, and other factors might improve the retrieval, as could a more accurate impulse

response model.

Unlike CALIOP's standard optical depth estimates, ODCOD does not suffer from missing optical depth due to undetected layers because the light from the lidar is attenuated by the entire column before reaching the ocean surface. Therefore, all attenuation in the column is accounted for in the ODCOD optical depth estimate. ODCOD provides an internally consistent

constraint for deriving extinction profiles that, on average, may be improved over the standard CALIOP profiles (Burton et al., 2010; Painemal et al., 2019; Li et al., 2021). ODCOD AOD retrievals are also being assessed for estimating regional layer-averaged lidar ratios for CALIOP by the Models, In situ, and Remote sensing of Aerosols (MIRA) group (Trepte et al., 2023). Because ODCOD is a single instrument technique that is available for the entire CALIPSO mission, ODCOD's total column optical depth estimates provide unique opportunities for analysis and improvements for the CALIPSO mission and

more.

**Appendices**



| Bit Number | Short Name | Description |
|---|---|---|
| 0 | Time delay shifted | Of the measurements provided to the ODCOD algorithm by the surface detection algorithm, ODCOD adjusted the IRM such that the first data point of the surface detection data was not the first point on the IRM |
| 1 | Surface has too many points | The surface detection algorithm provided surface measurements covering a range greater than 120 m |
| 2 | Surface point added to beginning | The ODCOD algorithm added measurements above the surface data provided by the surface detection algorithm |
| 3 | Surface point added to end | The ODCOD algorithm added measurements below the surface data provided by the surface detection algorithm |
| 4 | Surface data missing first point | When solving for the alignment of the IRM, the first measurement that should fall on the IRM curve was not originally provided by the surface detection algorithm |
| 5 | IRM Shifted | ODCOD had to adjust the IRM such that the first measurement provided by the surface detection algorithm was not the first point on the IRM |
| 6-9 | Unused | unused |
| 10 | No surface detected | The surface detection algorithm did not find a surface |
| 11 | The surface is not ocean | The International Geosphere–Biosphere Programme (IGBP) surface type is not 17 for ocean |
| 12 | Surface is sea ice | The depolarization ratio of the surface is greater than 0.15 |
| 13 | Wind speed is invalid | The Corrected MERRA-2 wind speed is outside of the inclusive range $0.025 \text{ m s}^{-1}$ to $43 \text{ m s}^{-1}$ |
| 14 | Time delay cannot be found | ODCOD has failed to find the time delay of the IRM from the surface measurements provided by the surface detection algorithm |
| 15 | Too few measurements | The surface detection algorithm failed to provide enough measurements to solve for the time delay |
| 16 | Area too large | When solving for the IRM area, the solution grew unrealistically large |
| 17 | Scale factor failed | While attempting to solve for the scale factor, a failure occurred |
| 18 | Surface saturation | Surface saturation was detected in the surface return |
| 19 | Negative signal anomaly | Negative signal anomaly was detected in the surface return |
| 20 | Surface had no valid data | The surface detection algorithm provided no valid data for the surface measurements |
| 21 | Bad input data | Can be caused by several conditions related to input data being fill values or invalid |
| 22 | Averaged surface not found by derivative method | The surface detection algorithm had to resort to an alternative method of finding the surface when the surface return was averaged to coarser resolutions that may not be reliable for ODCOD |

**Appendix A: ODCOD QC flag bit representation**

| Date | UTC Time Closest Approach | Day or Night | HSRL Flight Campaign | HSRL Instrument |
|---|---|---|---|---|
| 4 September 2006 | 19:50:37 | Day | GoMACCS | HSRL-1 |
| 20 September 2006 | 19:50:37 | Day | GoMACCS | HSRL-1 |
| 23 May 2007 | 18:32:25 | Day | CALIPSO Validation | HSRL-1 |
| 31 January 2009 | 07:15:03 | Night | CAL_VAL_2009 | HSRL-1 |
| 9 February 2009 | 07:09:13 | Night | CAL_VAL_2009 | HSRL-1 |
| 5 April 2009 | 07:16:17 | Night | CAL_VAL_2009 | HSRL-1 |
| 19 April 2010 | 06:54:35 | Night | CALNEX | HSRL-1 |
| 10 July 2010 | 19:21:24 | Day | CARES | HSRL-1 |
| 11 August 2010 | 17:43:58 | Day | Caribbean 2010 | HSRL-1 |
| 22 August 2010 | 06:29:15 | Night | Caribbean 2010 | HSRL-1 |
| 24 August 2010 | 06:14:35 | Night | Caribbean 2010 | HSRL-1 |
| 26 August 2010 | 06:02:25 | Night | Caribbean 2010 | HSRL-1 |
| 27 August 2010 | 17:40:40 | Day | Caribbean 2010 | HSRL-1 |
| 28 September 2010 | 19:20:36 | Day | Caribbean 2010 | HSRL-1 |
| 19 March 2011 | 07:06:08 | Night | DISCOVERAQ | HSRL-1 |
| 23 March 2012 | 06:52:21 | Night | CAL_VAL_2012 | HSRL-1 |
| 30 March 2012 | 06:59:14 | Night | CAL_VAL_2012 | HSRL-1 |
| 10 June 2014 | 17:48:48 | Day | Bermuda | HSRL-1 |
| 19 June 2014 | 17:43:51 | Day | Bermuda | HSRL-1 |
| 28 August 2020 | 18:21:22 | Day | ACTIVATE | HSRL-2 |
| 18 August 2022 | 07:49:55 | Night | CALIPSO NVF Bermuda | HSRL-2 |

**Appendix B: CALIPSO orbit date, orbit time, day or night orbit, field campaign, and NASA LaRC HSRL instrument flown.**

*Data availability.* The following CALIPSO data products were used in this study: the V4.50 CALIPSO level 1 profile product (Vaughan et al., 2024; NASA Langley Research Center Atmospheric Science Data Center; https://doi.org/10.5067/CALIOP/CALIPSO/CAL_LID_L1-ValStage1-V3-01_L1B-003.01; last access 16 Oct 2022); the V4.51 CALIPSO level 2 vertical feature mask product (Vaughan et al., 2024; NASA Langley Research Center Atmospheric Science Data Center; last access 16 Oct 2022); and the V4.51 CALIPSO level 2 5 km merged layer product (Vaughan et al.,
2024; NASA Langley Research Center Atmospheric Science Data Center; last access 23 Aug 2023). The CALIPSO level 1 and level 2 data products are also available from the AERIS/ICARE Data and Services Center. NASA/LARC/SD/ASDC. (2023). CALIPSO Night Validation Flights High Spectral Resolution Lidar (HSRL-2) Data. NASA Langley Atmospheric Science Data Center DAAC. Retrieved from https://doi.org/10.5067/SUBORBITAL/CALIPSO-NVF/DATA001; and additional HSRL data are available by request from the NASA-Langley HSRL team (John Hair at
johnathan.w.hair@nasa.gov). The SODA product used is developed at the ICARE data and services center (https://www.icare.univ-lille.fr) in Lille (France) in the frame of the CALIPSO mission and supported by CNES and is available through their website. MODIS data is produced by the MODIS Characterization Support Team (MCST), 2017. MODIS Geolocation Fields Product. NASA MODIS Adaptive Processing System, Goddard Space Flight Center, USA: http://dx.doi.org/10.5067/MODIS/MYD03.061 and Levy, R., Hsu, C., et al., 2015. MODIS Atmosphere L2 Aerosol Product.
NASA MODIS Adaptive Processing System, Goddard Space Flight Center, USA: http://dx.doi.org/10.5067/MODIS/MYD04_L2.061. And AMSRE and AMSR2 data are available from the Microwave Climate Data Center Remote Sensing Systems (www.remss.com/missions/amsr)



*Author contributions.* All coauthors have contributed to the paper, and the order in which they are listed is primary author's
best estimate as to their level of contribution. RR prepared the manuscript with contributions from all co-authors, developed
and implemented the algorithm from existing works, and performed comparison analysis. MV provided technical expertise
and performed comparison analysis, SR developed the algorithm wind speed corrections tables, JT provided technical
expertise and authored sections on aerosol type analysis, JR provided technical expertise on ocean surface retrievals, RF and
JH provided technical expertise and analysis on HSRL, and BG processed analysis data.

*Competing interests.* The authors declare that they have no conflict of interest.

*Acknowledgements.* We would like to thank the NASA EVS-3 ACTIVATE project for their efforts to collect HSRL-2 data
under CALIOP overpasses as well as the entire Langley Research Center HSRL team for their collaboration.

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
