# Peer review of "Total Column Optical Depths Retrieved from CALIPSO Lidar Ocean Surface Backscatter"

_Atmospheric Measurement Techniques, 2024_

## Referee Comment (RC2)

Review of

**Total Column Optical Depths Retrieved from CALIPSO Lidar Ocean Surface Backscatter**

Robert A. Ryan, Mark A. Vaughan, Sharon D. Rodier, Jason L. Tackett, John A. Reagan, Richard A. Ferrare, Johnathan W. Hair, Brian J. Getzewich

**General comments:**

This work basically documents an algorithm that is now being used to produce aerosol column optical depth over oceans from CALIPSO surface returns and MERRA-2 10m wind speed. The first part that describes the method is really more suited as an ATBD (Algorithm Theoretical Basis Document) than a research paper. In general, it is a well written and thought-out paper. A strength is that they present comparisons with MODIS, HSRL, SODA and CALISPO layer optical depths. Also important is their treatment of the CALIOP surface return, realizing that it can be saturated and using a fitting method to the instrument impulse response function to retrieve a more accurate measurement of the surface return magnitude. The filtering of suspect data based on surface signal magnitude, depolarization and wind speed is well done.

However, I think the paper is too long. At around 50 pages it becomes a very tedious read. I would suggest breaking the paper into two parts. The first would present the ODCOD algorithm and the uncertainty analysis and the second part would present results and comparisons. This would allow a better and more in-depth description of the surface return fitting to the IRF (which I think at present is confusing) and the results paper to include more examples and comparisons. A comparison that should be added are some island-based or costal AERONET comparisons of column optical depth.

A general question is why don't you use AMSR wind speeds exclusively and drop MERRA-2 all together? Why use AMSR as a correction to MERRA?

It would add value to include a figure showing the number of observations on a map. So, a figure like figure 11 but only showing number of observations. There must be a low number of observations in very cloudy areas such as 50-60S and the North Atlantic.

**Specific comments:**

Abstract: In the opening paragraph it should be stated that this paper presents a documentation of an algorithm that now produces column optical depth in the latest version of the CALIPSO data products and state what that version is.

Line 55 -60: Here you should include other pertinent work such as He et al., 2016 and others not cited.

Line 75: Isn't it the ratio of the measured and modeled return, not the difference?

General: How do you ensure that you are getting the bins associated with the surface return? In situations where you have very low clouds or fog and enough attenuation in the troposphere above, these low cloud layers can masquerade as a surface return. Is the surface return found in L2 processing and you accept that as truth?

Also what about multiple scattering which can be large for low clouds? This will increase the surface return magnitude and result in a lower than actual column optical depth. I do not see a correction for multiple scattering. Here you are filtering out clouds, but they are included in the ODCOD product. In general, multiple scattering is low for low optical depth layers like aerosol, but may be important for some optically thick aerosol layers such as Saharan dust over the ocean.

Line 110: I think you need a sentence like this after line 110: "The combined Htan and Gausian fit to the lab measured IRF points is here after called the instrument response model (IRM).

Lines 165 - 170: This discussion is not clear. The downlinked samples I assume are the lidar profile bins. You say: "The sample time delay is the temporal offset between the actual ocean surface signal onset and the midpoint of the CALIOP onboard or downlinked sample range bin." How can this be calculated? The actual surface return will occur in one or more bins. You do not know where in that 30 m bin the surface return occurred. Also what is an onboard average IRM? I thought the IRM is the fit to the lab measured instrument response model? This is not clear either.

I think what is not clear is the concept of the DIRM – a downlinked IRM? Where does it come from? I don't think you've explained the DIRM so that the reader can understand it.

Your terminology is also confusing. You use "downlinked samples", "onboard samples" and "test samples" What is the difference? Please explain what they are more clearly.

Line 200: How is the surface return "base" determined? Is that part of L2 processing? Why do you limit it to 4 bins? Why not 3 or 5?

Line 203: You write: "If a downlinked sample's time delay places it before the onset of the IRM (i.e. a time delay less than zero), that sample may still be part of the surface return as long as one of the averaged onboard samples is part of the surface." Your wording here is confusing. You reference "downlinked sample" and "average onboard samples". What is the difference between the two? And how do you determine that one of the averaged onboard samples is part of the surface?

Line 245: What about highly attenuated surface returns over a rough sea surface? Would not the below surface scattering be significant in that case? What about clear water less than 60 m deep? In that case you would have a second surface return.

Line 248: Why do you say this when you require the 10 m windspeed in order for your algorithm to work? Please add "and the MERRA-2 windspeed" to the sentence.

Line 285: Change "discontinuities will make the uncertainty estimate less certain" to discontinuities will make the uncertainty estimate less accurate.

Line 487: How did you determine these thresholds: SIAB > 0.0413 sr-1 in the daytime and > 0.0353 sr-1 in the nighttime? What happens if you increase them some?

495: The word "retrivals" is misspelled.

Figure 7: The discussion of this figure is somewhat confusing. You say the upper panel shows the distribution of valid ODCOD retrievals, but yet some of the points are above the limits (red line) you choose as a cutoff. Likewise, in the bottom panel, there are points below the cutoff (red line) and greater than 3 m/s that re not included. Why?

Line 505: What is "MAD"? Please explicitly say what it is.

Figures 7 and 8. You are calling these the distribution of valid ODCOD retrievals. Why then are there points below 3 m/s when you say that you are only using data between 3 and 15 m/s?

Figures 10 and 11. Please indicate the resolution of the grid used to make these figures. For instance is it a 1x1 degree grid?

Line 645-650: It is true that more clouds will inadvertently be included in the daytime measurements but this would tend to make the daytime OD greater than nighttime. So why is night OD larger than day as shown in figure 12?

Lines 669-685. I agree with your statement that averaging the profiles before attempting the retrieval helps in the IRM fitting process, but it is a little confusing when you say you average the profiles that did not have a surface detection. What are you averaging when there is no surface detection? Is the surface signal 0 in the averaging?
Also in this discussion, it is confusing that you are talking about profiles where the surface was not detected since I thought that you were only analyzing cloud-free profiles and those profiles should always detect the surface as the column optical depth for those profiles should always be less than ~1-2 and a surface return would always be present.

---

## Author Comment (AC1)

1. **Original Submission**

**1.1. Recommendation**

Major Revision

1. **Comments to Author:**

**Overall opinion:** The paper reports an attempt to retrieve Aerosol Optical Depth (AOD) using CALIPSO observations via alternative, non-conventional technique based on combination of ocean surface returns and modelled near-surface wind speeds over ocean. The method is called Ocean Derived Column Optical Depth (ODCOD) algorithm and is claimed to be new with regards to previously introduced methods of this type. Besides the aspects I elaborate below (such as methodological unclarities, poor structure of the manuscript, questionable choice of statistical metrics, lack of numerical arguments in the abstract, multiple non-academic English formulations), most critically – the novelty of the method is not explained. What is the fundamental difference with He et al. 2016 method for retrieving AOD using CALIPSO or with Venkata and Reagan 2016 approach? If there is a methodological difference, it has been not emphasized enough in the abstract. Moreover, it is not explained why can't you simply use surface integrated attenuated backscatter signal included in the official CALIOP output for linking it to near-surface ocean wind speed, so you need to find the CALIOP IRM. Does official CALIOP algorithm have problems with identifying the surface bin/bins boundaries? I guess the answer is hidden somewhere on page 9 in among the statement like *"In an ideal signal, any measurement that is not part of the surface return would be completely zero and any measurement on or after the surface return onset would be non-zero. However, the measured downlinked samples before the surface return onset are rarely if ever actually zero so it is critical that whatever consecutive sample pair is selected is part of the surface return."*. Without going deeper into that, I think you should articulate this unique aspect and advantages of your retrieval better in the revision. See the other comments below please.

Thank you for the thorough and detailed review of an admittedly long paper. While this comment touches on several areas the paper needs improvement, we feel it most notably highlights that neither the abstract nor the text emphasize enough how this new attempt at estimating atmospheric optical depth from lidar ocean surface returns is unique and has value over other similar established methods. This paper attempts to describe the ODCOD retrieval algorithm, and then demonstrate to the reader that the algorithm results are an acceptable estimation of the full column particulate optical depth. Beyond the algorithm description, we quantify the random uncertainty and then qualitatively characterize the performance and some systematic differences to other established datasets. The reviewer has also quite fairly pointed out that the differences between ODCOD and other publish techniques that utilize the ocean surface lidar return are not articulated well and we have added descriptions of the differences in the introduction which is summarized well by the lines starting at line 91:

> *While these retrieval advances offer new ways to estimate column optical depths, one drawback is that each technique requires measurements from multiple*

*instruments, which introduces collocation uncertainties along with random and systematic retrieval uncertainties from multiple sensors. A further impediment is that the requisite CPR and AMSR measurements are only available for parts of the CALIOP mission. By combining ideas from previously developed methods and replacing the AMSR wind measurements with wind data obtained from the Modern-Era Retrospective analysis for Research and Applications, Version 2 (MERRA-2) reanalysis (Gelaro et al., 2017), a new hybrid algorithm is constructed to estimate full column particulate optical depths from the lidar ocean surface return over the entire CALIPSO mission.*

Addressing the novelty of the method not being explained well, while the ODCOD algorithm is "new" in that this particular collection of techniques has not been done before, we do not intend to claim ODCOD is particularly "novel" and it is not fundamentally different than He et al. (2016), VR2016, or even Josset et al. (2010). In fact, it is very closely related and only deviates from their proposed algorithms when another algorithm does something in a way that better fits the ODCOD algorithm, or the team felt the deviation was an improvement. One major difference is each of the described techniques require other datasets external to CALIOP to perform their retrievals and thus cannot be applied to the entire CALIPSO mission as mentioned in the excerpt above.

The fundamental differences to He et al. (2016) is their use of AMSRE winds as an input to their algorithm and the discrete approximation of the surface integrated attenuated backscatter where ODCOD utilizes the CALIOP response model (CRM) fitting procedure. A discussion as to why the CRM fitting procedure was used is now added as Appendix A and discussed more thoroughly in the text. We have added reference to He et al. (2016) in the introduction and go into more detail on the differences between He et al. (2016), VR2016, and Josset et al. (2008).

To the comment regarding numerical arguments, the performance assessment presented is not intended to be a through validation of ODCOD but rather a brief assessment to see where ODCOD falls among three distinct datasets. The HSRL instruments provide very accurate measurements but are only available in one region of the world and have a limited number of underflights with well collocated data. The global scale AOD measurements of MODIS that we believe many in the community feel is the "gold standard" of global AOD measurements provide an assessment of how ODCOD performs globally but MODIS is only available in the daytime. SODA provides a less validated estimate but is available both day and night data and is collocated and reported on the same CALIPSO 333m footprint. SODA is also a similar ocean surface retrieval approach to ODCOD. Performing a more in-depth validation would significantly extend an already very long manuscript as well as push back the publication of this method to well beyond ODCOD's release date. However, omitting all analysis would not provide the reader with any impression of if the ODCOD algorithm has any validity as a data product. Other works in preparation will also be comparing ODCOD to AERONET in near future (See Thorsen et al. 2024 in preparation). We hope that the edits

based on the reviewers' comments have improved the paper and address these general comments. The abstract has been edited to emphasize that this is a measurement technique rather than a validation paper however, we have also added correlation coefficients to better express how well ODCOD compares to the datasets with which we compared to.

**2.1. Comments:**

1. **Abstract:** The paper is very long, but the abstract reports only five numbers of statistical agreement with references. This reflects a general problem of your manuscript that was confirmed after I fully read it. For instance, you are discussing the noisiness of your retrievals depending on the shot averaging strategy (lines 694 ...), but you have not quantified the rate of this noise (or temporal variability?). Thus, you have no numbers to report for the abstract. This is a systematic problem that can be seen throughout the entire "Results" section that ultimately yielded a critical lack of numerical arguments in your abstract. Moreover, I think that the abstract is imbalanced because you give too much introductory-alike information about the method itself and insufficient information about the findings form your study. Also, as mentioned above, this method does not look like "new" method, but it is rather built on the heritage of previous CALIPSO-based works (see comment about the introduction below). I also think clearer scientific implications of your method should be given in the end of the introduction. Namely, why this method is beneficial for scientists or users, compared to previous ODCOD-alike methods like He et al. [2016] method or conventional extinction-based AOD retrievals. Lastly, it is not clear from the abstract whether you are retrieving only aerosol optical depth (AOD) or total optical depth including molecular contribution.

The paper's purpose is to describe the ODCOD method and is not presenting a comprehensive validation of the ODCOD dataset. We have made changes to attempt to articulate this more clearly.

One significant scientific implication is that ODCOD does not assume a lidar ratio to estimate the full column optical depth where the traditional CALIOP approach does. Another is that particulates, water droplets, and ice crystals in the atmosphere who's attenuated backscatter is too weak to be directly detected by the lidar are not included in the CALIOP estimated column optical depth however, ODCOD estimates the optical depth for the full column. We have made changes to make this clearer in the introduction.

Regarding the noise discussed on line 694 of the old document, this comment was meant to only highlight that by inspection of Fig. 14, a reader could see that the averaging technique employed by ODCOD reduced the spread of the retrievals. However, to better quantify the noise, we included a running window mean fit on the plots and used it to quantify a mean squared error and described the differences in the text.

The molecular component is estimated from the MERRA-2 inputs, and factored out in the same way as the CALIOP system to report a particulate only optical depth where particulates are meant to indicate aerosols, water droplets and ice crystals. We have made edits in several places to make this clearer and to define particulates as both aerosol particles as well as hydrometeors.

1. **Introduction:** The introduction reminds a section from technical report of CALIPSO product or algorithm development document, not the introduction of a scientific study.
   - I think you should include paragraphs describing importance of retrieving AOD using spaceborne lidars and which gaps remained in this research field so your attempt looks justified.

In the introduction, we comment on how CALIOP's standard column optical depth estimates are unable to estimate optical depth of all particulates in the column since some will fall below the lidar's detection threshold. This leaves a low bias in any retrievals that would use CALIOP's estimates such as global mean aerosol direct radiative effect as commented by Thorsen at al. (2017). We then explain how ocean derived estimates do not have this problem and is the driving force for developing ODCOD. We hope these edits will highlight the importance.

The paragraph starting at line 53 describes the CALIOP technique of estimating column optical depth and the deficiencies of that technique.

> *Among the primary science data reported in the CALIOP data products are vertically resolved estimates of particulate extinction coefficients, their associated layer optical depths, surface detection and altitude, as well as estimates of wind speeds obtained from MERRA-2. To retrieve extinction coefficients, CALIOP first uses a feature detection algorithm to identify regions of the vertical profile with elevated attenuated backscatter (Vaughan et al., 2009), and then prescribes an extinction-to-backscatter ratio (i.e., lidar ratio) for various aerosol types based on the CALIOP aerosol classification and cloud/aerosol discrimination algorithms (Liu et al., 2019; Avery et al., 2020; Kim et al., 2018; Young et al., 2018). These prescribed lidar ratios are among the largest sources of uncertainty and error in the particulate extinction retrieval and they become increasingly significant lower in the atmosphere due to attenuation and rescaling errors inherited from overlying layers (Young et al., 2013). CALIOP also only retrieves extinction coefficients for regions of the vertical profile where the particulate attenuated backscatter signal rises above the layer detection thresholds (Young and Vaughan, 2009). Regions of faint scattering from diffuse particulates can fall below these limits and hence go undetected. This inherently means that a small fraction*

*of the overall particulate extinction will not be included in CALIOP's column optical depth estimates. Kim et al. (2017) estimate CALIOP's undetected optical depth to be on the order of 0.030 ± 0.046. Based on comparisons to Moderate Resolution Imaging Spectroradiometer (MODIS) AODs, Toth et al. (2018) report similar low bias estimates of 0.03 to 0.05 for daytime retrievals. Consequently, estimates of global mean aerosol direct radiative effect derived from CALIOP's standard aerosol optical depth (AOD) retrievals are biased low by ~54 % (Thorsen et al., 2017).*

The next paragraph starting at line 70 then describes some of the prominent ocean surface estimation techniques that ODCOD is built upon and how they estimate the full column without typing and lidar ratio assumptions.

*The low bias of CALIOP's standard optical depth product highlights a need for improved lidar retrievals to estimate the optical depths of the full atmospheric column. Previous studies have estimated the ocean surface integrated backscatter coefficient using only surface wind speed and viewing angle (Barrick et al., 1968; Bufton et al., 1983; Menzies et al., 1998; Lancaster et al., 2005; Hu et al., 2008). Their works make it possible to estimate the optical depth of the atmospheric column when the lidar signal is detected from the ocean surface. These techniques can be applied without assuming lidar ratios and incurring the uncertainties associated with them. Reagan and Zielinskie (1991) recognized that column optical depths could be estimated using "the strong return signals from ground/sea reflections to improve upon information that can be retrieved from spaceborne lidar observations." Leveraging the close formation flying of the A-Train satellite constellation, Josset et al. (2008) devised an innovative technique to retrieve column optical depths by synthesizing measurements from CALIOP, the Cloud Profiling Radar (CPR) aboard CloudSat, and the Advanced Microwave Scanning Radiometer (AMSR) aboard Aqua. Using only CALIOP and AMSR measurements, Venkata and Reagan (2016) (hereafter VR2016) developed a column AOD retrieval based on prelaunch laboratory characterizations of the CALIOP 532 nm detector system response. Using the same satellite measurements, He et al. (2016) formulated another approach for estimating "clear sky" optical depths. Each of these ocean surface retrieval techniques require an accurate estimate of the ocean surface integrated attenuated backscatter (IAB). Josset et al. (2008), He et al. (2016), and others integrate height resolved CALIOP measurements around the ocean surface. However, techniques which approximate an integral from discrete sampling can introduce truncation uncertainty if the original signal is under sampled. VR2016 chose to employ a novel technique of fitting a piecewise function to approximate the measured lidar pulse shape of the CALIOP post detector electronics and integrating the fit function to estimate the ocean surface integrated attenuated backscatter. Applying this approach avoids the*

*systematic uncertainties associated with discrete integration techniques and provides an accurate estimation of the surface IAB.*

The paragraph after that starting at line 91 discusses why another similar technique is needed which is that all the preceding techniques required data from other sources that are not available for the full CALIPSO mission.

*While these retrieval advances offer new ways to estimate column optical depths, one significant drawback is that each technique requires measurements from multiple instruments, which introduces collocation uncertainties along with random and systematic retrieval uncertainties from multiple sensors. A further impediment is that the requisite CPR and AMSR measurements are only available for parts of the CALIOP mission. By combining ideas from previously developed methods and replacing the AMSR wind measurements with wind data obtained from the Modern-Era Retrospective analysis for Research and Applications, Version 2 (MERRA-2) reanalysis (Gelaro et al., 2017), a new hybrid algorithm is constructed to estimate full column particulate optical depths from the lidar ocean surface return over the entire CALIPSO mission. This new algorithm, called Ocean Derived Column Optical Depths or ODCOD is implemented in the CALIOP Version 4.51 (V4.51) release of the Lidar Level 2 (LL2) CALIPSO data products. ODCOD is developed primarily from the work of VR2016 but also incorporates techniques from Josset et al. (2008) and Hu et al. (2008). In contrast to previously developed methods, ODCOD estimates are based solely on CALIOP measurements and MERRA-2 wind data. MERRA-2 also provides the profiles of number density, temperature, and pressure profiles that are used to calculate molecular and ozone two-way transmittances between the top of the atmosphere and the ocean surface. ODCOD retrievals of 532 nm optical depths are reported wherever a qualified ocean surface return signal is available.*

○ Moreover, your paper is partially built on the heritage of previous works, which exploited lidar surface returns over water (or land in some cases) to retrieve AOD. However, you only selectively included some works that were focused on this topic. I'd suggest you to include or go quickly over the following studies including fundamental works, demonstrate relationship between ocean surface returns and wind speed [Barrick, 1968; Bufton et al. 1983; Menzies et al, 1998], also mostly CALIPSO-based works on this topic [Josset et al., 2008; 2010; 2011; 2018; He et al. 2016], but also Aeolus-based studies [Li et al. 2010; Labzovskii et al., 2023] Why You can find most these works easily in Google Scholar by keywords or cross-references. Fundamental works here serve as a useful introduction into your research niche, CALIPSO-based works demonstrate what has been done already by

previous ODCOD-alike methods using CALIPSO (so which research gap you intend to close) Aeolus works indirectly indicate that CALIPSO is best-suited for such retrievals due to weakness of Fresnel reflection and weakness of ocean surface returns at non-nadir incidence.

Many of the works mentioned are cited and commented on such as Menzies et al. 1998, Josset et al. (2008), Josset et al. (2010a), Josset et al. (2010b), Josset et al. (2012), and we have added the He et al. (2016) reference that was kindly provided. Barrick (1968) and Bufton et al. (1983) while important precursory works, were not directly used in ODCOD and are the works that Lancaster et al. (2005) were built on and are cited. Additionally, while Li et al. (2010) and Labzovskii et al. (2023) are very interesting and use similar ideas, they are tangential works since their goal was estimating surface reflectance and don't provide much additional information to the reader on the utility of ODCOD.

1. **Methodology:** There is a room for improvement in the methodological part of the paper.
   o First, please avoid using unintroduced terms right away like "modelled surface return" or "idealized impulse return model" which might be confusing for a general reader.

We have completely rewritten Section 2: Ocean Derived Column Optical Depth Technique. We have tried to better describe what the modelled surface return and response model is and have changed what we call it to the CALIOP response model (CRM) because part of what is unique about the VR2016 technique and subsequently the ODCOD technique is that we are modelling the measured CALIOP electronics response rather than an idealized impulse response.

   o Second, the methodological description is not logical. From general reader perspective, it is more sensible to first describe CALIOP instrument with its specifications and the CALIOP optical parameters you used as input and only then start describing methodology behind retrieving optical depth using surface echoes. Otherwise, you assume that a general reader is familiar with a given lidar system, which is not true.

We now describe the CALIOP instrument in the introduction and have touched on the parameters used in the ODCOD algorithm so that the reader is more familiar with the CALIOP system and available data before discussing the ODCOD algorithm.

   o Cloud screening issue. Although you discussed the inclusion of the quality flags through "additional screening" (2.2.3) I have not noticed any discussion on how you tackled thick cloud cases and very hazy aerosol conditions. Assume you have a thick cloud undetected – it will attenuate your surface

echo not because the ocean roughness has changed, but due to atmospheric extinction. Many previous studies exploiting lidar surface returns have previously discussed this issue. Labzovskii et al. [2023] in the Aeolus-focused work showed that the resultant lidar surface return statistics in the cases where (a) clouds were not filtered and (b) clouds were filtered are very different. The presence of clouds and high aerosol load cases will plague the statistics with attenuated, weakened and noisy surface returns not suitable for ODCOD-alike retrievals.

Unlike Labzovskii et al. (2023), ODCOD is attempting to estimate the attenuation rather than the attenuation being a problem that must be avoided or corrected for. The inverse problem occurs with ODCOD where any deviations from expected surface reflectance will result in a systematic error in the ODCOD optical depth retrieved. Thick clouds and very hazy aerosol conditions are not a particular problem for the algorithm unless the surface return is not able to be identified i.e. the column is totally attenuated to the lidar in which case, ODCOD does not attempt a retrieval. Water clouds do provide erroneous values and retrievals with water clouds are not well understood which is why we recommend the data be filtered for cloud-free, aerosol-only profiles. Recommendations on ways to do this are provided in the text however, filtering for quality retrievals has to do with data analysis and not with the ODCOD algorithm itself. Cloudy retrievals should be ignored if the researcher is attempting to study AOD. A researcher should consider the recommendations provided in the manuscript in section 3.1.1 which include using the CALIOP detections to determine "aerosol-only" profiles. Aerosol-only profiles with high aerosol load are no issue for ODCOD unless the column becomes opaque to the lidar which happens extremely rarely (less than 0.1%). Higher optical depth aerosol-only profiles may be slightly noisier due to the reduced surface return signal but otherwise are fine for retrievals. If there was a problem with the filtering recommendations allowing cloud contamination this would be evident in the MODIS AOD comparisons by a significant number of ODCOD reporting very high optical depth but MODIS reporting much lower.

- o Molecular contribution issue. As I said, it is not clear whether you are retrieving only aerosol optical depth (AOD) or total optical depth including molecular contribution. Please clarify this aspect and explain how you tackled molecular contribution (even if it is small) if you retrieved AOD only.

ODCOD retrieves full column particulate optical depth which includes particulates like aerosols as well as water droplets and ice crystals. If a researcher wants to study specifically AOD, they must filter for ODCOD profiles where CALIOP had only detected aerosols in the profile, and this will eliminate all detected clouds. We attempted to clarify originally at line 132 in the old document that the molecular and ozone contributions are removed, "As in standard CALIPSO processing, $T^2_M$ is estimated from molecular and ozone number densities obtained from the MERRA-2

model (Kar et al., 2018)" but this information should be more prominent, so we have also explicitly stated the retrieval as full column particulate optical depth in the abstract, introduction, and in the technique description. We have added text explicitly describing what we are defining as particulates in the paper, "*a new hybrid algorithm is constructed to estimate full column particulate (i.e., cloud and/or aerosol) optical depths from the lidar ocean surface return over the entire CALIPSO mission*" and have also better described how the molecular contributions are factored out in the ODCOD retrievals.

- o Line 484 "samples are averaged vertically prior to downlink, surface saturation can still go undetected" this is critical. In other words, you are stating that your quality flagging procedure is not effective?

That is correct. The surface saturation flag included in the standard CALIOP data products is not effective enough if the intention is to retrieve using the surface signal. This is why section 3.1.1 suggests more stringent ways to filter for surface saturation.

- o If you refer to some parameters totally unknown to a general reader like SIDR (which is even worse, it is unintroduced acronym) or MODIS confidence flag, ensure you explain what are these and ideally – where they can be found.

SIDR was first used on line 462 of the old document and is the introduction of the acronym, "and **surface integrated depolarization ratio 532 nm** (SIDR) < 0.05 improves agreement of ODCOD to the other datasets," emphasis added here. We have added additional information regarding the MODIS data products quality flags in section 3.1.3. Starting at line 574:

*The MODIS Quality_Assurance_Ocean science data set (SDS) is used for MODIS data quality screening. This SDS is a 5-byte composite informational flag that includes a retrieval QA Confidence flag (QAC) and a QA usefulness flag (QAU) (Levy et al., 2009). The possible QAC flag values are 0 to 3 and indicate confidence levels that are poor, marginal, good, and very good, respectively. The QAU flag values can be either 0 or 1 and indicate not useful and useful data. MODIS data are chosen such that the QAC flag is marginal or better and QAU flag indicates a useful retrieval (Levy et al., 2009).*

- o I think the entire section 3 belongs to the methodology as a sub-section, but under some different name. The title 'Performance Assessment' is currently misleading and sounds like you are going to present results.

Section 3 is the methodology of the performance assessment and is not related to the methodology of the ODCOD algorithm which is presented in Section 2. The performance assessment is characterizing the performance of ODCOD and

systematic differences between ODCOD and other established datasets. HSRL has the highest accuracy when measuring AOD so is a good measure but is limited to only one region, MODIS we feel is seen as the "gold standard" by the community for global AOD measurements, and SODA is matched to the CALIOP and thus ODCOD footprints and utilizes a different but similar approach as discussed in section 3.1. Each of these performance assessments are meant to help the reader understand systematic differences between the datasets and how ODCOD performs to these more well-known quantities. It is most certainly not a validation which would take many more pages to cover in detail and is not a good fit for a measurement technique paper.

2. **Results:**
   - The first problem is that you start demonstrating your results from three year seasonal medians. I assume you realize how many things are put together into these three years? Would it be logical to go from smaller time and spatial scales (orbit-based or something like this?).

Smaller time and spatial scales are a more detailed kind of analysis and would better fit a validation paper which would hopefully look at shot by shot variability, regional variability, as well as global variability. However, it is too involved for this already, "very long" paper that is only attempting to introduce the ODCOD method. This performance assessment only looks at how ODCOD compares globally to MODIS and SODA and to as much data as available with HSRL.

   - In line with this recommendation, I think the validation results should go before analyzing general patterns of AOD behavior from ODCD algorithm plotted on maps (Fig. 11). We first need to understand how biased is your AOD retrieval and then after understanding the actual validity of ODCD algorithm we can see how AOD from ODCOD is distributed regionally.

Figures 10 & 11 introduce the global patterns that ODCOD sees and provides a baseline for looking at the bias day to night and the subsequent internal analysis (Figs. 12 & 13) and then later, understanding the global patterns compared to MODIS (Fig. 17). This starting point answers the question, what does ODCOD even look like (section 3.2.1) and then in section 3.2.2, is it internally consistent. Without this baseline, the comparisons or biases to other datasets don't have much meaning so we therefore elect to present them first. The point of the performance assessment is to determine if ODCOD provides estimates of optical depth that are in line with our expectations and general understanding of where optical depth, in this case aerosols, persist globally.

   - The ODCOD-reference comparison lacks a detailed scrutiny. You report median/mean differences between hugely populated samples of 3-year seasonal global aggregates. What such differences could potentially tell

you? "Month-specific correlations and biases would be much more appropriate to make any conclusion here. This applies to comparisons vs HSRL, MODIS and SODA.

Since all data are matched one-to-one, the 3-year global aggregates tell the reader on a large scale if ODCOD is systematically higher or lower than the comparing dataset. By using such a large volume of data, we reduce random uncertainty such that the systematic error becomes clearer. Breaking it down seasonally helps to determine if there is some seasonal variability in the comparisons which could be introduced from inputs to the ODCOD algorithm or from the comparing dataset. Breaking it down smaller to month specific correlations wouldn't provide much new information for a general assessment and would drag out a paper that already, "is very long."

1. **Conclusions**: I think conclusions should be revisited after revision and better aligned with results and abstract. Don't forget to emphasize why your method of AOD retrieval is unique and different from previous SIAB-based attempts to derive AOD using CALIOP data.

Of the ocean surface retrieval techniques reviewed, ODCOD is the only technique able to be applied to the entire CALIPSO datasets and the small differences to other ocean surface techniques are now discussed well in the introduction. We feel the key points are; ODCOD is now available for the entire CALIPSO mission as an integral CALIPSO level 2 data product which other techniques cannot; ODCOD compares reasonably to other established datasets; ODCOD shows differences day to night that can be attributed to sampling bias and is not a bias of the algorithm itself; ODCOD has a random uncertainty estimate of 0.11 ± 0.01; ODCOD provides an optical depth estimate for the full column unlike CALIOP which will not include particulates below the lidar's detection thresholds.

1. **Language and Format:** Although I am not qualified to evaluate the language of the article (being non-native English speaker), I still notice the use of non-academic style in writing. Short forms of verbs like "don't" or non-academic formulations like "...time delay tells us" or "this is unfortunate" are spotted. Please upscale the style of your writing to the minimum requirements of academic English.

Any contractions such as "don't" left in the manuscript are an editing oversight and we have attempted to eliminate them.

**Minor comments (line references on the left)**
107 – Which concepts they outlined? Readers are not aware about that.
Section 2 in general has been heavily re-written and the details of the CRM and fitting procedure are hopefully clearer now.

128 – Particulates = Aerosol particles? Particulate matter?

This was intended to mean any atmospheric material that is not considered molecules. We have added the description to the abstract at line 13, "*ODCOD uses the lidar integrated attenuated backscatter from the ocean surface, together with collocated wind speed estimates from Modern-Era Retrospective analysis for Research and Applications, Version 2 (MERRA-2), to estimate the full column optical depths of particulates (i.e., clouds and aerosols) in the Earth's atmosphere.*" As well as section 2.1 line 117, "*The particulate optical depth ($\tau_P(z_s)$) of both aerosols and hydrometeors is related to the particulate two-way transmittance ($T_P^2(z_s)$) at the range from the receiver, $z_s$ by the relationship $\tau_P(z_s) = -1/2 \cdot ln(T_P^2(z_s))$.*"

144 – You can use Li et al. [2010] Aeolus-based work as a reference to justify this statement.

During the re-write of Section 2, this information is no longer discussed. The information is more relevant in discussions about the uncertainty and is addressed there as a consequence of the reflectance model used. We would prefer to not confuse this point with references to other works when the point we are attempting to make here is that the equation for the whitecap fraction will mathematically increase regardless of if it is physical or not, even though it is in fact physical. It is good to know Li et al. (2010) found agreement with the work of Lancaster et al. (2005) but that is outside the scope of what this section is discussing.

152 – estimate or take from references?

This was calculated from Fresnel equations and confirmed in Vaughan et al. (2019). We have changed from, "we estimate as" to, "is estimated as" for simplicity.

154 – Few words why you chose Hu et al. (2008) approximation and not previous approximations like Wu et al or anything else? For clarity.

We changed the next sentence to "*This approximation was chosen because it was developed using CALIOP measurements and AMSR wind speeds and directly relates the two primary quantities used in the ODCOD retrieval and is shown in Eq. (10):*"

172 – "Don't" -> do not use short versions of verbs like this in academic studies please.

We have corrected the contraction.

191 – To take the ratio of what?

We have made significant edits to this section with the goal of making the text clearer. The ratio being taken is of the two largest attenuated backscatter measurements of the ocean surface. Starting at line 280 we have, "*The CALIOP surface detection algorithm provides the downlinked measurements of attenuated backscatter of the ocean surface. The two largest consecutive measurements of the detected surface return are selected and the first of the two is chosen as the reference measurement. The ratio of these measurements is compared to ratios of samples of the DCRM until a matching ratio is found to find the reference measurement time.*" Where DCRM stands for downlinked CALIOP response

model which is related to the CRM described in detail in the text as well as descriptions of the reference measurement and how it is used to align the measurements to the DCRM.

197 – What is "the largest measurement of the suspected surface return"? By magnitude? Suspected surface return is hidden where? In which parameter here?
In this section we are discussing fitting the CALIOP response model to the measured attenuated backscatter signal of the surface return. Starting at line 133, we have, "*Surface IAB estimates are reported by CALIOP's surface detection algorithm which uses the trapezoid rule to numerically integrate the ocean surface return between the CALIOP detected top and base altitudes (Vaughan et al., 2017). This approximation of surface IAB suffers from underestimates of 3 % to overestimates of 2.5 % that arise from discrete integration of a "hard target" return like the ocean surface when recorded by CALIOP's receiver system (see **Error! Reference source not found.**). In contrast ODCOD uses a technique which is largely identical to the method described in VR2016. ODCOD constructs a model that approximates the CALIOP post-detector electronics response to a laser pulse, fits the attenuated backscatter measurements of the surface return to the modeled response shape, then analytically integrates the model to solve for the area under the surface return pulse, and calculate the surface IAB.*"

200 – Even over mountainous regions? It is hard to make my own conclusion here because you just state and announce what you do/did methodologically without explaining why you introduced such assumption in many cases including the explanation here. Apply here and elsewhere. "...surface detection failure is suspected" in such case because "abc... xyz... -> **reason why you think such assumption is valid ideally supported by numerical arguments, references or common sense***"
ODCOD is estimated from ocean returns so mountains don't come into play. If the number of downlinked samples between the CALIOP detected surface top and base exceed four range bins, then the surface is being reported as broader than the CRM. Since we know an ocean return should be the shape of the CRM, a surface detection failure is suspected. Since this falls under a, "not qualified" surface return and is an algorithm detail that is part of the QC flags, this discussion was omitted from section 2.

210 – Time delay between what and what?
In the section 2 re-writes, we have attempted to improve the descriptions of the time of measurements and time of CRM pulse onset. Line 158, "*The CRM is defined by Eq. **Error! Reference source not found.** where t quantifies the elapsed time between the CRM model pulse onset and the data acquisition time of the measurements of the surface return.*"

225 – Where have you found these biases? In order to make a statement like this, you need either to rely on your new or previous results (reference).
The discussion of the biases is now in section 2.2.1 and other than our investigation described, the biases are also discussed in (Carvalho 2019).

233 – The knowledge about total extinction of 532 nm light few meters below the ocean surface is not coming from VR 2016 reference, but from previous studies. Do not overexploit the references from a specific expert pool when you refer to results, previously published by other studies and more specifically focused on these topics please.

The following quote is taken directly from VR2016, "While water is opaque to 1064 nm light, it is slightly transmissive/transparent at 532 nm, although light at this wavelength is largely extinguished within a few tens of meters below the water surface..." and therein they calculate the subsurface backscattered light is extinguished to nearly zero (within 0.1%) within 45 m below the surface. VR2016 of course uses values from various sources to make this derivation such as Churnside et al. (2014) but the derivation is reported in VR2016, so we have used VR2016 as the reference for this statement. The new line in section 2.1 reads, *"Another source of systematic error that affects ODCOD retrievals is ocean subsurface scattering. VR2016 show that the 532 nm light is largely extinguished to less than 0.1 % within 45 m below the surface but will make a small contribution to the overall return."*

236 – Can you find a reference to prove this statement about insignificance of subsurface return at this wavelength? Perhaps, Josset et al. [2010] paper about ocean surface reflectance model for different angles and wavelengths or Li et al. [2010] study that applied similar model for Aeolus pre-launch study where different incidence angles were tested. Also, what are the "two largest points"?

We have edited the text to be more transparent about subsurface scattering. During development of the algorithm, we found any attempts to correct for a subsurface component consistently underestimated the IAB and the suspicion was due the CRM fitting only the two largest measurements of the surface return, the full subsurface contribution was probably not being captured due to the contribution being not the same for each measurement of a given surface return. The new statement reads:

> *Another source of systematic error that affects ODCOD retrievals is ocean subsurface scattering. VR2016 show that the 532 nm light is largely extinguished to less than 0.1 % within 45 m below the surface but will make a small contribution to the overall return. Due to the CALIOP onboard electronics system, subsurface scattering will effectively widen the surface pulse but does not introduce a uniform enhancement of the measurements. Some subsurface enhancement will occur in the individual measurements as a function of their respective pulse onset time delays, but fitting only the largest two points of the CRM means the magnitude of any enhancement will vary as a function of time delay. Attempts to correct for subsurface contribution using conventional theoretical corrections as proposed in VR2016 and Josset et al. (2010b) consistently underestimates the IAB, so ODCOD applies no such correction. The failure of the CRM to avoid the subsurface component of the ocean surface return introduces a small systematic error that needs additional study to fully understand and quantify before a correction is attempted.*

239 – Once again, a very crude assumption without attempt of justifying it from previous works' experience or sensitivity analysis.

At this time, we are stating the assumptions used in the ODCOD algorithm. We have not quantified how much of a contribution subsurface return might make to ODCOD and is likely a function of where within the 30 m range bin the beginning of the surface return lies. We have found applying a subsurface correction overestimates the optical depth and believe it is due to the CRM being designed to fit the air-ocean interface return only and not the combined air-ocean interface and subsurface return which is superimposed. Sensitivity analysis is certainly a topic we hope will be covered in a validation work but is not attempted here.

241 – Do you refer to saturation or attenuation here? I am sure there is an optical term to nail down this effect.

This is not referring to saturation or attenuation. The best term might be ring in the detectors. The effect is described in more detail in McGill et al. (2007). The detectors cannot recover fast enough when a large signal is measured so there is a "tail" seen in the signal. But, since the "tail" is in the measurements after the large signal and the data points of the tail are not used to fit the CRM as previously discussed in the section, this does not affect the estimated magnitude of the surface return in the ODCOD algorithm.

245 – 'Largest points of surface returns' is ambiguous term even for lidar experts, so it will be a puzzle for general readers.

The line in section 2.1 now reads, "*However, since only the two largest measurements of the CALIOP detected surface return are used in the CRM fitting process, the CRM overcomes possible effects from the non-ideal transient recovery as any enhancement in the tail of the return does not affect the scaling of the CRM and thus the estimated magnitude of the surface return.*" This means the largest measurements within the range of the CALIOP surface detection algorithm's indicated surface return measurements of attenuated backscatter.

264 – Another crude assumption without justification here. This seems to be a systematic problem in the paper.

To justify our assumption, we have included the following text in section 2.2, "*The standard deviation of the off-nadir angle is estimated from internal CALIPSO engineering documents as 0.16 ° making its contribution to the overall uncertainty approximately 0.01 % of the uncertainty overall. Uncertainty in the MERRA-2 model temperature data used to estimate particulate and ozone two-way transmittance for ODCOD are estimated to be less than 1.5 K (M. Rienecker, personal communication, 2013). Even assuming a uniform 4 K error in the MERRA-2 temperature profile, the fractional error in the molecular two-way transmittances is on less than 0.004. Using this as an estimate of the random uncertainty, the contribution to the overall uncertainty should be less than 0.02 %. As these uncertainties are small compared to that of wind speed and area fit, they are not included in the ODCOD uncertainty estimates.*"

286 – Unfortunate? Does not sound good for academic paper.

These discontinuities are and will always be part of the CALIOP version 4.51 data product and while a better estimate could be made with more time and analysis, they are within the version 4.51 data. It is unfortunate that we cannot improve upon the already published and reported values in that product, but we have changed the line in section 2.2 to, "*This reduces the utility of the uncertainties reported because…*" to be more informative to the reader.

287 – Please introduce the commonly accepted term AOD (Aerosol Optical Depth) for a reader and stick to it in the paper instead of "aerosol-only optical depth".

It is important to note that ODCOD does not retrieve AOD but rather a full column optical depth and to study AOD with ODCOD, filtering must be performed to select cloud-free, aerosol-only profiles. We have attempted to make this clearer in several places and have changed aerosol-only optical depth to AOD in most cases.

304 – Call figures either Fig. or Figure please

We believe that the AMT guidelines are to use Fig. within a sentence and Figure at the beginning of a sentence. But we will defer that to the copy editor.

305 – Outside which range?

We have changed the line in section 2.2 to, "*being much outside this wind speed range.*"

306 – Can it be also related to the effect of noise at low AOD values?

It is unlikely to be related to noise at low optical depth because the SNR of attenuated backscatter from the ocean surface is exceptionally high which is why surface saturation is causing the sampling bias.

308 – What is "statistically higher wind speed"? From significance point of view or?

We have changed the wording in section 2.2 to, "*will have systematically higher wind speeds.*"

319 – This raises a question on what is the skill of your AOD retrieval in terms of precision? What is the threshold below which, one can assume AOD output value as noise? AOD = 0.05? 0.03? Please reflect on this

Noise will most significantly affect the high optical depths as that is where the surface return attenuated backscatter measurements will be very small. When optical depth is very small, the measurements are very large and the signal to noise is significantly better. Where a lower cutoff does occur, it is due to surface saturation and is different day to night. The random uncertainty of the estimate is stated in the conclusion of section 2.2, "In general, when filtered for wind speeds between 3–15 m s$^{-1}$, ODCOD AODs have an uncertainty on the order of 0.11 ± 0.01 (75 % ± 37 % relative) day and night."

329 – 'Mixture of retrievals', what is the mixture of retrievals? You use many unclear terms like this in your paper.

We have re-written portions of section 2.2 to be clearer and no longer use the term "Mixtures of retrievals."

322 – What is bright surface return in terms of backscattering? Strong backscattering?
Yes, we meant in terms of backscatter. Lines near 354 now read, "*The increase in the median wind speeds for AOD near and below zero as well as the crossing of the means and medians of the uncertainties are mostly due to the sampling bias caused by detector saturation by the surface return. More specifically, since ODCOD retrievals are not performed for saturated surface returns and surface saturation occurs more frequently at lower wind speeds and lower optical depths, the returns that do qualify for quality ODCOD retrieval will have systematically higher wind speeds when optical depths are low. This surface-saturation sampling bias is discussed in more detail in Sect. 3.1.1.*"

329 – Do mean and median random uncertainties have statistical sense for non-systematically biased systems?
In this case, yes. The systematic bias is in the selection of the uncertainty estimates, not the uncertainty calculation itself. The means and medians changing at lower wind speeds are due to the sampling bias of the uncertainties and do not reflect on the individual measurement's estimate of the random uncertainty.

342 – It is a very large uncertainty, no? Can you find a reference of some organization or a study which recommends a minimum accuracy of AOD retrieval like WMO or something like this and explain how your method can satisfy minimum accuracy requirement for aerosol retrievals of this type?
The uncertainty being discussed here is only the precision or random uncertainty and there is no minimum precision requirement for CALIOP reported ocean derived full column optical depth retrievals. Additionally, as optical depths become larger, this random uncertainty does not change much so relative to the measurement it becomes small. The accuracy of the measurement is much harder to quantify and would require a more significant validation study. None-the-less, we attempt to give an indication of the accuracy of the retrievals with the performance assessments.

346 – MERRA-2 is a model and they do not provide instrumental uncertainties. I do not think that your uncertainty assumption is valid here. Put it simply, 2005 was almost 20 years ago and ocean wind speed measurements have been significantly advanced since then; see all scatterometer-based studies available in the literature. The modern requirements of wind speed uncertainties is around 3 m/s. Coming back to the modelling data, their uncertainties can be normally inferred from the bias estimation-focused studies performed using comparison of modelled data versus true measurement data.
The CALIPSO data record, over which the ODCOD algorithm is applied, spans the time period from mid-June of 2006 to August 1st, 2023 and as a result uses data from all of the instruments ingested into the MERRA-2 reanalysis data including AMSRE so, while the assumptions might not be the most up to date for cutting edge measurements, they are relevant to the data and time periods the CALIOP project is using. Additionally, these are

the estimates used in the ODCOD algorithm and cannot be changed so any argument for updated assumptions would be best suited for a validation paper and discussions of a future version of the algorithm.

365 – Show this comparison or refer to previous studies where you have done it please. The intent of this statement was not to suggest we had performed an analysis but to indicate that we were not aware of any significant biases between the two instruments based on our literature search at the time of development (circa 2022) and since the instruments are very similar in design, we elected to combine the datasets as described. Performing this analysis after the fact will have no bearing on what has already been done but would be a topic that could be explored in a validation paper. To better express what was done, we have changed the lines to, "*The AMSRE and AMSR2 instruments are very similar in design and no significant bias between their wind speed estimates was noted by a search of the available literature at the time of ODCOD development.*"

369 – Give references to these products as well as to GMAO MERRA-2 10 m wind speeds. Each of the data products used in the production of ODCOD and the performance assessments conducted for the paper are cited in the data availability section. Since the MERRA-2 data is reproduced in the CALIPSO data products the MERRA-2 data access was not included but has been added with a note that the data used for ODCOD is taken from the CALIPSO data products. "*MERRA-2 wind speed data is reproduced in the CALIPSO level 2 5 km merged layer product and is used from within that data product but can be accessed at MDISC (https://disc.gsfc.nasa.gov/datasets?project=MERRA-2.) and is managed by the NASA Goddard Earth Sciences (GES) Data and Information Services Center (DISC).*"

379 – AMSR does not retrieve wind speeds over land, no? That is correct which is why we had to average over larger gird cells to get over land values. We built a lookup table that was interpolated to provide a value anywhere on the globe even though ODCOD is only over ocean. In hindsight, populating the over land values with simply zero i.e no correction may have been a better choice, but we are reporting what ODCOD does rather than what it could or should do.

405 – What are the consequences of not including this uncertainty. What if one suggests it is critical? In that case, the uncertainty reported might be too small but mathematically, these uncertainties will be small because the whitecap backscatter reflectivity only begins to affect the overall backscatter reflectivity above wind speeds of approximately 8 m s$^{-1}$ and above that, a maximum of ~20% of the overall backscatter reflectivity at 15 m s$^{-1}$.

420 – What about potential attenuation of surface return by thick clouds or very high aerosol conditions? See the major comment about the methodology above in this document. Attenuation from any particulate source, be it cloud, or aerosol is the quantity of interest and is what is being retrieved by ODCOD. If the attenuation of the signal that reaches the

surface causes the surface return to not be detected or the light is fully extinguished by attenuation, no ODCOD retrieval is attempted. If the surface is detected, but the signal is very small, certainly the noise would be larger but, with aerosol only profiles, the fraction of nearly opaque aerosol only profiles is very small.

454 – 470 This information looks like it fits methodological description
We do not want to mix the ODCOD methodology with the performance assessment methodology. This information is discussing how we selected and filtered the resulting ODCOD retrievals to compare with the other datasets in the performance assessment. So, we believe it belongs here in the Data Selection section.

477 – In peer-review, "significantly" in most cases means statistical term. Also, did I get it right that you filtered out all winds of < 3 m/s from your analysis. If yes, please mention this in the methodology as well.
We have removed the word "significantly". Filtering of winds < 3m s$^{-1}$ and greater than 15 m s$^{-1}$ is not part of the ODCOD methodology. It is a recommendation to researchers for filtering and what we did for the data used in the ODCOD performance assessment.

487 – 490 You stated here that you applied statistical filtering of the data instead of physical-based filtering because "surface saturation can still go undetected", but value-based thresholds are introduced below without taking physical attenuation into account. Can you comment on this in the major response to the major comment about the methodology I placed above?
Physical attenuation is the primary reason the surface return does not saturate. It is the attenuation that allows the surface return to be a useful quantity. Additionally, the data used here to determine the threshold are for cloud-free scenes.

505 – During study period or for which period?
All data used in the performance assessment is from the same periods unless explicitly stated otherwise. In the paper we comment, "*Unless otherwise stated, all ODCOD data in the comparisons in Section 3.2 is from March 2008 through February 2011 and are the latest version 4.51 Lidar Level 1 (LL1) and V4.51 LL2 data products.*"

509 – What is SIDR? Do not use unexplained acronyms please
Our introduction of the term SIDR is in section 3.1.1, "*surface integrated depolarization ratio 532 nm (SIDR) < 0.05 improves...*"

517 – How have you come to the conclusion about the slight wind dependance?
Inspection of Fig. 8 shows a distribution that appears to be higher on the right side of the plot. This is not a conclusion and is merely an observation about the plot. Since this statement is not relevant to the ODCOD filtering recommendation, we have removed the observation.

529 – 537 – A general reader would not understand what is "averaging the surface return before retrieving". You averaged SIAB before retrieving what? AOD? I assume you mean that signal-to-noise ratio of horizontally averaged SIAB is higher and is therefore more applicable to be used in the final AOD retrieval, thus reducing resultant uncertainties. Alas, I can only assume because you use somewhat unclear language.

We have changed the description of the average-then retrieve approach to the following, "*For the coarser resolution products (1 km and 5 km), the retrieval is applied to the surface return detected in the horizontally averaged level 1 profiles. Because the position of the ocean surface is relatively constant from shot to shot, this average-then-retrieve approach is expected to increase the SNR of the surface return data and hence yield more confident fits of the DCRM to the surface data points.*"

541 – It seems that the methodological description continues from here and much further through the results further...

In this section, we are describing the methodology for assessing ODCOD's performance compared to the HSRL. We do not introduce results for this analysis until section 3.2.3 once we have covered the assessment methodology for all assessments. By doing this, we hope that the reader can cover the results from each study without being distracted by jumping back and forth to essentially very similar data filtering for each study.

545 – "Significantly lower" -> statistically speaking? If yes, please provide arguments.

We have changed the statement to, "*The HSRL aircrafts generally fly at approximately 9 km, so when comparing AODs it is important to consider attenuation above the altitude at which the HSRL measurements begin.*"

551 – What is the difference between background particulate optical depth and aerosol optical depth (AOD)?

Background is the aerosol particulate that is below the detection threshold of the CALIOP instrument and in this case located above the HSRL aircraft. Since scenes where CALIOP detects anything above the aircraft are removed, the remaining CALIOP undetected aerosols are estimated by the procedure described.

558 – Why don't you use CALIPSO classification scenes instead of this? By applying lidar ratio assumptions, you nullify one of the best advantages of ocean surface return-based AOD retrievals using lidar – you do not normally need to assume lidar ratio. Thus, no bias contribution stemming from this aspect is expected. Am I wrong?

CALIOP does not retrieve background aerosol optical depths except in the level 3 data products which are monthly averages and thus don't give an estimate at the time of measurement. Also, level 3 monthly estimates of background aerosols make the same lidar ratio assumption. The only lidar ratio assumption here is for the very tenuous background aerosols and the correction is only on the order of 0.018±0.005. In contrast, if we instead used a lidar ratio assumption of 29, the correction would be 0.009 ± 0.003 as stated in the manuscript which gives a lower bound on any uncertainty this assumption might add.

570 – Did I miss the description of cloud filtering using CALIPSO data?
In section 3.1.1 we discuss how we filtered the data for the ODCOD performance assessment, "*To assess ODCOD AOD retrievals, profiles are selected in which CALIOP has not detected clouds at any resolution.*"

577 – General users have no idea what is 'confidence flag' in MODIS data.
We have changed section 3.1.3 to include information about the MODIS quality assurance flags as follow, "*The MODIS Quality_Assurance_Ocean science data set (SDS) is used for MODIS data quality screening. This SDS is a 5-byte composite informational flag that includes a retrieval QA Confidence flag (QAC) and a QA usefulness flag (QAU) (Levy et al., 2009). The possible QAC flag values are 0 to 3 and indicate confidence levels that are poor, marginal, good, and very good, respectively. The QAU flag values can be either 0 or 1 and indicate not useful and useful data. MODIS data are chosen such that the QAC flag is marginal or better and QAU flag indicates a useful retrieval (Levy et al., 2009).*" As well as included the reference in the references section. As follows:

*Levy R., Remer L., Tanré D., Mattoo S., and Kaufman Y.: Algorithm for Remote Sensing of Tropospheric Aerosol Over Dark Targets from MODIS: https://atmosphere-imager.gsfc.nasa.gov/sites/default/files/ModAtmo/ATBD_MOD04_C005_rev2_0.pdf, last access: 5 June 2024, 2009.*

585 – I think you missed a better moment to familiarize your reader with SODA-alike approaches and other lidar surface return-based approaches in the introduction.
Because we are explicitly comparing ODCOD's performance to SODA, we are going into some detail here about the SODA technique so that we may compare the result we will present in the performance assessment results section. Discussion on other lidar surface return-based approaches is limited to the introduction.

600 – I think some kind of diagram, showing how you filter your data and which (1) satellite, (2) parameter, (3) criterion applied to this parameter you use is needed to understand the pipeline of filtering.
Thank you for the recommendation. Since this is just basic filtering for SODA-specific analysis which is only a portion of the performance assessment and could be summarized as, "we used good SODA data," we feel the description is simple enough to follow and another diagram would add length to the paper that is not needed. We have made the addition of adding the actual SODA product SDS names so that a researcher might better find the data used for this filtering by changing lines in section 3.1.4 to, "*The SODA Scene_Flags and QA_Flags SDSs are informational and quality assurance flags reported in the SODA data products.*"

601 – 610 – These remarks about clouds sound irrelevant, no results about AOD were shown yet.

We have moved the comments regarding clouds to the performance assessment summary section.

614 – 620  I see two problems here. First, you refer to some consistency of AOD with MODIS. Numerical agreement analysis is missing though. Does the consistency mean that AOD is distributed in slightly similar way to MODIS? If yes, where the agreement is highest, where is the lowest? Can you elaborate? Ideally, there should be numerical agreement analysis. At least, you should provide articulated textual description on where you see highest agreement with MODIS, where it is lower. Can you add a figure, to which you refer to while talking about agreement (from Remer et al., 2008 as I assume). Second you tailor these AOD anomalies to hand-picked aerosol-related events from the literature. However, how did you ensure these are the exact phenomena you are talking about? If I follow the same rationale, I can say, there are frequent biomass burning events in Amazon as well, but do we see their effects?

Indeed, a quantitative comparison of ODCOD AOD values and the AOD values of Remer et al. (2008) have not been performed. Therefore, the sentence with that reference is removed from the manuscript. The locations and seasonality of the aerosol phenomena described are well established in the literature, as noted by the references which discuss their behavior. The language has been modified to say that the locations and seasons of enhanced ODCOD AOD are qualitatively consistent with the locations and seasons of these phenomena. It therefore no longer implies that these phenomena are the cause of the enhancement, it just says that these are the locations and seasons where such enhancements would be expected. Proving that these enhancements are caused by the indicated phenomena is a validation exercise that is outside the scope of this manuscript.

630 – Does it mean we are talking about bias in your algorithms? I still do not understand how you are going to address the actual bias without introducing a validation AOD dataset like MODIS or CALIPSO original AOD retrieval or AERONET if some cases from islands are applicable for such comparison. Can you reflect on this?

The daytime nighttime differences being discussed are not an algorithm bias but a sampling bias when attempting to gather data for statistics. This is discussed in detail in the next two pages. The sampling bias is due to the preferential inability to retrieve (or reduced quality of the retrieval) of ODCOD in profiles where the atmosphere is exceptionally clean at night compared to daytime due to different detector gain settings. Any single measurement bias due to surface saturation will either not exist because ODCOD was not attempted or should be filtered out when filtering for the best quality data by the SIAB filters described. To help highlight this, we have added, "*If CALIOP's detectors did not saturate, then sampling opportunities would be equal day to night.*" This paper is not attempting to address any specific biases as it is not a validation paper. What this paper does provide is a performance assessment that shows in general how ODCOD performs generally compared to HSRL, MODIS, and SODA. In the future, a validation paper should be written to investigate bias in the ODCOD method presented in this paper.

633 – 635 What does the fact about statistically significant difference between night and day observations? Time-driven bias also? If it is a serious issue you are addressing in your paper, I think you need to make a sub-section of ODCD called 'Nighttime vs daytime measurements'.

The statistical significance comment is meant to show that the median optical depths day and night are in fact not the same due to the sampling bias mentioned and that the difference is approximately between the confidence interval of 0.026 and 0.027. Most of the bias is not time-driven but rather instrument driven due to the different sensitivity settings day and night. This is not a "serious issue" except if a researcher were to naively aggregate data for statistics without considering this sampling bias and how it affects where ODCOD successfully retrieves data. Any bias day to night that is not due to the sampling bias would require significantly more research because the 0.010 ± 0.006 difference left after the sampling discussion might be due to natural variability, sensor calibration, or algorithm inputs just to name a few.

660 – 667 – "Some good quality low daytime data" -> please explain how much to make your experiment reproducible. Also, it is desirable to illustrate in in figure or tabular form not just mention it in the text. The same applied so "modified the surface saturation filter" -> modified to what value? Only one value or a range of values to estimate an overall sensitivity? Please clarify.

The amount of data removed is not as significant as what data is removed which is any retrievals where the daytime SIAB is greater than the 0.0353 sr-1 threshold. Incidentally it is about 20% which shows just how much more data is saturated at night than in the day. Figure 13 is a map showing the comparison results of filtering in this way. We have changed the line in section 3.2.1 to, "*To demonstrate the impact of the surface-saturation sampling bias, we experimentally modified the surface saturation filter described in Sect. 3.1.1 to use the nighttime SIAB threshold of 0.0353 sr-1 for both day and night observations.*"

669 – Did you mention original resolution of ODCOD before in the text (like methodology)? Please point out where, if not available, the description of the resolution choices should be provided in the methodology.

ODCOD does not have an "original" resolution. ODCOD is a method applied to CALIOP surface return measurements of attenuated backscatter. The attenuated backscatter data is nominally provided at CALIOP resolutions of 333m, 1km, and 5km horizontally averaged resolutions and ODCOD is applied to each and provided to the user. We do not feel this is a methodology topic because the ODCOD algorithm is resolution agnostic. It is important to address that ODCOD is provided at these three standard CALIOP resolutions and show consistency between them which is why it is noted in this section. We have changed the line in section 3.2.2 in the new text to, "*ODCOD is reported in the CALIOP LL2 data products at the standard CALIOP horizontal averaging resolutions of single shot (333 m), 1 km, and 5 km resolutions.*" to attempt to clarify the resolutions ODCOD is applied to.

671 – Is it a common procedure to average SIAB horizontally? Provide some arguments if no, provide some references to previous CALIPSO-based studies if yes.

The horizontal averaging that ODCOD uses is dictated by the standard CALIOP averaging regimes of 333 m, 1 km, and 5 km. We have changed the beginning of the paragraph in section 3.2.2 to make it clearer:

> *ODCOD is reported in the CALIOP LL2 data products at the standard CALIOP horizontal averaging resolutions of single shot (333 m), 1 km, and 5 km resolutions. For the coarser resolution products (1 km and 5 km), the retrieval is applied to the surface return detected in the horizontally averaged level 1 profiles. Because the position of the ocean surface is relatively constant from shot to shot, this average-then-retrieve approach is expected to increase the SNR of the surface return data and hence yield more confident fits of the DCRM to the surface data points.*

680 – Your language is unclear here, can you reformulate this sentence?
We have changed the line to:

> *Because the position of the ocean surface is relatively constant from shot to shot, this average-then-retrieve approach is expected to increase the SNR of the surface return data and hence yield more confident fits of the DCRM to the surface data points. A retrieve-then-average schemes can offer an alternative to the average-then-retrieve approach. However, care must be taken not to bias the estimate by assuming missing retrievals are like surrounding retrievals or worse an optical depth of zero. A common reason for a missing ODCOD retrieval is no surface return detected due to high optical depths. Assuming a value for these missing retrievals will bias the average.*

691 – You normally describe the figure first and then provide a figure itself below in peer-reviewed studies.
We have moved the figure after the description.

695 – 700 Please provide numerical arguments behind such statements as "noisy" (maybe compare their standard deviations in the text?)
The intent in this section is not to quantify the variability but rather simply highlight how the different resolutions behave compared to one another. A more quantitative analysis would be fitting in a rigorous validation paper. We have now provided some measure of the variability by applying a sliding window fit and reporting the mean squared error. The new paragraph reads:

> *In general, the retrievals show that 5 km retrievals display less variability and fall on top of the 1 km retrievals, which are again less noisy and fall on top of the single shot retrievals. After applying a 31-profile sliding window fit to the single shot data, the noise for each resolution is estimated by calculating the mean squared error (MSE) between the fit (black dashed line, Fig.14) and the data for each resolution. In the daytime, the estimated MSE for single shot, 1 km, and 5 km are 0.0031, 0.0016, and 0.0024, respectively. In this scene, the effects of the solar background radiation can*

*be seen from the larger spread of the data compared to night. The MSE at 5 km is worse than the 1 km due to the occasional outliers of the retrieved optical depth at coarser resolutions from the neighboring retrievals. These deviations occur due poor fit of the CRM to the surface return due to a difference in altitude registration of the surface return which causes a deformation of the expected surface return shape and thus poor fit to the DCRM. This effect is not isolated to daytime only scenes and can be identified by increased uncertainty compared to neighboring retrievals. The estimated MSE for nighttime single shot, 1 km, and 5 km resolutions are 0.0020, 0.00078, and 0.00015, respectively and show over an order of magnitude improvement between the single shot and 5 km resolutions.*

710 – What about correlations?

We have added correlation coefficients to the text and changed the paragraph to the following:

*In general, ODCOD 5 km retrievals show little to no bias compared to HSRL aerosol optical depth retrievals when day and night are considered together. The median difference is 0.009 ± 0.043 (6 % ± 28% relative difference; N=395) with ODCOD higher and a correlation coefficient of 0.724 and a 95 % confidence interval for the mean difference of -0.005–0.014. Separately, ODCOD estimates are relatively lower in the daytime and relatively higher at night but with uncertainties larger than the difference in either. The median difference in the daytime is -0.037 ± 0.052 (-12 % ± 25%; N=149) with ODCOD lower and a correlation coefficient of 0.775. The median difference at night is 0.021 ± 0.032 (14 % ± 25%; N=246) with ODCOD higher and a correlation coefficient of 0.721.*

725 – Statistically significant difference here is not giving any valuable information I think. The What do you intend to infer from such comparison? Also, mean differences might be not representative. You are having 3 years of global data populated with 3 months seasonally. Are not you interested in region-specific biases and correlations rather median bias?

The purpose of the statistical significance statement was to indicate that we cannot rule out a systematic bias between the instruments, but the 95 % confidence interval also makes the same statement with more information so removing the statements regarding statistical significance is warranted. We have changed the paragraph to:

*In general, the global median difference between ODCOD 5 km daytime retrievals and MODIS interpolated 532 nm AOD is -0.009 ± 0.041 (8% ± 35%; N=1,999,068) with ODCOD lower and with a correlation coefficient of 0.834. Regionally, ODCOD tends to report higher aerosol optical depths in the southern oceans from March through August and seems to show lower optical depths in December through February. ODCOD also tends to report higher aerosol optical depths north of 30 º N from September through February but the difference is less during March through August.*

All the analysis mentioned in the reviewer's comment is interesting and would be useful in a validation paper. Since the focus of this paper is only to introduces the method used and provide a minimum analysis to give the reader an understanding of how the dataset compares globally to other established datasets, additional analysis will need to be in a future publication.

731 – Table 3 or 2? Also, why you are not interested in correlations? If your correlation is high, are not you interested in which AOD intervals or maybe regions contribute to the lack of ideal correlation, thus plaguing your linear relationship pattern between ODCD and the reference?

Thank you for catching this error. The correct table is 2 and we have corrected the table number in the text. We previously had correlation coefficients only on the figures, but we have now gone back and added the general correlation coefficients to the text. Correlation speaks to how well the datasets agree with one another throughout the optical depth range and confidence interval speaks more to the magnitude of any systematic bias between the datasets. Both are useful for understanding ODCOD compared to others, so we are happy to provide both.

770 – 784 – Likewise, stick to correlations, seasonal biases, regional differences, not to long-term mean differences.

We have added general correlations to the text. Seasonal correlations can be found on the plots and within Table 3. Additional analysis regarding regional bias is reserved for a dedicated validation paper. We have attempted to quantify the random uncertainty with the reported ODCOD uncertainties and the discussions in Sect. 2.2. Long-term mean differences speak to the larger systematic biases between the datasets by reducing the random uncertainty through data volume. By analyzing the data in this way, we have highlighted both the random and systematic uncertainty that may be present in ODCOD but without a known truth or a much more detailed analysis, which is outside of the scope of this paper, doing more than stating the bias between ODCOD and a few established datasets is all we will provide. However, even without a quantitative analysis of the systematic biases in ODCOD, the information provided here is enough to provide the reader with an understanding of how ODCOD performs in general.

The new lines read:

*In general, daytime ODCOD 333 m retrievals show relatively small differences globally compared with SODA 333 m aerosol optical depth retrievals. The daytime median difference is 0.004 ± 0.035 (1 % ± 34% relative difference; N=21,270,202), ODCOD higher, with a correlation coefficient of 0.887. At nighttime, the median difference is 0.027 ± 0.034 (20 % ± 33% relative difference; N=10,536,357), ODCOD higher with a correlation coefficient of 0.879. Unexpectedly, SODA reports similar values both day and night with global median values of 0.102 ± 0.045 daytime and 0.105 ± 0.045 nighttime.*

787 – What is anomalous SODA data from numerical point of view? Indicate a range here at least

The anomalous data occurs across the full range of optical depths from -1.5 to 1.5. It only becomes apparent when binned by ODCOD optical depth. The lines in section 3.2.5 attempts to provide this information with, "*This artifact becomes apparent when plotting ODCOD as a function of SODA, as the anomalous points form striated lines in what appear to be somewhat quantized groupings, many of which are relatively large negative values in Fig. **Error! Reference source not found.** and Fig. **Error! Reference source not found.**. Preliminary investigations indicate that one primary cause of these SODA outliers is the inadvertent use by the SODA algorithm of CPR data acquired during CPR calibration maneuvers (Tanelli et al., 2008).*" Any value that falls outside of the defined "Tukey" fence as stated in the text.

823 – Result summary is a very strange name of the section. Normally, you would have "Discussion" and then "Conclusions" or "Summary" then "Conclusions" or "Discussion and Conclusions" at once.

We have changed the section title to "*Performance Assessment Summary*."

830 – I would still call it discussion...

In the interest of reducing the significant length of the paper, we have elected to remove the Future Work with CALIOP section. The comparisons to CALIOP are interesting but are not useful in providing confidence or information regarding ODCOD as a data product.

830 – 854 This is a very weird paragraph. First, you say that future work can focus on comparing ODCOD with CALIOP standard AOD profiles? Or standard ODCOD-alike SIAB-based AOD profiles? Unclear here. If these are your future plans, just move them to discussion shortly.

We have removed this section.

855 – 910 Please move this comparison to validation section is possible. The structure of the manuscript becomes non-conventional here. You showed results, then went to future work discussion and now showing some new results again.

We have removed this section.

913 – Once again, it's unclear why this development is different from He et al. 2016 or Venkata and Reagan 2016 attempts? Apart from the fact that it will be the official Level 2 data product

Hopefully we have now addressed this with the previous comments, and it is now clearer.

915 – Multiyear -> name exact years

The exact years can be found in numerous places within the manuscript, and explicitly in Sect. 3.1.1 and are not as relevant here. Since this is the conclusions, we wish to highlight

only the important topics touched on by the paper. Namely, the algorithm and what it provides, in general how it compares internally and to other datasets, estimates of the random uncertainty, and some closing remarks.

916 – Once again, I think statistical significance in the difference between these datasets make no sense from remote sensing point of view. You can easily have two similar AOD density distributions with statistically insignificant differences (because AOD is normally distributed across the globe within the same range), but with great biases (due to regional differences) and very low correlation between each other.
To improve the conclusions, we have re-written part of the opening paragraph of the conclusions section to the following.

*CALIPSO's Version 4.51 Lidar Level 2 data products report a new estimate of full column effective optical depth retrieved from the ocean surface lidar backscatter return by the Ocean Derived Column Optical Depth (ODCOD) algorithm. Accurate estimates of the ocean surface integrated attenuated backscatter (IAB) are obtained by fitting a model of CALIOP's expected ocean surface return shape to the 532 nm surface return measurements. Particulate two-way transmittances, from which optical depths are derived, are retrieved by scaling the estimated IAB to an unattenuated modeled surface reflectance that has been corrected for molecular and ozone two-way transmittances. ODCOD total column optical depth estimates are derived for the entire CALIPSO data record wherever qualified ocean surface detections are made.*

*Relative to daytime retrievals, ODCOD nighttime AOD estimates tend to be higher; however, in-depth global comparisons are hindered by the lack of well understood and validated nighttime data derived from other sensors. ODCOD retrievals in the daytime were compared to 10 collocated airborne HSRL underflights, 3 years of MODIS AODs interpolated to 532 nm and the ODCOD retrieval location, and 3 years of collocated SODA 333m retrievals. The median daytime differences found were -0.037 ± 0.052, with ODCOD lower than HSRL; -0.010 ± 0.041, ODCOD lower than MODIS; and 0.004 ± 0.035, ODCOD higher than SODA. Correlation coefficients were found to be 0.775, 0.834, and 0.887 respectively. Nighttime retrievals of 11 HSRL underflights and 3 years of SODA data showed median differences of 0.021 ± 0.032 and 0.027 ± 0.034, both with ODCOD higher and correlation coefficients of 0.721 and 0.891 respectively. However, the expected sampling bias between daytime and nighttime data, inherent in all CALIOP-based surface return optical depth estimates and seen in ODCOD, is not found in the SODA datasets. This apparent bias in the SODA data may explain the larger differences between the two techniques. Different CALIOP amplifier gains during the daytime and nighttime portion of the orbit cause the lidar surface return to saturate more frequently at night; however, the lack of solar background also allows the surface to be detected more readily when the surface return is very small. Since ODCOD requires unsaturated surface detections, both effects will cause sampling biases where aggregated average ODCOD optical*

*depths are typically higher at night than day. However, these sampling biases does not account for differences between datasets when compared on a profile-by-profile basis.*

931 – 940 Do not overgeneralize the conclusions about your own method (which you seemingly do according to ample references provided here) using very general rationale based on previous methods. Just state shortly why your attempt of using ODCOD is successful and useful for future studies? This conclusion should be based on the results of YOUR study and not the general benefits of any SIAB-based AOD retrieval for CALIPSO. We have removed the sentence in the paragraph, "ODCOD provides an internally consistent constraint for deriving extinction profiles that, on average, may be improved over the standard CALIOP profiles (Burton et al., 2010; Painemal et al., 2019; Li et al., 2021). ODCOD AOD retrievals are also being assessed for estimating regional layer-averaged lidar ratios for CALIOP by the Models, In situ, and Remote sensing of Aerosols (MIRA) group (Trepte et al., 2023)" and revised the conclusions.

---

## Author Comment (AC2)

This work basically documents an algorithm that is now being used to produce aerosol column optical depth over oceans from CALIPSO surface returns and MERRA-2 10m wind speed. The first part that describes the method is really more suited as an ATBD (Algorithm Theoretical Basis Document) than a research paper. In general, it is a well written and thought-out paper.  A strength is that they present comparisons with MODIS, HSRL, SODA and CALISPO layer optical depths. Also important is their treatment of the CALIOP surface return, realizing that it can be saturated and using a fitting method to the instrument impulse response function to retrieve a more accurate measurement of the surface return magnitude. The filtering of suspect data based on surface signal magnitude, depolarization and wind speed is well done.

However, I think the paper is too long. At around 50 pages it becomes a very tedious read. I would suggest breaking the paper into two parts. The first would present the ODCOD algorithm and the uncertainty analysis and the second part would present results and comparisons. This would allow a better and more in-depth description of the surface return fitting to the IRF (which I think at present is confusing and the results paper to include more examples and comparisons. A comparison that should be added are some island-based or costal AERONET comparisons of column optical depth.

Thank you for the thorough review and for the encouraging words. It has been the practice of the CALIPSO team to introduce new as well as changes to existing data products via papers submitted to journals rather than changes to the CALIPSO ATBDs. See Getzewich et al. 2018, Hu et al. 2009, Kim et al. 2018, Omar et al. 2009, Powell et al. 2009, Vaughan et al. 2004, Vaughan et al. 2019, Winker et al. 2009, etc. and we hope with the changes we are proposing based on the reviewer's comments that this paper is now less ATBD-like and will be a fitting addition to the AMT journal.

We agree with the assessment that the paper is long. We also recognize that there is a need for a full validation of the ODCOD algorithm however, this manuscript is not attempting to fulfill that need. Instead, this paper intends to describe the technique and its inputs, to quantify the random uncertainties and describe their estimation, and then to finally provide only an initial assessment for the performance of the ODCOD algorithm by considering global systematic differences to other established datasets. We specifically chose the airborne HSRL for its accuracy, MODIS for its well validated and long running global aerosol optical depth record, and SODA for its nighttime data, matched footprint, and similar retrieval technique. While AERONET comparisons are crucial to validating ODCOD, this paper is already long, and additional analysis would delay publication of the algorithm technique further from the already released data product. Also, Thorsen et al. 2024 in preparation is currently working on presenting an AERONET validation of the ODCOD data product.

To address the comments specifically, we have re-written the abstract to better state the goal of the paper and deemphasize "results," instead framing them more as what they are intended to be which is an assessment of performance and a characterization of the systematic differences to other datasets. To address the paper length, we have made the decision to remove Section 4 completely and reduced the body of the text by over 900 words. Section 4, while interesting and good to promote discussion, doesn't further the purpose of the manuscript enough to retain it. Some portions of Section 4 fit well in the introduction and we have moved important parts there. However, this along with the numerous edits based on reviewer comments has not successfully shortened the overall length of the paper. We feel that providing the algorithm to readers with no analysis into how the algorithm performs would generally not be well received. Breaking the current work into two parts would leave the analysis underwhelming without additional and more detailed validation. To perform the necessary validation work would take some time to perform and separate the release of the algorithm and validation papers by an undesirable amount of time. The length of the paper is unfortunate but with the removal of section 4 we hope each section now is an important and necessary part of the paper.

The fitting of the CALIOP response model (CRM previously called IRM) is a difficult procedure to describe succinctly but in short is done by finding the measurement time of a reference measurement by taking the ratio of that measurement and the next downlinked measurement which we are confident are part of the surface return. That ratio is unique for any time within the surface return. Since the reference measurement time is unique, it allows us to identify where within the surface return each measurement is taken. Once the positioning is known, it is possible to iteratively solve the scale of the CRM by minimizing the error between the measured points of the CALIOP surface return and points on the CRM but averaged in the same way the CALIOP measurements are averaged onboard the spacecraft. We call this mapping of the CRM to the downlink averaged samples the DCRM (downlinked CALIOP response model). We have completely re-written section 2.1to attempt to make the procedure clearer.

Getzewich, B. J., Vaughan, M. A., Hunt, W. H., Avery, M. A., Powell, K. A., Tackett, J. L., Winker, D. M., Kar, J., Lee, K.- P., and Toth, T. D.: CALIPSO lidar calibration at 532 nm: version 4 daytime algorithm, Atmos. Meas. Tech., 11, 6309–6326, https://doi.org/10.5194/amt-11-6309-2018, 2018.

Hu, Y., Winker, D., Vaughan, M., Lin, B., Omar, A., Trepte, C., Flittner, D., Yang, P., Nasiri, S. L., Baum, B., Holz, R., Sun, W., Liu, Z., Wang, Z., Young, S., Stamnes, K., Huang, J., & Kuehn, R. (2009). CALIPSO/CALIOP Cloud Phase Discrimination Algorithm. Journal of Atmospheric and Oceanic Technology, 26(11), 2293-2309. https://doi.org/10.1175/2009JTECHA1280.1

Kim, M.-H., Omar, A. H., Tackett, J. L., Vaughan, M. A., Winker, D. M., Trepte, C. R., Hu, Y., Liu, Z., Poole, L. R., Pitts, M. C., Kar, J., and Magill, B. E.: The CALIPSO version 4 automated

aerosol classification and lidar ratio selection algorithm, Atmos. Meas. Tech., 11, 6107–6135, https://doi.org/10.5194/amt-11-6107-2018, 2018.

Omar, A. H., and Coauthors, 2009: The CALIPSO Automated Aerosol Classification and Lidar Ratio Selection Algorithm. J. Atmos. Oceanic Technol., 26, 1994–2014, https://doi.org/10.1175/2009JTECHA1231.1.

Powell, K. A., Hostetler, C. A., Vaughan, M. A., Lee, K., Trepte, C. R., Rogers, R. R., Winker, D. M., Liu, Z., Kuehn, R. E., Hunt, W. H., & Young, S. A. (2009). CALIPSO Lidar Calibration Algorithms. Part I: Nighttime 532-nm Parallel Channel and 532-nm Perpendicular Channel. Journal of Atmospheric and Oceanic Technology, 26(10), 2015-2033. https://doi.org/10.1175/2009JTECHA1242.1

Vaughan, M., Young, S., Winker, D., Powell, K., Omar, A., Liu, Z., Hu, Y., and Hostetler C.: "Fully automated analysis of space-based lidar data: an overview of the CALIPSO retrieval algorithms and data products", Proc. SPIE 5575, Laser Radar Techniques for Atmospheric Sensing, https://doi.org/10.1117/12.572024, 2004.

Vaughan, M., Garnier, A., Josset, D., Avery, M., Lee, K.-P., Liu, Z., Hunt, W., Pelon, J., Hu, Y., Burton, S., Hair, J., Tackett, J. L., Getzewich, B., Kar, J., and Rodier, S.: CALIPSO lidar calibration at 1064 nm: version 4 algorithm, Atmos. Meas. Tech., 12, 51–82, https://doi.org/10.5194/amt-12-51-2019, 2019.

Winker, D. M., M. A. Vaughan, A. Omar, Y. Hu, K. A. Powell, Z. Liu, W. H. Hunt, and S. A. Young, 2009: Overview of the CALIPSO Mission and CALIOP Data Processing Algorithms. J. Atmos. Oceanic Technol., 26, 2310–2323, https://doi.org/10.1175/2009JTECHA1281.1.